# Dynamics of the Mediterranean droughts from 850 to 2099 AD in the Community Earth System Model

Woon Mi Kim[1,2] and Christoph C. Raible[1,2]

[1]Climate and Environmental Physics, University of Bern, Switzerland
[2]Oeschger Centre for Climate Change Research, University of Bern, Switzerland

**Correspondence:** Woon Mi Kim (woonmi.kim@climate.unibe.ch)

**Abstract.**

In this study, we analyze the dynamics of multi-year long droughts over the western and central Mediterranean for the period of 850 - 2099 AD using the Community Earth System model version 1.0.1. Overall, the model is able to realistically represent droughts over this region, although it shows some biases in representing El Niño Southern Oscillation (ENSO) variability and
mesoscale phenomena that are relevant in the context of droughts over the region.

The analysis of the simulations shows that there is a discrepancy among diverse drought metrics in representing duration and frequencies of past droughts in the western and central Mediterranean. The self-calibrated Palmer Drought Severity index identifies droughts with significantly longer duration than other drought indices during 850 - 1849 AD. This re-affirms the necessity of assessing a variety of drought indices in drought studies also in the paleoclimate context.

Independent from the choice of the drought index, the analysis of the period 850-1849 AD suggests that Mediterranean droughts are mainly driven by internal variability of the climate system rather than external forcing. Strong volcanic eruptions show no connection to dry conditions but to wet conditions over the Mediterranean. The analysis further shows that Mediterranean droughts are characterized by a barotropic high pressure system together with a positive temperature anomaly over central Europe. This pattern occurs in all seasons of drought years, with stronger amplitudes during winter and spring.

North Atlantic Oscillation (NAO) and El Niño Southern Oscillation (ENSO) are also involved during Mediterranean multi-year droughts, showing that droughts occur more frequently with positive NAO and La Niña-like conditions. These modes of variability play a more important role during the initial stage of droughts. Then, the persistence of multi-year droughts is determined by the interaction between the regional atmospheric and soil moisture variables, i.e., the land - atmosphere feedbacks, during the transition years of droughts.

These feedbacks are intensified during the period 1850 – 2099 AD due to the anthropogenic influence, thus reducing the role of modes of variability on droughts in this period. Eventually, the land - atmosphere feedbacks induce a constant dryness over the Mediterranean region for the late 21st century relative to the period 1000 – 1849AD.

# 1 Introduction

Drought is an extreme weather and climate event characterized by a prolonged period with persistent depletion of atmospheric moisture and surface water balance from its mean average condition. Drought is also characterized by a slow onset and devastating impacts on society, the economy and the environment (Wilhite, 1993; Dai, 2011; Mishra and Singh, 2010), and it can be classified in four types: meteorological drought, associated with the decrease in precipitation; agricultural drought, associated with the depletion of soil moisture and impacts on crops and plants; hydrological drought, characterized by the depletion of streamflow and water reservoirs, and lastly socio-economic drought, that occurs when the other types of droughts cause impacts on society, in a way that the water supply cannot meet the demand from society (Mishra and Singh, 2010). If a meteorological drought lasts for a longer period, it has the potential to propagate to other types of droughts, such as agricultural or hydrological drought. In this sense, different types of droughts can become connected to each other. Thus, meteorological drought is one of the causes of other types of droughts, among other processes such as seasonal changes of run-off or an increase in evapotranspiration demand (Wang et al., 2016; Zhu et al., 2019). The severity and duration of a drought can be quantified through different indices that capture hydrological conditions associated with a regional water balance (Dai, 2011). However, a single universal index cannot characterize the entire complex nature of droughts (Lloyd-Hughes, 2014) and the connection among different types of droughts (Mukherjee et al., 2018). Thus, one index does not necessarily show a similar value to other indices even for the same region and period (Raible et al., 2017; Mukherjee et al., 2018). Some of the widely used indices are the self-calibrated Palmer Drought Severity Index (Wells et al., 2004), the Standardized Precipitation Index (McKee et al., 1993), and the Standardized Precipitation Evapotranspiration Index (Vicente-Serrano et al., 2009), among many others.

The Mediterranean region is known as a climate change hot-spot (Giorgi, 2006), i.e., the region is highly responsive to current and future global warming, showing a decrease in precipitation and an increase in drought episodes (Dubrovský et al., 2014; Liu et al., 2018). The climate of the Mediterranean is characterized as semi-arid with a pronounced annual cycle, which means a high temporal variability of the availability of water resources (Lionello et al., 2006). Therefore, droughts or periods with scarcity of water are intrinsic parts of the climatic conditions over Mediterranean. Overall, the region shows mild and wet winters, and hot and dry summers (Lionello et al., 2006). The variability of precipitation is not uniform across the entire Mediterranean. The western and eastern regions show different precipitation regimes, in particular in winter. A regional mode of circulation that explains this spatial difference is the Mediterranean Oscillation characterized by the opposite pressure and precipitation patterns between the west-central and eastern region (Dünkeloh and Jacobeit, 2003). Besides, the regional precipitation is strongly influenced by the mid-latitude storm tracks and cyclones, which become stronger during the winter (Lionello et al., 2016; Raible et al., 2007, 2010; Ulbrich et al., 2009), regional cyclones (Alpert et al., 1990), and large-scale modes of variability, such as the North Atlantic Oscillation (NAO), East Atlantic - West Russian pattern (EA - WR) and El Niño-Southern Oscillation (ENSO) (Lionello et al., 2006; Raible, 2007). The influence of these large-scale patterns varies within the Mediterranean region. The NAO exerts its control on precipitation by affecting the strength of westerlies and latitudinal movement of storm tracks. Precipitation decreases during the positive phase of the NAO, mostly in the west-central Mediterranean, whereas it increases during the negative phase of NAO (Wallace and Gutzler, 1981; Hurrell, 1995). The EA-

WR influences the southeastern Mediterranean hydroclimate causing drier conditions during its positive phase (Barnston and Livezey, 1987; Krichak and Alpert, 2005). The response of the Mediterranean climate to ENSO is more complex: it varies over time, as illustrated by historical cases (Brönnimann, 2007; Brönnimann et al., 2007), it depends on the maturity of the ENSO state (Vicente-Serrano, 2005), on the seasons (Mariotti et al., 2002) and on the co-occurrence with NAO (Brönnimann, 2007; Raible et al., 2001, 2003). Mariotti et al. (2002) demonstrated that precipitation decreases over the western Mediterranean during La Niña in autumn and spring. Brönnimann (2007) showed a connection between La Niña (El Niño) and the positive (negative) phase of the NAO in the late winter.

In the Mediterranean, the increases in the severity and number of droughts have been already observed since the mid to late 20th century (e.g., Mariotti et al., 2008; Philandras et al., 2011; Sousa et al., 2011; Seager et al., 2014; Vicente-Serrano et al., 2014; Spinoni et al., 2015). In recent decades, the occurrence of droughts with a pan-European characteristic that cover a large part of the west-central Mediterranean region have been also detected (García-Herrera et al., 2019; Spinoni et al., 2017). The increase in dryness is attributed to the increase in the atmospheric greenhouse gases (GHG) concentrations, which causes a strong increase in the surface temperature and a decrease in precipitation over this region (Mariotti et al., 2008). General circulation models (GCMs) project that this drying trend together with the increases in dry days and drought episodes will be intensified in the future under the business-as-usual scenario, causing substantial socio-economic impacts and changes in the region (Mariotti et al., 2008; Field et al., 2012; Lehner et al., 2017; Naumann et al., 2018). The future changes in the Mediterranean droughts are due to increasing tropical SSTs (Hoerling et al., 2011), changes in the mean regional circulation associated with intensified subsidence and low-level mass divergence (Seager et al., 2014), the expansion of the Hadley cell and of the subtropical subsidence zones (Previdi and Liepert, 2007), an intensification of subtropical highs (Li et al., 2012), and a northward shift of the storm tracks (Raible et al., 2010).

Although the dryness projected in the future scenarios is unprecedentedly intense, multi-year dry periods are not a completely new phenomena over the Mediterranean. Using the summer self-calibrated Palmer Drought Severity Index (scPDSI) based on tree ring reconstructions (also known as the Old World Drought Atlas; OWDA; Cook et al. 2015), Cook et al. (2016a) found that the region has experienced several dry periods during the last 900 years, some with persistent pan-Mediterranean characteristics. The variability of these Mediterranean droughts shows the frequencies of not only interannual, but multidecadal timescales.The causes of these past droughts are still unclear, but a connection to large-scale patterns, such as the NAO, Eastern Atlantic and Scandinavian patterns may be relevant.

Besides paleoclimate proxies, GCMs have been used to study long-term changes and continuous variability of global and regional hydroclimate and extreme events during the last millennium (PAGES Hydro2k Consortium, 2017; Haywood et al., 2019). Modelling studies on the long-term variability of droughts are focused on the U.S continent, mainly to investigate the variability and mechanisms of South Western United States (SW) droughts (Coats et al., 2013, 2016; Parsons et al., 2018; Parsons and Coats, 2019) and North American pan-continental droughts (Coats et al., 2015; Cook et al., 2016b). The results show that different GCMs are able to reproduce the duration and intensity of SW megadroughts and North American pan-continental droughts. The authors further suggest that internal variability is the main driver of these droughts, although the specific processes are largely model dependent.

Globally, Stevenson et al. (2018) used the Community Earth System Last Millennium Ensemble (CESM-LME; Otto-Bliesner et al. 2016) to examine the connection between past global hydrological mega-events, and climate variability and external forcings during the last millennium. Among the major modes of climate variability, ENSO and AMO are identified to have an influence on mega-events; both modes significantly alter the megadrought risks and persistence in drought-prone regions, such as the southern Australia, the Sahel and the southern United States. The study provides insights into the dynamic of megadroughts associated with different mode of variability on global scales. However, a detailed analysis on Europe and the Mediterranean is missing.

Over the European domain, Ljungqvist et al. (2019) examined a long-term covariability between the summer temperature and hydroclimate during the Common Era. By comparing the instrumental records, tree ring-based reconstructions, and model simulations, they found that a warm-dry relationship with multi-decadal variability is more dominant in the southern Europe. Though, all datasets share some common leading modes of covariability across different time frequencies, there are some discrepancies among instrumental records, proxies, and model. The proxies present a stronger positive temperature-hydroclimate relationship, while the model exhibits a stronger negative relationship than the instrumental records. Xoplaki et al. (2018) investigated the interaction between the past central and eastern Mediterranean societies and the hydroclimate conditions including droughts, by comparing the historical records, proxies and GCM simulations. Analyzing three particular historical periods, they concluded that the multidecadal variability of precipitation in the region is driven by internal dynamics of the climate system: large discrepancies between the model trajectories are detected. Therefore, no agreement in timing between models-proxies-historical records can be expected. Nevertheless, the models elucidate some possible explanations about the dynamics of extreme dry and wet events in some past periods.

Despite a number of studies on past hydrological variability, a long-term continuous perspective on the mechanisms of past extreme hydrological events, specifically of droughts over the Mediterranean during the last millennium, is still missing. As a long trend of dryness has already been detected in the instrumental era and is expected to intensify in the future scenario, it is necessary to provide a long-term picture on the variability and changes of past dry events and their mechanisms. Therefore, we aim to examine the physical mechanisms involved in yearly and multi-year long droughts during the Common Era (850 - 1849 AD) and historical, present and future periods (1850 - 2099 AD) over the west-central Mediterranean region. We choose this specific area, as this region has been affected by recent large scale droughts (García-Herrera et al., 2019; Spinoni et al., 2017), which can be seen as pan-west-central Mediterranean droughts. Moreover, the region shows coherent desiccation in the future scenario (Dubrovský et al., 2014). From now on, for simplicity, we refer to the west-central Mediterranean region in our study simply as the Mediterranean region. We focus on understanding the dynamics that induce past persistent pan-regional multi-year droughts, and whether the dynamics that induce droughts in the past will change in the historical and future periods with the anthropogenic increase in GHG.

For our purpose, we use the Community Earth System Model version 1.0.1 (CESM), which includes the active biogeo-chemical cycle and has a horizontal resolution of $1.25° \times 0.9°$ (Lehner et al., 2015). The spatial resolution of the model is a clear advantage for our study on a relatively small confined area. The precipitation of the region is strongly influenced by extratropical cyclones and in general, GCMs have difficulties in reproducing the dynamics and precipitation associated with

these meso-scale phenomena (Raible et al., 2007; Watterson, 2006). Nevertheless, these atmospheric dynamics and precipitation are better represented in GCMs with finer spatial resolutions (Champion et al., 2011; Watterson, 2006). Hence, using a model that provides a seamless simulation for the period 850 – 2099 AD with a relatively finer spatial resolution can improve a representation of precipitation related processes, thus, drought associated mechanisms over the region.

This paper is composed of the following sections: in section 2, we introduce the model and simulations, the hydrological variables given by the model, definition of droughts, drought indices, proxy and observation datasets and methods. In section 3, we present the results of the analysis: first, we compare the simulation, proxy reconstruction and observation to validate the model simulation; second, we describe how the model depicts past droughts, whether the quantification of past droughts over the region is sensitive to the choice of drought metrics, and whether there is some possible connection between the volcanic eruptions and droughts; third, we present the climate conditions associated with the past Mediterranean droughts and their connection with regional scale circulation and modes of variability, i.e. the NAO and ENSO; lastly, we discuss whether mechanisms that induce past droughts have changed in the historical and future periods. Finally, the conclusions are presented in Section 4.

## 2   Model description and methods

### 2.1   Description of the model and simulations

We use the Community Earth System model version 1.0.1 in this study. Two simulations are used: a continuous transient simulation of 1250 years (850 - 2099 AD) and a control simulation of 400 years at perpetual 850 AD conditions (Lehner et al., 2015). In the simulations, the atmosphere has the horizontal resolution of $1.25° \times 0.9°$ and 26 vertical layers, and the land has the same horizontal resolution as the atmosphere with 15 sub-surface layers. The ocean has the horizontal resolution of $1.25° \times 0.9°$ with displaced pole grids with 60 ocean layers. The sea ice component shares the same horizontal resolution of the ocean.

The control simulation uses constant forcing parameters set to the 850 AD values: the land use changes, the total solar irradiance in which the value is 1360.228 W m$^{-2}$, the GHG concentration, such as the $CO_2$ of 279.3 ppm, $CH_4$ of 674.5 ppb and the $N_2O$ of 266.9 ppb. Unlike other forcings, the orbital parameters are set to 1990 AD conditions.

The transient simulation includes the active biogeochemical cycle and forcing, such as the land use changes, total solar irradiance, volcanic eruptions and greenhouse gases concentrations that vary over time. The GHG concentrations and land use changes vary little before 1850, showing pronounced changes and increases after that year. Then, the period 2005 – 2099 AD is run with the RCP 8.5 scenario. The transient forcing follows the third Paleoclimate Modelling Intercomparison Project (PMIP4; Schmidt et al. 2012) – fifth Coupled Model Intercomparison Project (CMIP5; Taylor et al. 2012) protocols. A more detailed overview of the forcing and initial set-up of the simulations is presented in Lehner et al. (2015).

## 2.2 Region of study, analysis and methods

The focus area of the study is the western and central Mediterranean region (15°W - 28°E and 33° - 45°N; Fig. 1). The extent of the region is selected based on Empirical Orthogonal Function analysis on the monthly precipitation from the observation (gridded station precipitation from U.Delaware v5.01; Willmott and Matsuura 2001). The region shares overall a similar variability in the first EOF (13.28%) and second EOF (11.01%), in which this similarity can also be attributed to the influence of the NAO on the precipitation over the region (Dünkeloh and Jacobeit, 2003).

For the analysis, the monthly anomalies of variables associated with the hydrological condition, such as the surface and air temperatures, precipitation, zonal and meridional winds, geopotential heights, and sea level pressure are calculated with respect to the 1000-1849 AD (850 years) mean annual cycle for each grid point in the transient simulation. For the control simulation, the entire 400 years are taken as a reference period to calculate the anomalies.

We split the transient simulation into two parts: the first period from 850 to 1849 AD is used to study the natural variability of droughts excluding the effect of an accelerated increase in the GHGs, and the second period from 1850 to 2099 AD is used to examine the effects of anthropogenic changes on the natural variability of droughts. For the first period (850 – 1849 AD), the drought condition in the transient simulation is compared to that in the control simulation to assess the influence of the natural variability and forcings. A Mann-Whitney U-test is performed to statistically compare the distributions between the transient and the control simulation. The null hypothesis of a Mann-Whitney U-test states that the distributions of both populations (in this case, the transient and the control simulations) are equal. For the second period (1850 - 2099 AD), a linear detrending method is applied to the anomalies in order to examine changes in the variability and mechanisms associated with droughts during this period. For this, the time period is split into two and the least square method is applied to each of the period separately: from 1850 to 2000 and from 2001 to 2099 AD (Fig.A1). Then, the detrended anomalies are compared to the anomalies in the non-detrended 1850 - 2099 AD and the first (850 - 1849 AD) periods.

Composites of positive and negative phases of two modes of variability, NAO and ENSO, are also investigated. The NAO is taken as the difference in the sea level pressure anomalies between the regions confined to 33° - 21°W / 35° - 39°N and 25° - 13°W / 63° - 67°N, which reflects the Azores high and the Iceland low, respectively (Wallace and Gutzler, 1981; Trigo et al., 2002). The ENSO is characterized by the annual mean sea surface temperature anomalies over Niño 3.4 region in the Tropical Equatorial Pacific (170° - 120°W and 5°S - 5°N) (Trenberth, 1997).

We further perform a wavelet coherence analysis (Grinsted et al., 2004; Gouhier et al., 2018) and a superposed epoch analysis (e.g., Malevich, 2018; Rao et al., 2017) in order to find a possible connections between droughts and volcanic eruptions. Thereby, we use the time series of the drought indices (Sect. 2.3) and of volcanic eruptions (Gao et al., 2008). Note that all time series for these analysis are normalized to have a zero mean and one standard deviation.

## 2.3   Drought definitions

We use four drought metrics to quantify droughts and to perform the comparison among them: the Standardized Precipitation Index (SPI), Standardized Precipitation Evapotranspiration Index (SPEI), self-calibrated Palmer Drought Severity Index
(scPDSI), and annual soil moisture anomaly (SOIL).

The SPI only requires a long-term precipitation record, and the accumulated precipitation is fitted to a probabilistic distribution, in our case a gamma distribution. Then, the fitted distribution is transformed to a normalized Gaussian distribution (McKee et al., 1993). The SPEI is similar to the SPI, but instead of only using a precipitation record, it considers the climate water balance given by the difference between the precipitation and atmospheric evaporative demand. This difference is fit-
ted to a log-logistic probability distribution, then, transformed to a normal distribution (Vicente-Serrano et al., 2009). For the atmospheric evaporative demands, we use the potential evapotranspiration derived from Thornthwaite equation, which only requires surface temperature and latitude (Thornthwaite et al., 1948). The scPDSI computes the water balance by assuming a two-layer soil bucket model, and it requires temperature and potential evapotranspiration records. Other necessary variables, such as runoff and losses, are estimated from the temperature and potential evapotranspiration (Palmer, 1965; Wells et al.,
2004; Zhong et al., 2018). Again, the potential evapotranspiration is calculated by using Thornthwaite equation similar as for the SPEI. The SOIL is the upper 10 cm soil moisture anomaly calculated with respect to the 850 years-mean (1000-1849 AD) annual cycles. The soil moisture is a direct output from the model.

All indices are calculated with respect to the same reference period (1000-1849 AD) and with the 12 months-annual time scale for the SPI, SPEI and SOIL. The scPDSI has an inherent time scale that ranges from 9 to 14 months depending on the
region (Vicente-Serrano et al., 2010, 2015). Thus, we use a 12 month-time scale for the other indices in order to be comparable to the scPDSI. Then, the area- weighted average of each index is calculated over the Mediterranean region (Fig. 1). The summer scPDSI is also calculated by averaging the June-July-August scPDSI, in order to compare with the summer scPDSI from the tree ring-based reconstruction, the Old Word Drought Atlas (OWDA; Cook et al. 2015).

For all indices, we define a drought event as consecutive years with negative indices, in which at least one year with the index
falling below the 10th percentile of its 850-year (1000 - 1849 AD) distribution. In such a way, we assure that the dry condition is maintained consistently during drought years, without being interrupted by one wet year or season. This method, which imposes a threshold based on the extreme percentiles, assures that strong negative anomalies persist throughout the entire year with droughts. Thus, we only include relatively severe droughts in the analysis.

Droughts with a duration of more than 3 years are considered as multi-year droughts. In the Sect. 3.3., we analyze the mean
condition during droughts in the control and transient simulations taking into account all short (1 and 2 years of duration) and long (more than 3 years of duration) Mediterranean droughts. For the next part of the analysis in the Sect. 3.4., we examine the dynamics associated with persistent multi-year droughts (more than 3 years of duration). These long droughts are separated into three stages: the initiation years as the first years of droughts, the termination years as the last years, and the rest as the transition years. The evolution of droughts is analyzed for each of the stages. This separation method is similar to the one used
by Parsons and Coats (2019).

Lastly, in order to define droughts with pan-west-central Mediterranean characteristic, we select only drought events where more than 70% of the region of study is occupied by negative indices (Fig. 1.(b)). Though, for multi-year droughts, we allow that this condition does not need to be fulfilled for the initiation and termination years but only for the transition years: a drought can start weak and with a more local characteristic, then expand to a larger proportion of the Mediterranean, and weaken again in the termination years.

## 2.4 Observation and proxy reconstruction datasets for the validation

To validate the model simulation, we compare mean seasonal and annual precipitation, time series of droughts and climate conditions associated with droughts among the observation, proxy reconstruction and model simulation for the period of 1901 – 2000 AD. We use the gridded station data for temperature and precipitation from U.Delaware v5.01 (Willmott and Matsuura, 2001), the sea surface temperature from the Extended Reconstructed Sea Surface Temperature v5 (ERSST v5; Huang et al. 2017), and geopotential heights from the 20th Century Reanalysis v2 (CR v2; Compo et al. 2011). The anomalies of variables are calculated by extracting the mean annual cycles with respect to 1950 – 1979 AD. The scPDSI is calculated using the U.Delaware v5.01 temperature and precipitation. The same calibration period 1950 - 1979 is used to calculate the scPDSI for the observation and model. Furthermore, we take the gridded tree-ring-based reconstruction of European summer scPDSI, OWDA (Cook et al., 2015). Among all drought indices, we use only scPDSI for comparison and validation, as the OWDA only provides this specific drought metric. The time series of scPDSI during this period are statistically compared to each other using the t-tests with the null hypothesis of equal means between two time series. The analysis of patterns associated with droughts is performed by calculating the spatial correlations between the scPDSI and SST, scPDSI and geopotential height fields.

## 3 Results

### 3.1 Validation of CESM: comparison among observation, proxy and model (1901 – 2000 AD)

In this section, we compare the mean precipitation, mean scPDSI, number and duration of droughts and atmospheric conditions associated with Mediterranean dry conditions among the observation, OWDA and CESM simulation for the period of 1901 – 2000 AD.

The model simulation exhibits similar spatial patterns of mean seasonal precipitation to those from the observation (Fig. 2), although some regions are statistically different. In the summer, both central and western Mediterranean present dry conditions, whereas in the winter, both regions are less dry than the summer with wet conditions over Portugal and Balkans. For the mean annual cycle, the model in general shows less precipitation than the observed values over both central and western Mediterranean. Nevertheless, the model reproduces well the annual precipitation cycle, depicting correctly the maximum and minimum periods of precipitation.

Comparing the time series of scPDSI (Fig. 3.(a) and (c)), the summer means between the model and OWDA are statistically similar to each other (p-value from the t-test of 0.28). The same happens between the OWDA and observation but with much lower confidence level of 1% (p value of 0.01). However, the means of both summer and annual scPDSI in the model are statistically different to those in the observation (p-values of 0.001). Nevertheless, all three scPDSI show negative trends during 1901-2000 AD (Fig. 3.(c)), also in each sub-period (1901 - 1950 and 1951 - 2000 AD), indicating a continuous increase of drying over the region, which is in line with the previous studies (Mariotti et al., 2008; Sousa et al., 2011; Spinoni et al., 2015). Based on the Mann-Kendall tests, these trends are all statistically significant at 95% confidence interval. For the number of droughts (Fig. 3.(b)), the observation presents 4, OWDA 7 and CESM 3 events during the last century. For the duration of droughts, the mean (5.43 years) and median (3 years) in OWDA are lower than those in the observation (11.50 and 12.50 years respectively). CESM also exhibits a lower median (6 years) compared to the observation, though, its mean (9.67 years) resembles better the observation than the one in OWDA. These differences in the means and medians between the observation and CESM are also present in annual scPDSI.

Overall, the model tends to underestimate the duration of present-day droughts than those from the observation. However, the model still shows to a certain extent its ability to reproduce persistent droughts of multi-year long duration. Additionally, the mean duration of droughts in CESM is longer than in OWDA. Still, it is important to note that the analysis on annual-scale extreme dry events, based on a relatively short present period of 100 years, is not sufficient to draw comprehensive conclusions on present-day multi-years droughts due to the limited number of events.

One reason for the difference in the scPDSI between the model and observation is potentially related to the model performance on meso-scale phenomena, which play an important role for the regional precipitation during the wet season (Alpert et al., 1990; Ulbrich et al., 2009; Champion et al., 2011; Watterson, 2006). Additionally, the model performance on internal variability may contribute to this discrepancy, which will be discussed in the following paragraphs. The difference in scPDSI between OWDA and the observation, which shows a p-value at the limit of 1% confidence level, can be explained by some characteristics of tree-ring based reconstructions. The annually resolved summer (JJA) OWDA is based on tree ring reconstructions, which are known to be biased towards the growing season. Thus, the annual signal is not fully preserved (Franke et al., 2013). Moreover, tree ring-based reconstructions for droughts tend to overestimate low-frequency variability compared to the instrumental observations (Franke et al., 2013), and the distribution of tree rings used to generate the gridded reconstruction over Mediterranean may not be enough to capture precipitation events associated with regional-scale cyclones and to fully depict dry/wet variability for the entire region (Babst et al., 2018).

After all, the model simulation and observation share many common patterns associated with the variability of scPDSI. Fig. 4 shows the correlation patterns between the scPDSI and SST, and the scPDSI and geopotential height at 850hPa. The observation and model exhibit significant positive correlations over the central Equatorial Pacific, though in the observation, the region with statistically significant correlation is located more on the north-central Pacific than in the model. The observation and model simulation share a common wave-like pattern from the North Pacific to Siberia over the extratropical latitudinal belt. Within this wave-like pattern, there is a prominent bipolar pattern with a significant negative correlation centered over the Mediterranean region and a positive correlation over the northern high latitudes. In the observation, the area of negative correlation of this

bipolar pattern is larger over Europe and the positive correlation is shifted to the Scandinavian region compared to the model simulation. Additionally, the observation and model present some common patterns occurring over the regions of ENSO, the Tropical equatorial Pacific, and NAO, the North Atlantic.

As these two modes of variability are important in the climate system, including the European climate, it is necessary to point out that the amplitude of ENSO is too strong in this version of CESM (Parsons et al., 2017; Stevenson et al., 2018). In the case of the NAO, the seasonal variability of this mode seems to be amplified, similar to many other CMIP models (Fasullo et al., 2020). These inherent biases related to the modes of variability can partially explain the differences we observe here between the model simulation and observation. Nevertheless, it is found that CESM is still able to capture relatively well the hydroclimate condition associated with the ENSO teleconnection (Stevenson et al., 2018). The model also resembles the present-day NAO pattern well including the spatial precipitation and temperature associated with this mode of variability in Europe (Deser et al., 2017).

Although there are some discrepancies between the model simulation and the observation, the model is able to reproduce the climate conditions associated with the variability of present-day scPDSI. In particular, the model is able to simulate multi-year droughts and these droughts have longer duration than those in OWDA. As it also shows statistical similarity to OWDA over the region, the model is suitable to analyze past Mediterranean droughts. The biases mentioned here are considered in detail in the Sect. 4 when discussing the results of this study.

### 3.2  Variability of Mediterranean droughts during the 850 - 1849 AD and their connection to the volcanic forcing

To gain an overview of drought conditions in the Mediterranean, we assess the indices defined in the Sect. 2.3 using the period 850 to 1849 AD by focusing on the drought events and their duration. We compare the variability and duration of droughts of the summertime scPDSI in CESM with those in OWDA. Note that a direct comparison between the proxies and the model simulation is not possible due to the different initial conditions and the chaotic behavior of the climate system (PAGES Hydro2k Consortium, 2017; Xoplaki et al., 2018). Thus, we focus on comparing the simulated summer drought variability with the one of OWDA. Then, we assess whether the simulated and reconstructed droughts respond similarly to the same external forcing, i.e. the volcanic eruptions.

Fig. 5.(a) and (c) exhibit the distributions and the 100-year running means of duration of summer droughts in OWDA and CESM. For the duration of summer droughts in 850 - 1849 AD (Fig. 5.(a)), the discrepancy between CESM and OWDA is clear, with OWDA presenting the mean duration of 5.38 years and CESM with 7.89 years. The distributions associated with these means are statistically different to each other. This seems to be consistent with the result in the previous section (Sect. 3.1) that shows that OWDA identifies droughts with shorter duration compared to the present-day observation and CESM in the Mediterranean region. Thus, this characteristic is still present during the entire last millennium. The variability of droughts over time in OWDA and CESM is also different to each other (Fig. 5.(c)). A period of increase in droughts, which is common in OWDA and CESM, is not identified. This gives us a first hint that the occurrence of droughts over the region are not mainly driven by the external natural forcings. Still, both time series present a common period of decrease in droughts around 1600 AD.

Similarly, Fig. 5.(b) and (d) show the distributions and the running means of duration of annual Mediterranean droughts in CESM using different indices. As expected, different indices do not exactly behave similarly in terms of the occurrence, number of events, and duration (Raible et al., 2017; Mukherjee et al., 2018). However, in the 850 - 1849 AD period, the indices coincide for some years: in 89 years, all indices indicate the same overlapped drought periods. In terms of duration in 850 - 1849 AD, the scPDSI is the one which shows more longer lasting droughts than other indices, with a mean duration of 9.1

years. Then, the SPEI, SOIL and SPI follow it with the mean durations of 2.9, 2.8, and 2.3 years, respectively. The SPI presents more events than other indices, but with shorter duration. All these means are statistically different among each other, except the means between SPEI and SOIL, which are statistically similar (p-value of 0.87). This difference in duration of droughts among the indices is also evident in the control simulation.

The difference in duration of droughts among indices can be explained by the water balance variables involved in the

computation of each index. For instance, the SPI only takes precipitation as its input variable. Thus, it does not consider the atmospheric evaporative demands, which can be intensified during dry periods. Therefore, we expect that the SPI shows a reduced duration of droughts compared to the scPDSI and SPEI, which include the potential evapotranspiration in their water balance. The same holds true for the SOIL index. Though, the soil moisture in the model is closely connected to the hydrological cycle reflecting the balance between the precipitation and actual evapotranspiration, the magnitude of actual evapotranspiration

over the region is smaller than the potential evapotranspiration derived from the Thornthwaite method. Hence, the water balance involved in SOIL is affected in such a way that the drought duration is reduced compared to the scPDSI and SPEI. Lastly, droughts with relatively longer duration in the scPDSI is explained by the memory effect embedded in the calculation scheme of scPDSI (Palmer, 1965; Wells et al., 2004), which other indices, that are obtained by being normalized with respect to certain statistical distribution families, do not include. The scPDSI is an accumulating index; therefore, during the calculation process,

the weighted value of preceding months is used to estimate the index for the current month, implying a persistence of the events. Hence, with the scPDSI, an intense yearly drought would likely induce a drought in the following year and this effect can be exacerbated in the context of intense multi-year droughts.

Importantly, for the same indices, the distributions of duration of droughts are statistically similar to the distributions in the control simulation at 99% confidence interval (Fig. 5.(a) and (b), and table A1). This implies that the variability of droughts in

the transient simulation is within the range given by the internal variability in the control simulation.

In terms of the timing of the occurrence of droughts over the period of 850 - 1849 AD (Fig. 5.(d)), coherent changes among all indices are not identified, which is expected due to the different input variables and calculation schemes among drought indices. This fact also indicates that each index responds differently to the changes in precipitation and temperature, hence potential evapotranspiration, caused by the externally forced variability, e.g., the volcanic forcing.

To further assess a potential connection between the drought indices and volcanic eruptions, a wavelet coherence analysis is applied (Fig. 6). The analysis shows that significant co-variability between the simulated drought indices (SOIL and scPDSI) and eruptions are found during periods with strong and frequent volcanic eruptions, for instance, around 1257 Samala and 1600 Huaynaputina eruptions. For small eruptions, the signals of co-variability are not uniform among the eruptions, some showing significant co-variability while other not. In addition, the phase relationships between the eruptions and the drought indices

also vary among the eruptions (not shown). This non-uniformity of co-variability between the small eruptions and the drought indices is a strong indication that no physical connection between both exists, i.e., the significant co-variability is merely due to statistical artefacts. OWDA does not show a strong significant co-variability during the 1257 Samala eruption, which was the strongest eruption in the last millennium (Gao et al., 2008). Still, OWDA shows a significant co-variability during the period of Little Ice Age around 1400 - 1600 AD, similarly to CESM.

The wavelet coherence analysis is clearly useful to distinguish the effects of strong and small eruptions on the variability of drought indices. However, the analysis poses some problems in handling discontinuous time series with a sporadic occurrence of the events, such as the volcanic eruptions. Filtering certain frequency bands from this kind of discontinuous time series smears out the eruption (i.e., an eruption starts earlier and last longer than in reality), therefore, adding some non-physical artefacts in the time series. Some of these are also reflected in Fig. 6 showing that the significant co-variability occur earlier

that the actual eruption years.

Hence, for a more detailed analysis on causal effects of volcanic eruptions on drought variability, the superposed epoch analysis is applied to the 16 largest eruptions and the 16 smallest eruptions (Fig. 7; for the list of the eruption years, see table A2). The analysis shows that the increases in drought indices are followed after large eruptions, and this positive association lasts up to 3 years in CESM. On the other hand, no significant response of drought indices to small eruptions is noted. This

finding is in line with Rao et al. (2017) and McConnell et al. (2020) that demonstrated wetter conditions in the Mediterranean region after strong eruptions. Thus, the analysis shows that large eruptions are associated with an increase in drought indices, i.e., wet periods. In other words, the occurrence of yearly and multi-year Mediterranean droughts are not driven by the volcanic eruptions, but the internal variability.

In the next sections, we investigate the underlying dynamics using only the SOIL as the drought indicator in order to

understand the role of the internal variability in Mediterranean droughts. We use SOIL as it reflects the regional hydrological balance associated with the precipitation and evapotranspiration. Another advantage of this index is that the variable is a direct output from the model, thus, it does not require any further step, except for calculating the anomalies, and statistical assumptions as other indices do. In addition, the SOIL overlaps full or a part of drought periods given by the other three indices, without significantly underestimating the multi-year duration of droughts. The droughts in SOIL overlap the 36%, 25% and 29% of

droughts in the scPDSI, SPEI and SPI, respectively. Also, the SOIL and each of the indices are statistically correlated at 1% confidence level for the entire period of 850 -1849 AD with Pearson correlation coefficients of 0.81 (thus, 66% of variance) with scPDSI, 0.78 (0.61%) with SPEI and 0.86 (74%) with SPI. Hence, the results in the following sections can be partially transferred to the other indices. To guarantee the transferability, the analysis in the next sections was repeated with each of the drought indices, showing similar results as for SOIL (therefore figures not shown).

**3.3 Atmospheric circulation associated with Mediterranean droughts (850 - 1849 AD)**

In this section, the atmospheric circulation associated with Mediterranean droughts is investigated by using the SOIL in the control and transient simulations up to 1849 AD. The control simulation presents 7.25 droughts per century, with the mean duration of 3.06 years and the transient simulation shows 8 droughts per century with the mean duration of 2.81 years.

To get a first glance of the atmospheric circulation during drought conditions, we analyze the mean circulation conditions during all short (1 and 2 years of duration) and long (more than 3 years) Mediterranean droughts together. Fig. 8 shows the anomalies of geopotential height at 850 hPa and surface temperature during Mediterranean droughts for each simulation. The structures of geopotential height and temperature anomalies during Mediterranean droughts are similar, with a high-pressure system centered over central Europe accompanied by a positive temperature anomaly. This high-pressure anomaly, which from now on is called the drought high, is found in all heights from the 850 to 300 hPa (figures not shown), indicating a barotropic nature of this atmospheric circulation system. Additionally, a low pressure anomaly is situated over the area of Scandinavia to Russia. Thus, the atmospheric circulation shows a north easterly shift of the westerlies over Europe, so that moist air masses from the North Atlantic are passed around the Mediterranean.

Outside the European continent, a negative temperature anomaly over the Tropical Equatorial Pacific and a positive anomaly over the North Pacific are prominent in both simulations. These temperature patterns resemble the cold phase of ENSO and the positive phase of Pacific Decadal Oscillation (PDO), respectively. Besides, a positive geopotential height anomaly at the mid-latitudes and a negative anomaly at the high latitudes over the North Atlantic region is another pattern that both simulations share in common during droughts. This pattern is similar to the positive phase of the NAO; however, the southerly center of action is shifted to central Europe, which also resembles partially the East Atlantic Pattern (EA). The distributions of these common patterns are also statistically similar between both simulations, indicating that they are derived from the same statistical population. Thus, the simulations share common mechanisms associated with droughts, mainly driven by the internal variability of the climate system in the model. This reaffirms the result in the previous section.

Some statistically significant dissimilarities between the control and transient simulations are noticeable mostly in the temperature anomalies. Over the regions where the temperature anomalies are statistically different, the anomalies are rather weak, except the warming in Siberia. However, this positive temperature anomaly in Siberia is not associated with a geopotential height pattern, showing statistically indistinguishable geopotential height anomalies in the region between the control and transient simulations (Fig. 8). This indicates that there is no change in the circulation pattern over this region that can possibly be connected to the Mediterranean drought condition.

The drought high is a clear feature that appears during all droughts over the region. This pattern over central Europe and the western Mediterranean is similar to the pattern of the first mode of canonical correlation described by Xoplaki et al. (2003). In Xoplaki et al. (2003), this pattern is associated with the variability of temperature during the summertime in the Mediterranean region. In this study, the high-pressure system is present during all seasons of years with droughts, showing a relatively stronger intensity in winter and spring than in summer (Fig. 9). This is expected, as the variability of the geopotential height fields over Europe and the North Atlantic is reduced in summer compared to the other seasons, because the meridional temperature gradient on the Northern Hemisphere is also reduced. Therefore, the main forcing of the atmospheric circulation is weakened. The findings show that the atmospheric conditions in wet seasons, i.e. winter and spring, are determinant at controlling the annual mean hydroclimate, thus, indicating the importance of precipitation and dynamics during the wet season in annual-scale Mediterranean droughts (Lionello et al., 2006; Xoplaki et al., 2004; Zveryaev, 2004).

As expected, a decrease in precipitation occurs in all seasons during droughts: the winter and spring precipitation decreases by 13%, and the summer and autumn precipitation by 11% compared to non-drought periods. These changes in precipitation are in the range of the rates of the expected decrease in annual precipitation over the Mediterranean region in the future scenario. In the future, the regional precipitation is expected to reduce by 5 – 30% from its present-day value and this will cause water shortage related issues in the region (Dubrovský et al., 2014; Mariotti et al., 2008). The temperature shows a positive anomaly over the region in all seasons, with strongest signals during summer and autumn, a finding, which is in line with, e.g., Xoplaki et al. (2003). This finding indicates that Mediterranean droughts are more associated with anomalously warmer atmospheric conditions.

## 3.4   Dynamics of multi-year droughts

For the analysis on multi-years Mediterranean droughts, we focus on droughts with a minimum duration of 3 years. As shown in the previous section in Fig. 8, the drought high is a prominent atmospheric circulation pattern during Mediterranean droughts. The pattern resembles the positive NAO-like pattern, although with a shift to the North-East. At the same time, a colder than normal condition over the Tropical Equatorial Pacific is detected, which is similar to a La Niña-like condition. Here, we investigate the origin and the evolution of Mediterranean long droughts associated to NAO, ENSO-like condition and drought high using the transient simulation up to 1849 AD.

The phases of NAO and ENSO are defined with respect to the non-drought period: the values below 25th (above 75th) percentile of NAO and ENSO during the non-droughts period are considered as negative (positive) phases of NAO and ENSO, respectively (Fig. 10). The extreme NAO and ENSO are also defined in a similar way by taking the values below 5th percentile for negative extremes and above 5th percentile for positive extremes. Defining thresholds relative to the non-drought period facilitates the comparison between the two periods, showing whether and how these modes of variability during droughts differ from those during the opposite hydroclimate conditions. For simplicity, we call these relative negative and positive phases simply as positive and negative NAO and ENSO without referring constantly that they are defined relative to the non-drought period.

Considering all short and long droughts, the simulation shows that droughts occur more frequently during the positive phase of annual and winter NAO than during the non-drought period: 38% of the drought years show the positive phase of annual NAO and 32% for the winter, while the negative annual NAO occupies 16%, and 23% in the winter. However, the positive NAO condition does not persist throughout the entire years of droughts. Rather, it fluctuates from the positive to negative phases during multi-year droughts. For the co-occurrence between droughts and phases of ENSO, we find that La Niña-like conditions are present in 37% of total drought periods, El Niño-like in 22%. Similar to NAO, La Niña-like conditions do not persist throughout the entire drought years.

The extreme positive NAO occurs slightly more frequently during droughts. In addition, the distributions of extreme positive NAO during droughts is statistically different from the distribution during the non-drought period at 5% confidence level for both annual and winter NAO (p-values from MW-U test of 0.02 for annual and 0.002 for winter). However, for ENSO, there is no statistically significant change in the distribution and frequency of extreme La Niña-like condition during droughts.

To examine the behavior of the atmosphere during multi-years Mediterranean droughts, the frequency of modes of variability and mean composites of circulation during droughts are split in three stages: initiation, transition and termination years (Sect. 2.3). The frequencies of occurrence of NAO and ENSO in each stage is presented in Fig.10. The positive NAO occurs more frequently in the initiation years than in the transition and the termination years of droughts. The positive NAO occupies 49% of the initiation years.This shows that the occurrence of positive NAO almost doubles in the initiation years of droughts compared to the non-drought period (25% of the occurrence of positive NAO). In the transition years, positive NAO decreases and finally fall to 29% in the termination years. The frequency of extreme positive NAO also decreases over time. In the case of ENSO, the frequency of La Niña-like is 40% in the initiation years relative to the non-drought period. The occurrence of La Niña-like conditions increases slightly in the transition years, though, this increase is not statistically significant with respect to the previous stage. Then, it decreases to 20% in the termination years. After the initiation years, there are increases in negative NAO and El Niño-like states. These changes in the frequencies of NAO and ENSO in each stage of droughts indicate a weakening role of large-scale circulation patterns at sustaining the persistence of droughts over time. As the intensities of droughts become more severe with their duration, some other factors need to be involved in sustaining the longevity of multi-years Mediterranean droughts from the transition to termination years.

The mean circulation and atmospheric conditions during the development of multi-year Mediterranean droughts are shown in the Fig. 11 depicted by the specific humidity, temperature, and winds at 925 hPa level. In the initiation years, the southerly winds prevail over the southern Mediterranean region. These southerlies block the intrusion of the westerly systems from the North Atlantic, and together with the cyclonic winds in the East Atlantic distribute dry and warm air masses from the East Atlantic, southern Mediterranean and west Africa to the continent. In the transition years, a complete anticyclonic circulation associated with the high over central Europe and the Mediterranean region is developed. This anticyclonic system distributes the dry and warm air to north and west of the continent and sustains dry and warm condition over the region. During these years, the westerlies from the North Atlantic are clearly weakened. In the termination years, the anticyclonic circulation over Europe is not observed anymore, indicating a break-up of the high.

The mean circulation in each stage shows that the development of the high pressure system, namely the drought high in the Fig. 8, takes place in the transition years. This indicates that some mechanisms associated with this circulation are possibly important at determining the longevity of droughts after the initiation years. A possible candidate of an important process for the transition stage of Mediterranean droughts is the interaction among regional atmospheric and soil variables initiated by the anticyclonic circulation system over the region.

The presence of the atmosphere - soil interaction during the transition years is supported by the increases in frequencies of positive surface temperature (TS) and sensible heat flux (SH), and negative soil moisture (SOIL), evapotranspiration (EV), and latent heat flux (LH) anomalies during this period (Fig.12). The mechanism associated with these regional atmospheric and soil variables is explained as follows: a decrease in precipitation supported by the positive NAO and/or La Niña-like induce initial regional dryness and a stable atmospheric condition associated with the increase in geopotential high anomaly (GP) over the region. A positive GP induces an initial increase in the regional TS. This positive TS decreases SOIL and EV. The latter increases SH and decreases LH during the initiation year. During the transition years, the positive TS is even magnified due to

the stable atmospheric condition that still persists (positive GP) and the increase in SH and loss of LH in the previous stage. TS in turn again increases SH, decreases EV and LH, thus, decreases SOIL. Over time, the complete high is developed and persists, fuelling again this positive temperature - soil moisture feedback (Seneviratne et al., 2010; Yin et al., 2014). Thus, during the transition years, the means and occurrence of these variables associated with the feedback (positive TS, negative SOIL, negative EV, positive SH, negative LH) are clearly larger than their values during the initiation years (Fig.12). This mechanism continues until the termination years. During the termination years, the positive GP and TS still prevail over the region, though, with reduced anomalies. Also, the magnitudes of other variables are reduced compared to the previous stage.

This result indicates that once the drought high is developed, the temperature - soil moisture feedback is a more important mechanism than the connection to NAO and La Niña-like patterns in order to sustain the continuous depletion of soil moisture. This means that, although the large scale circulation patterns help to support the regional dry conditions, after the initiation years, their roles in sustaining droughts are diminished.

### 3.5 Historical and Future condition on droughts: 1850 to 2099 AD

Here, the behavior of Mediterranean droughts as well as the associated mechanisms for the period 1850 - 2099 AD are presented. For an overview, the time series of the soil and precipitation anomalies (with respect to 1000 -1849 AD) over the Mediterranean region for the period 1850 - 2099 AD are shown in the Fig. 13. The simulation indicates that the region becomes drier in this period than in the past, showing pronounced decreases in soil moisture and precipitation. The reduction in these two variables is already apparent from the beginning of 1850 AD concomitantly with the anthropogenic increase in GHG. By the end of the 21st century, the region experiences a continuous drought without any wet anomaly with respect to the 1000 - 1849 AD condition. This indicates a shift of the mean climate of the region to a drier climate, and this transition is initiated since the pre-industrial period and intensified under the RCP 8.5 scenario.

We examine whether the mechanisms associated with Mediterranean droughts described in the previous section are affected by the anthropogenic influences on climate and whether these changes contribute to the intensification of droughts and eventual aridification in the region occurring in this period. For this, the detrending method is applied to the simulation following the steps mentioned in the Sect.2.2. First, we analyze the non-detrended drought related variables with the anthropogenic influences on them; then, the detrended variables to see the background climate during droughts excluding the linear trends.

As for the period 850-1849 AD, droughts during the period 1850 - 2099 AD are associated with the intense positive geopotential height and temperature anomalies over central Europe and the Mediterranean. However, these features show more intense amplitudes in geopotential height (GP) and temperature (TS) anomalies than during the period 850 - 1849 AD (Fig. 14.(a)). The variances of GP, TS and SOIL are also enlarged compared to the past; therefore, their medians and extreme tails are also magnified, which imply that the dryness and its associated atmospheric conditions become more frequent and severe in 1850 - 2099 AD. The increases in GP and TS clearly intensify the above mentioned interaction among regional atmospheric and soil variables, i.e the positive temperature – soil moisture feedback. This intensification aids the longevity and intensity of droughts, which is reflected by a reduction in the surface soil moisture anomaly (Fig. 13). Additionally, the precipitation - soil

moisture feedback is also involved: a continuous reduction in precipitation decreases the available soil moisture, thus, inducing less evapotranspiration, which leads again to a reduction in precipitation (Seneviratne et al., 2010).

Related to the modes of variability, the frequencies of positive NAO and La Niña-like conditions during droughts also seem to be affected by the overall change in global temperature (Fig. 14.(b)). Compared to 850 – 1849 AD, the non-detrended 1850 – 2099 AD period shows reduced frequencies of both positive NAO and La Niña-like conditions during droughts. This result
is in line with the previously mentioned intensification of land - atmosphere feedbacks: in this situation where the GP and TS become intense, the regional atmospheric variables play a more dominant role on Mediterranean droughts and the importance of modes of variability is reduced, even during the initial stages of droughts. Hence, the role of positive NAO and La Niña-like in different stages of droughts is diminished. Note however, that model biases in representing large scale modes of variability, in particular ENSO, might be relevant. Many CMIP5 models have problems in realistically reproducing the cold SST in the
east Tropical Pacific. Therefore, these models would show less La Niña events in the future warmer conditions (Seager et al., 2019). An overall increase in El Niño-like condition here (Fig. 14.(b)) can also be partially related to this bias. Nevertheless, considering this bias does not affect the result that the interaction among the regional variables is changed by the increase in temperature, causing more intensified regional land-atmosphere feedbacks during this period.

For the detrended variables, the GP anomalies and SOIL during droughts of the period 1850-2099 AD are statistically similar to the ones in the period 850-1849 AD (p-value of 0.09 for GP and 0.44 for SOIL). The same is also true for the NAO and
540 ENSO during droughts (p-values of 0.19 and 0.29 for each). The detrended TS over the region is statistically similar from the 850-1849 AD value but only at 2% confidence level (p-value of 0.02). This indicates that the detrending method is not able to fully remove the strong effects of anthropogenic changes on temperature in future droughts. Nevertheless, the mean spatial composites of the detrended surface temperature and geopotential height at 850 hPa during Mediterranean droughts in
the period 1850 - 2099 (Fig. 15) statistically agree with the ones in 850 - 1849 AD period (Fig. 7) over a large portion of the Mediterranean region.

Hence, when the anthropogenic effect is removed (i.e., the variables are detrended), the mechanisms involved in droughts remain unchanged during 1850 - 2099 AD. The comparison of non-detrended with the detrended variables, thus, indicates that no other factor than the anthropogenic influence in temperature is the cause of the severe dryness in this period. In the future
scenario, the intensities of both land - atmosphere feedbacks are magnified due to the increases in GP and TS caused by the increases in GHG, and these feedbacks become the dominant one at controlling the desiccation over the region.

## 4 Conclusions

We have investigated the variability and mechanisms of multi-years droughts over the western and central Mediterranean region with pan-regional characteristic for the period of 850 - 1849 AD and whether these mechanisms associated with Mediterranean
droughts have changed after the pre-industrial period from 1850 to 2099 AD with the anthropogenic increase in GHG. For this, we used simulations from CESM (Lehner et al., 2015).

Firstly, the quantification of droughts is sensitive to the choice of drought index even in the paleoclimate context. For example, the scPDSI exhibits drought events with longer duration than other indices, such as SPEI, SPI and soil moisture anomalies. Although, the major mechanisms that induce multi-years droughts over the region remain similar due to the same overlapping periods and statistically significant correlations among indices, this discrepancy among indices can lead to different conclusions, mostly in the number and duration of past drought events. This shows that using just one unique index is still complicated, even in the paleoclimate context. Hence, the uncertainty associated with different indices must be taken into account when comparing indices in drought studies (Dai, 2011; Raible et al., 2017; Mukherjee et al., 2018), in particular, in the case when only a single drought index is used and the focus is on the assessment of the duration of extreme hydrological events in past periods.

Secondly, we found that the past Mediterranean droughts are mainly induced by the internal dynamics of climate system supporting the finding of Xoplaki et al. (2018): the duration of droughts and the patterns of surface temperature and circulation over the Mediterranean during droughts in the control simulation is statistically similar to those in the transient simulation with external forcing. Moreover, a causal connection between volcanic eruptions and Mediterranean dry conditions is not identified. However, our result indicates a connection between large volcanic eruptions and wet periods over the region supporting findings of previous studies (McConnell et al., 2020; Rao et al., 2017). A distinct atmospheric pattern occurring during Mediterranean droughts is a barotropic high pressure system accompanied by a positive temperature anomaly over central Europe and the Mediterranean region. This warm high persists during all seasons when droughts occur in the region, showing stronger intensity during winter and spring. This result emphasizes the importance of the wet cold seasons, i.e. winter and spring climate and circulation in annual Mediterranean droughts. Additionally, the positive NAO and La Niña are other patterns that occur more frequently during Mediterranean droughts.

Thirdly, the mechanisms associated with sustaining multi-years droughts change through the stages of droughts. We found that the large-scale circulation patterns, such as the positive NAO and negative ENSO, play a more important role during the early stage of droughts, by providing dry condition over the western and central Mediterranean region to initiate such events. Then, the longevity of droughts is determined by the interaction of regional circulation variables, which involve stable atmospheric conditions, an increase in temperature, changes in evapotranspiration and surface heat fluxes. Namely, this is the temperature - soil moisture feedback, which continues until the termination of droughts. During these transition years of droughts, the role played by the large-scale patterns is reduced. Hence, the persistence and duration of multi-years droughts should not be fully attributed to the states of large-scale circulation patterns, such as NAO and ENSO, as the roles of regional feedback and circulations on droughts become as important as or more important than large-scale modes of variability after the initial development of droughts.

It is important to note that the model inherent biases in representing ENSO and NAO (Bellenger et al., 2014; Fasullo et al., 2020) can have some implications in our results on the frequencies of ENSO and NAO at different stages of droughts. The model may produce too frequent and strong La Niña conditions and positive NAO during droughts due to its amplified variability (decadal for ENSO and seasonal for NAO). Moreover, due to the uncertainty associated with the changes in these modes in the future scenario, caution is required when interpreting the connection between droughts and modes of variability

in the future warming scenario. Many CMIP5 models show an overall warming of the Tropical Equatorial Pacific reducing the west-east gradient of SST which is different from what is observed in the present period (Seager et al., 2019). This model bias to the observation implicates reduced La Niña conditions in CMIP5 models in a warmer world. Nonetheless, this problem does not affect our conclusion: the roles of ENSO and NAO become weaker with the longevity of droughts, while the regional circulation and feedback become more dominant at maintaining the persistence of droughts, which is also found in the future warming scenario.

Fourthly, the decreases in soil moisture and precipitation anomalies are already detected since the pre-industrial period concomitantly with the anthropogenic increase in GHG. This means that the intensification of droughts and the shift of the mean climate over the region to a drier climate have already started since the pre-industrial era. This regional desiccation is principally caused by the anthropogenic increase in GHG, which induces the intensification of interactions between the regional atmospheric and soil variables, associated with the temperature - soil moisture and precipitation - soil moisture feedbacks. If the increase in temperature and decrease in precipitation continue, the region will suffer from a continuous aridification instead of droughts, as droughts are the deviation from the mean hydrological condition.

Fifthly, it is important to mention that our analysis is based on a single model output and this raises questions related to single model studies, such as boundary condition problems and model-dependent biases and physics (PAGES Hydro2k Consortium, 2017). Nevertheless, for a small confined area that surrounds a large body of water mass, the Mediterranean Sea, and where the land coverage is limited, a finer horizontal resolution is needed to represent more realistically the regional climate and the meso-scale processes which are involved. In the end, our study provides a useful understanding on the long-term variability and mechanisms of Mediterranean droughts by analyzing the entire last millennium. We addressed the influences of external and internal variability on Mediterranean droughts and different roles of the large-scale modes of variability and regional circulation during the different stages of multi-year droughts.

Lastly, we emphasize again the importance of assessing different drought indices in the paleoclimate context, and also in the present and future warming scenario. The reason is that most of the commonly used offline drought indices, such as scPDSI, are based on a water balance that considers only the atmospheric moisture supply and demand, and these indices tend to overestimate drought risks in the future warming scenario (Berg et al., 2017; Mukherjee et al., 2018; Swann et al., 2016). Moreover, Berg et al. (2017) found that the upper-level soil moisture indicates droughts, whereas the mean 3-meter soil moisture shows wet or relatively weak dry conditions compared to the surface level. In our study, we used the upper 10 cm soil moisture anomaly that partially reflects the water stress on plants. However, the upper 10 cm of soil level is not enough to fully assess the complex atmosphere-soil-vegetation interaction and the variability in the deeper levels of the soil. In addition, the upper 10 cm soil moisture used here also magnifies drought risks to some extent same as other offline drought indices. Nevertheless, the Mediterranean is one of the regions where still the depletion of soil moisture occurs both at the surface and in the mean 3-meter soil level, though the amplitude of the rate of decrease is reduced in the 3-meter soil moisture compared to the rate in the surface soil moisture (Berg et al., 2017).

As a next step, more studies on drought metrics need to be conducted in order to assess properly the future drought risk in the region. In addition, as vegetation is known to have more complex responses to the changing climate and droughts

(Swann, 2018; Swann et al., 2016), the role of vegetation in extreme hydrological events shall be investigated to get a more comprehensive view on drought mechanisms and their changes in the future. The Mediterranean region is considered as one of the most vulnerable regions under the future warming scenarios (e.g. Giorgi and Lionello, 2008; Lehner et al., 2017) and human impacts can modify the natural mechanisms and propagation of droughts increasing droughts risks and water shortage issues over the region (Van Loon et al., 2016). Hence, more studies on the topics related to droughts and permanent future aridification in the Mediterranean region including the role of vegetation during this period are necessary to develop a better preparedness for upcoming changes.

*Code availability.* Two R packages were used to calculate the drought indices: the *scPDSI* (Zhong et al., 2018) and the *SPEI* (Vicente-Serrano et al., 2009). The *biwavelet* (Gouhier et al., 2018) was used for the wavelength coherence analysis and *burnr*(Malevich, 2018) for the superposed epoch analysis.

*Data availability.* The summer scPDSI from the Old World Drought Atlas (Cook et al., 2015) and the Sea Surface Temperature ERSST v5 (Huang et al., 2017) are available on https://www.ncdc.noaa.gov/ . The 20th Century Reanalysis V2 (Compo et al., 2011) and UDel_AirT_Precip data (Willmott and Matsuura, 2001) are provided by the NOAA/OAR/ESRL PSL, Boulder, Colorado, USA, from their Web site at https://psl.noaa.gov/. The CESM simulations (Lehner et al., 2015) are available on request at the University of Bern.

*Author contributions.* WK and CR discussed and set up the initial research idea. WK performed the analysis and drafted the manuscript under the supervision of CR. CR provided critical feedback on the results and the manuscript. Both authors contributed to the interpretation and discussion of the results and edited the manuscript together.

*Competing interests.* The authors declare that they have no conflict of interest.

*Acknowledgements.* The study is funded by the Swiss National Science Foundation (SNSF, grant 200020_172745). We acknowledge that some simulations were performed at the Swiss National Super Computing Centre (CSCS). Support for the Twentieth Century Reanalysis Project dataset is provided by the U.S. Department of Energy, Office of Science Innovative and Novel Computational Impact on Theory and Experiment (DOE INCITE) program, and Office of Biological and Environmental Research (BER), by the National Oceanic and Atmospheric Administration Climate Program Office, and by the National Oceanic and Atmospheric Administration Climate Program Office, and by the NOAA Physical Sciences Laboratory. The 20th Century Reanalysis V2 and UDel_AirT_Precip data are provided by the NOAA/OAR/ESRL PSL, Boulder, Colorado, USA, from their Web site at https://psl.noaa.gov/. The authors greatly thank the editor Hugues Goose and the anonymous reviewers for their constructive feedbacks and insightful comments. The comments from the reviewers helped us to improve the analysis and the presentation of the results.

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

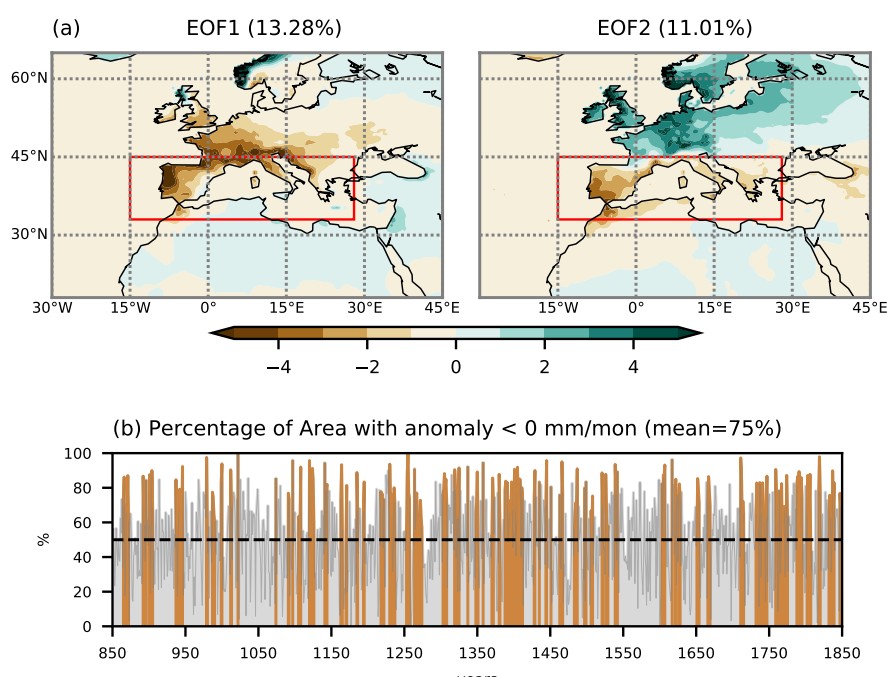

**Figure 1.** (a) Variance explained by the first EOF and second EOF in the observed monthly precipitation from U.Delaware v5.01 dataset for the period of 1901 – 2000 AD. The red rectangle indicates the region of study: west- and central Mediterranean. (b) Percentage of area with the soil moisture anomaly below 0 mm.mon$^{-1}$ in the region of study during the last millennium in CESM. Shaded in brown indicates the years with droughts.

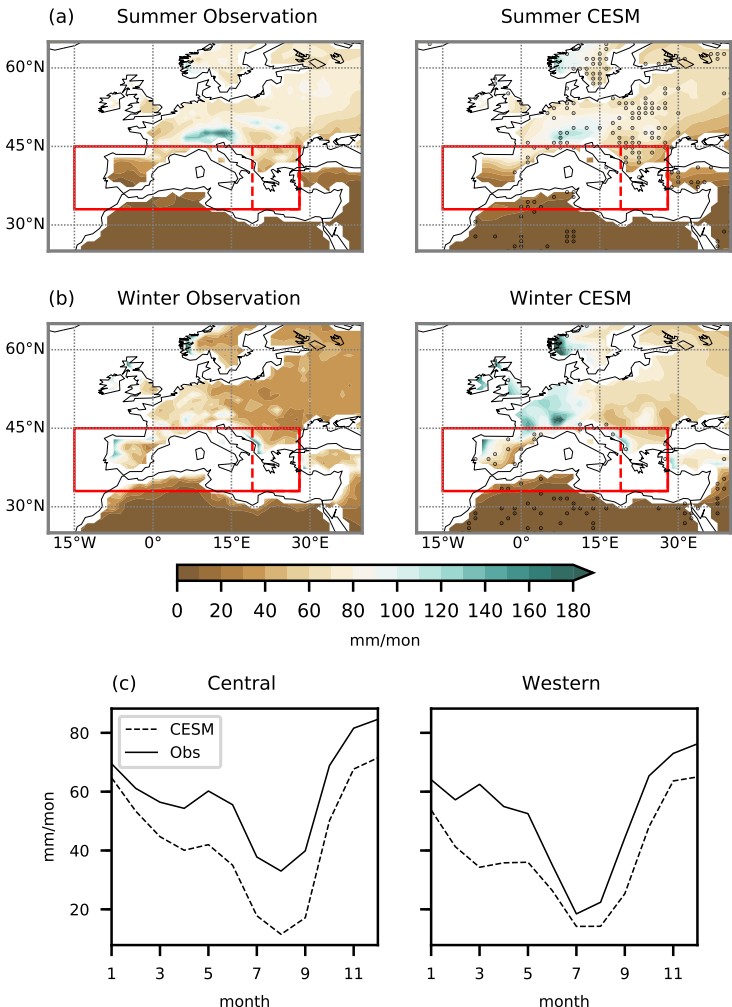

**Figure 2.** Mean seasonal precipitation for the observation (left) and CESM (right) in the (a) summer and (b) winter for the period of 1901 – 2000 AD. Black dots on the composites of CESM indicate the regions where the means between the observation and model are not statistically similar at 5% confidence level from the t-tests. (c) Mean annual cycles of precipitation for the same period over the areas in the rectangles. The observation is in continuous lines and CESM in dashed lines.

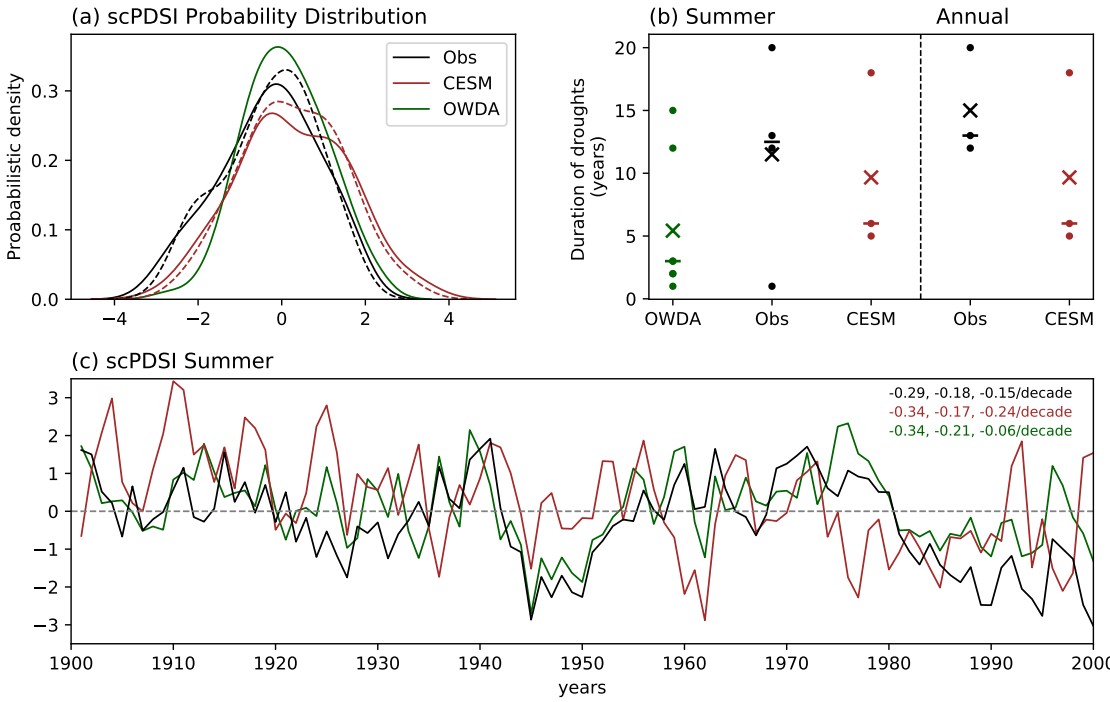

**Figure 3.** (a) Probabilistic distribution of the yearly self-calibrated Palmer Drought Severity Indices (scPDSI) from the U.Delaware v5.01 observation (black), CESM (red) and OWDA (green) smoothed by kernel density estimates using Gaussian kernels. Continuous lines indicate the summer and dashed lines the annual scPDSIs. The t-tests are applied among these distribution under the null hypothesis of equal means between two time series.The p-values between the summer scPDSI from CESM and OWDA is 0.28, between the OWDA and observation is 0.01, between the CESM and observation are 0.001 both for summer and annual values. (b) Distribution of duration of annual droughts in different datasets. Crosses are the means and horizontal dashes are the medians of duration of droughts. (c) Time series of summer scPDSIs from the observation (black), CESM (red) and OWDA (green) for 1901 - 2000 AD. The numbers on the upper-right indicate the values of the trends per decade, from left to right for the period of: 1901 - 1950 AD, 1951 - 2000 AD, and 1901 - 2000 AD. All these trends are statistically significant at 95% confidence interval based on the Mann-Kendall tests.

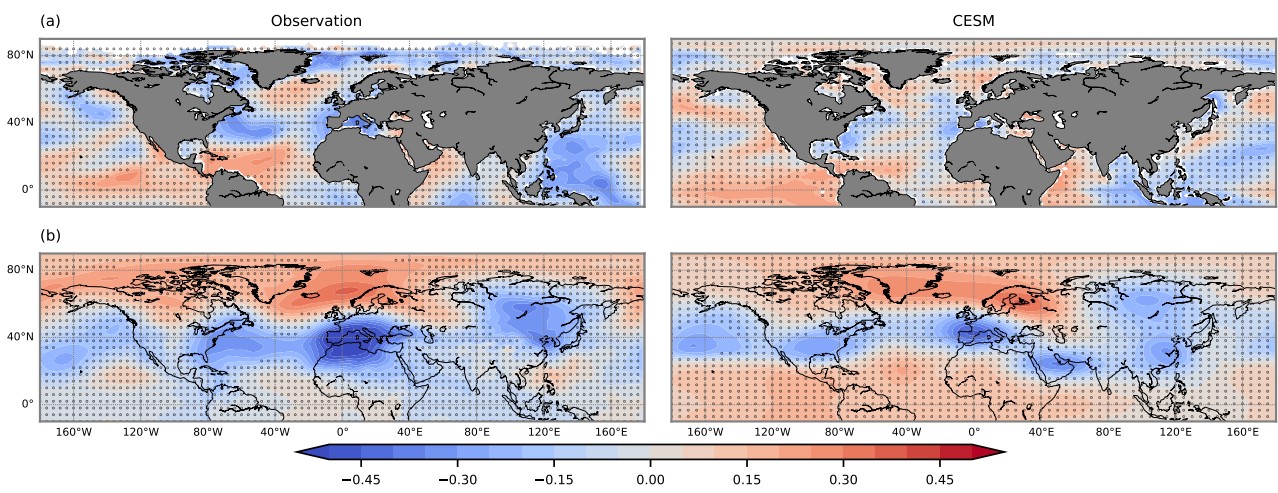

**Figure 4.** Pearson correlation coefficients between the scPDSI and anomalies of (a) sea surface temperature from ERSST v5 and (b) geopotential height at 850 hPa from the CR20 for the observation (left) and CESM (right) during the period of 1901-2000 AD. The linear trends of variables are removed before applying the correlation. Black dots on the maps show the regions where correlations are statistically not significant at 5% confidence level.

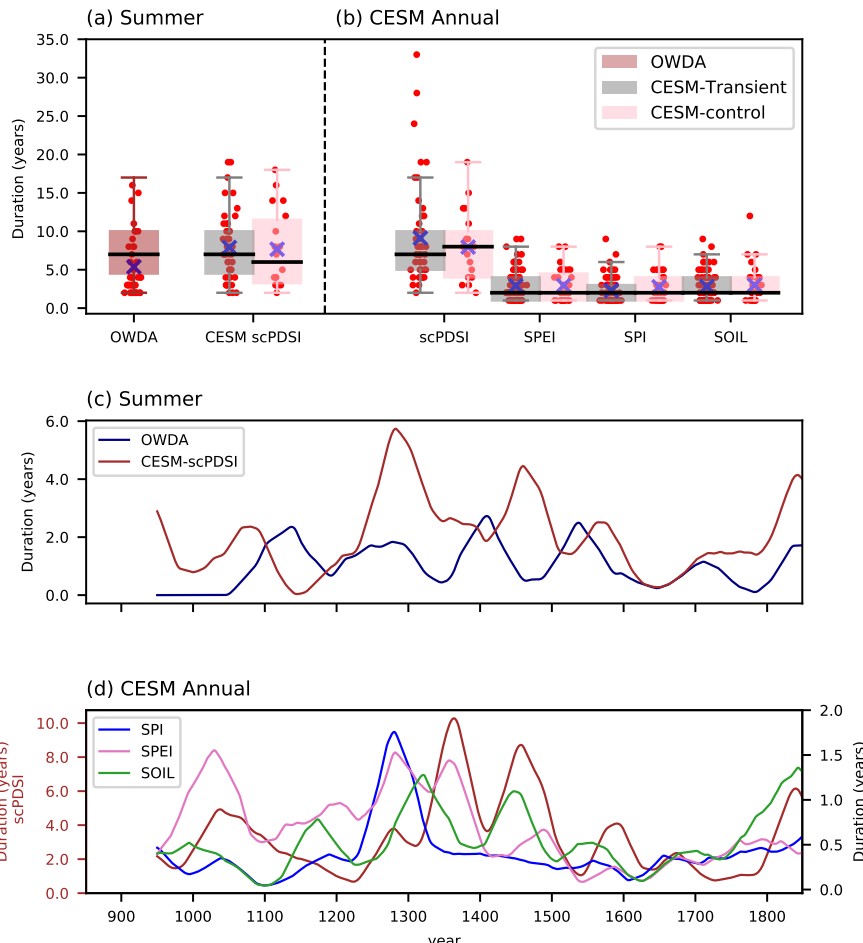

**Figure 5.** (a) Distribution of duration of droughts for the summer scPDSIs from OWDA, the transient and control simulations, and for (b) the annual drought indices from the transient and control simulations in CESM. Red points indicate individual drought events, black lines on the boxes are the medians and blue crosses are the means of duration. (c) 100-year running means of the duration of droughts for the summer scPDSIs and (d) the annual drought indices. The indices are the self-calibrated Palmer Drought Index (scPDSI) from OWDA, summer scPDSI (CESM-scPDSI), annual Standardized Precipitation Index (SPI), Standardized Precipitation Evapotranspiration Index (SPEI), soil moisture anomaly (SOIL), and annual scPDSI from CESM. Note that the annual scPDSI (brown line in (d)) has a separate y-axis for its duration. The Mann-Whitney U-tests (M-W tests) are applied to the duration of droughts in (a) and (b) under the null hypothesis of an equal distribution between two time series of duration of droughts. The p-values between the duration of summer scPDSI in OWDA and CESM is 0.003, which indicates that their distributions are statistically different to each other. This is also the case for all the annual indices, except between the SPEI and SOIL with the p-value of 0.87. The p-values between the indices in the transient and control simulations are all statistically similar, indicating that the distribution of duration of droughts in the transient simulation is in the range of variability of the control simulation. The p-values from the M-W tests between the transient and control simulations are presented in table A1.

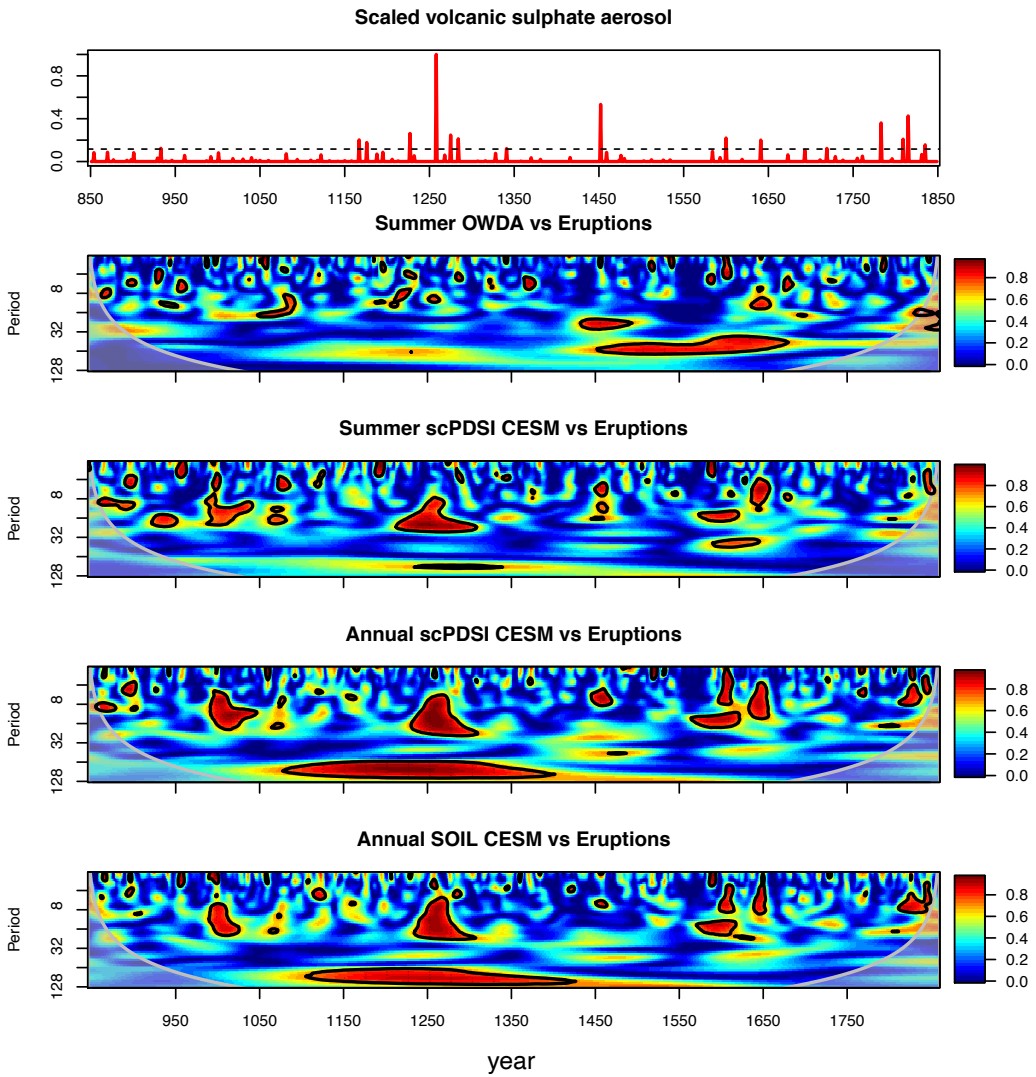

**Figure 6.** In the first panel: time series of scaled total stratospheric sulphate aerosol injections for the volcanic eruptions in 850 – 1849 AD (Gao et al., 2008). Dashed black line indicates the threshold for large eruptions with the stratospheric sulphate aerosol loading of more than 30 Tg. In the lower panels: wavelet coherence analysis between the time series of volcanic eruptions and the drought indices. The red shaded regions are where the coherence of two time series are statistically significant at 95% confidence level, estimated from Monte Carlo resampling of the time series.

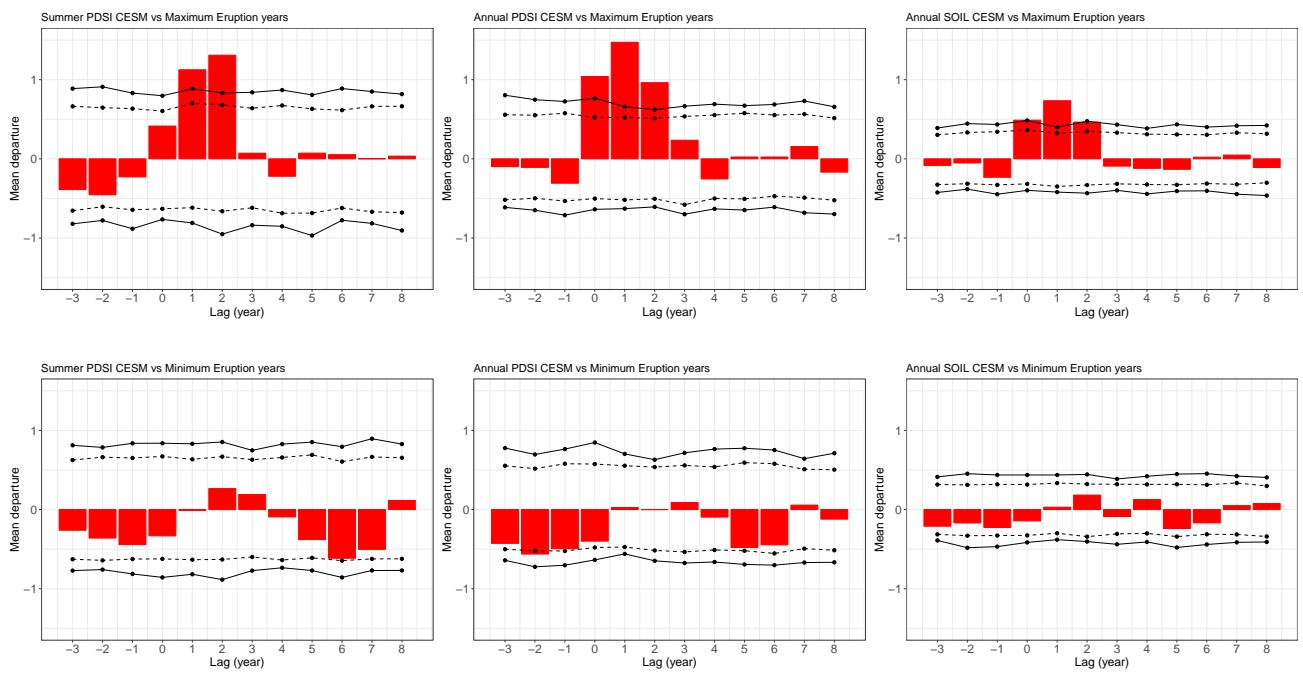

**Figure 7.** Superposed epoch analysis between the yearly drought indices in CESM (Summer scPDSI, annual scPDSI and SOIL) and years of eruptions for: (upper panels) the 16 largest eruptions, and (lower panels) the 16 smallest eruptions. The upside bars indicate the positive changes and downside bars show the negative changes before and after the eruptions. Continuous lines indicate the 99% and dashed lines the 95% confidence intervals from the bootstrap resampling of the time series. The list of the eruptions used for the analysis is in table A2.

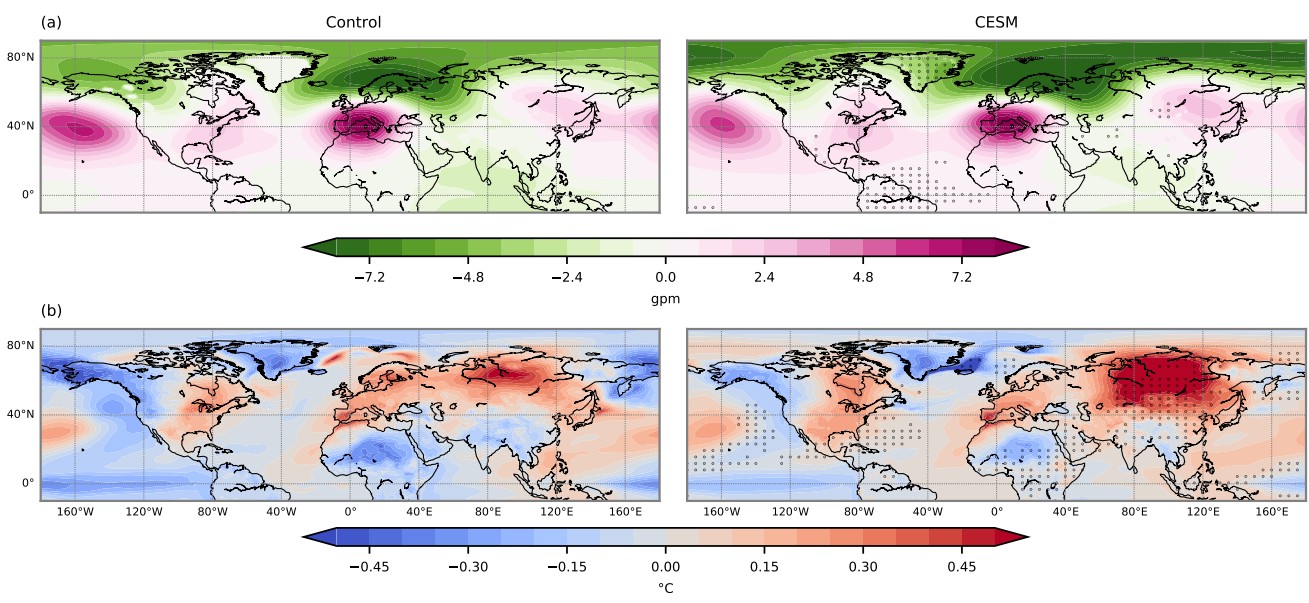

**Figure 8.** (a) Mean geopotential height anomaly at 850 hpa, and (b) mean surface temperature anomaly for the control (left) and transient (right) simulations during Mediterranean droughts in 850 – 1849 AD. Black dots on the composites of the transient simulation indicate the regions where the distributions between the control and transient simulations are statistically different to each other at 5% confidence level from the Mann-Whitney U-tests.

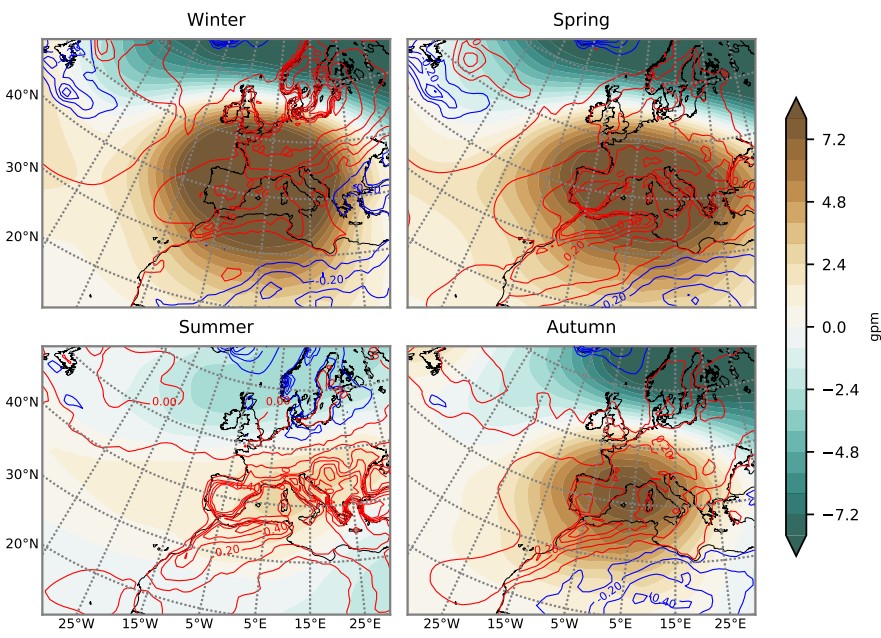

**Figure 9.** Mean geopotential height anomaly at 850 hpa (color shaded) and surface temperature anomaly (contours every $0.2°C$, positive in red and negative in blue) during Mediterranean droughts for each season in the transient simulation.

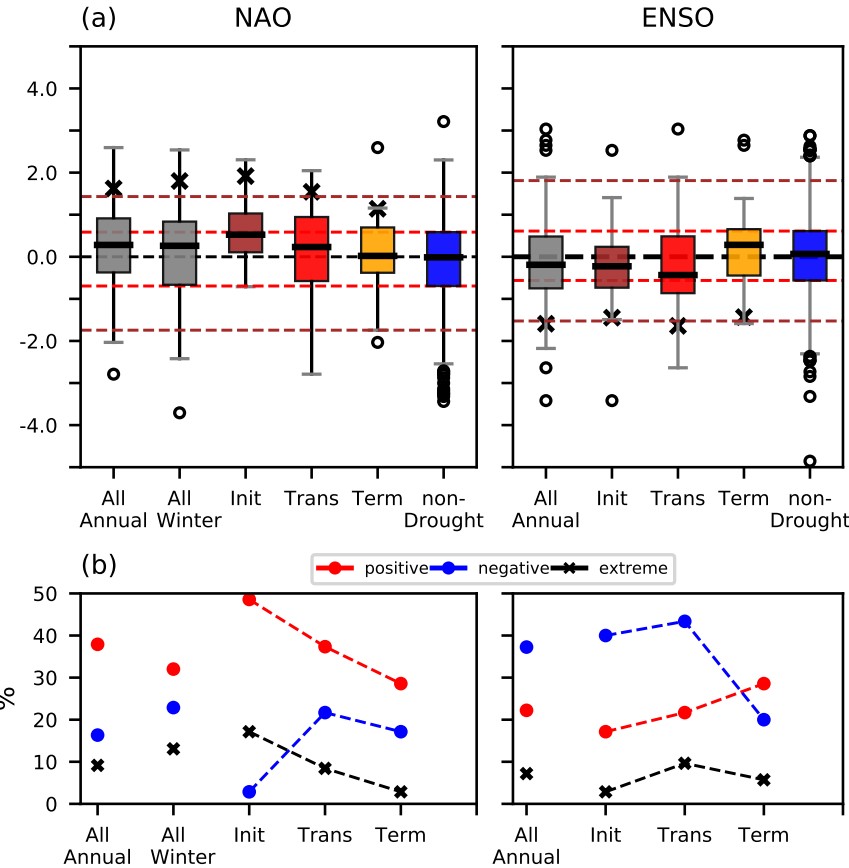

**Figure 10.** (a) Box plots of NAO and ENSO during multi-year Mediterranean droughts. Dashed red lines indicate the 25th and 75th percentiles of distributions of NAO and ENSO during non-drought periods which are taken as thresholds to discern relative negative/positive phases of NAO and ENSO. Dashed brown lines indicate the 5th and 95th percentiles of distributions which are taken as thresholds for extreme negative/positive NAO and ENSO. Black crosses represent the means of extreme positive NAO (values above the 95th percentiles), and the means of extreme negative ENSO (values below the 5th percentiles). (b) Frequencies of occurrence of positive and negative phases of NAO (left) and ENSO (right) in annual, winter, and each stage of droughts. Black crosses indicate the frequencies of occurrence of extreme positive NAO and extreme negative ENSO.

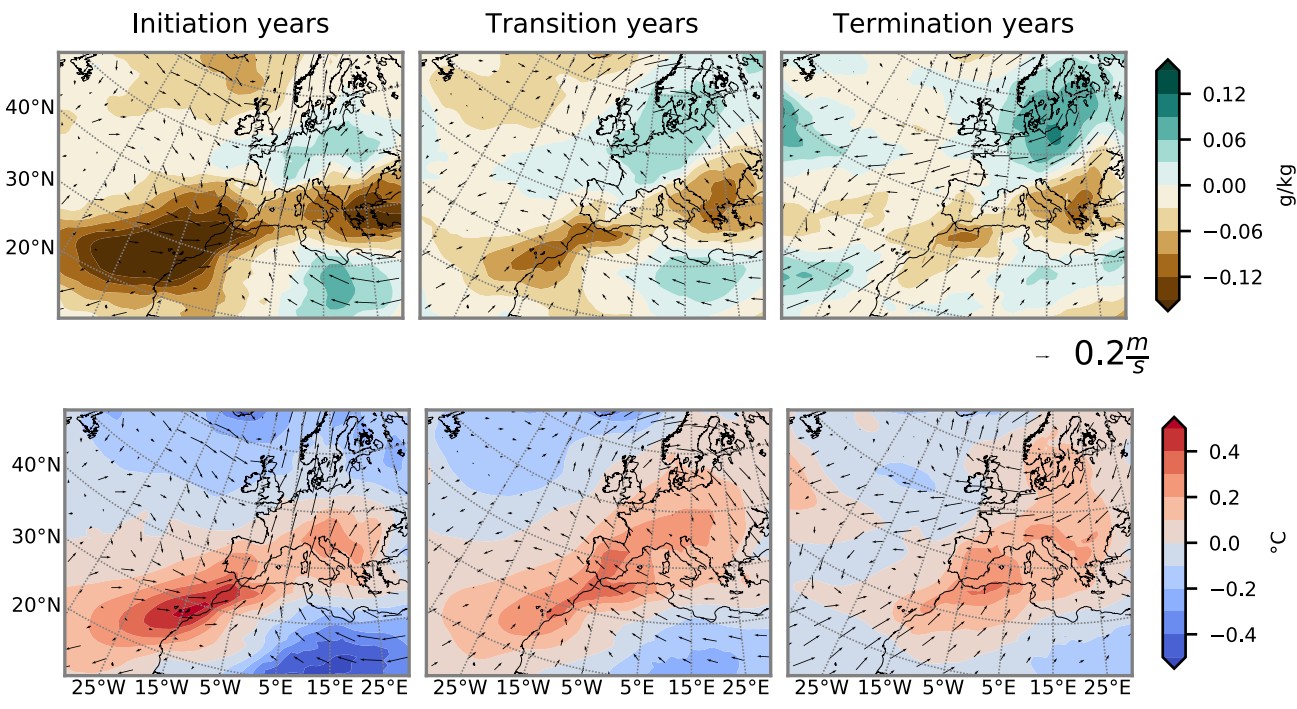

**Figure 11.** Evolution of atmospheric conditions in each stages of droughts. Anomalies of (above) specific humidity, and (below) temperature, both at 925 hPa during the initiation, transition and termination years of droughts. Arrows indicate winds at 925 hPa.

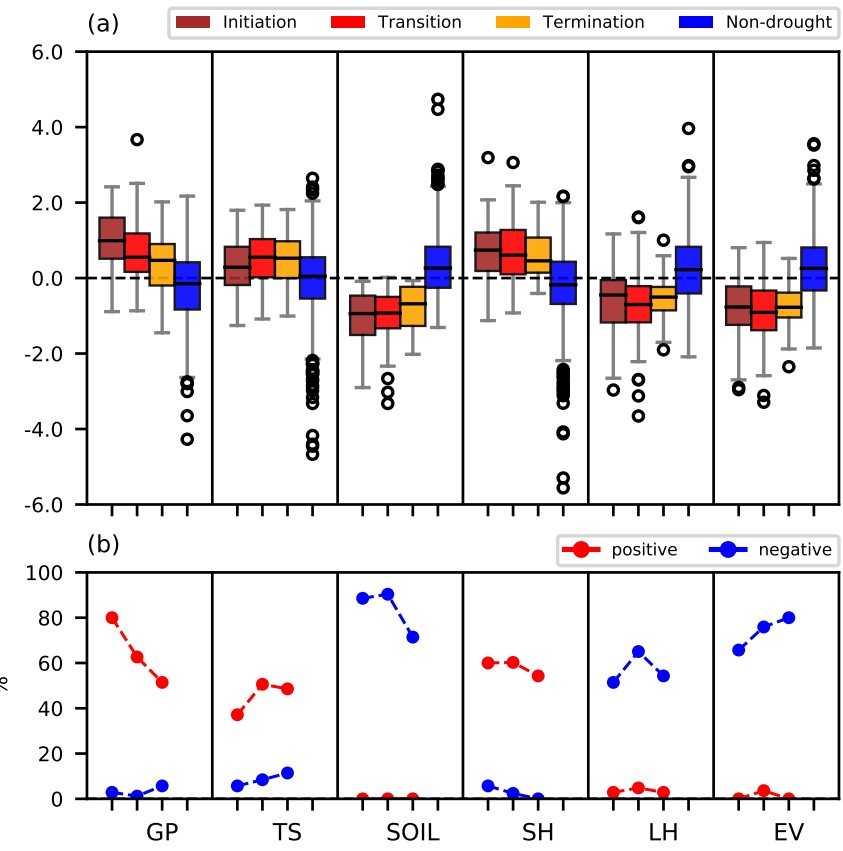

**Figure 12.** Same boxplot as Fig. 10 but for standardized regional atmospheric and soil variables over the region of study during Mediterranean droughts: anomalies of geopotential height at 850 hPa (GP), surface temperature (TS), soil moisture (SOIL), sensible heat flux (SH), latent heat flux (LH) and evapotranspiration (EV). (b) Frequencies of occurrences of positive and negative anomalies in each stage of droughts in order: initiation, transition and termination years.

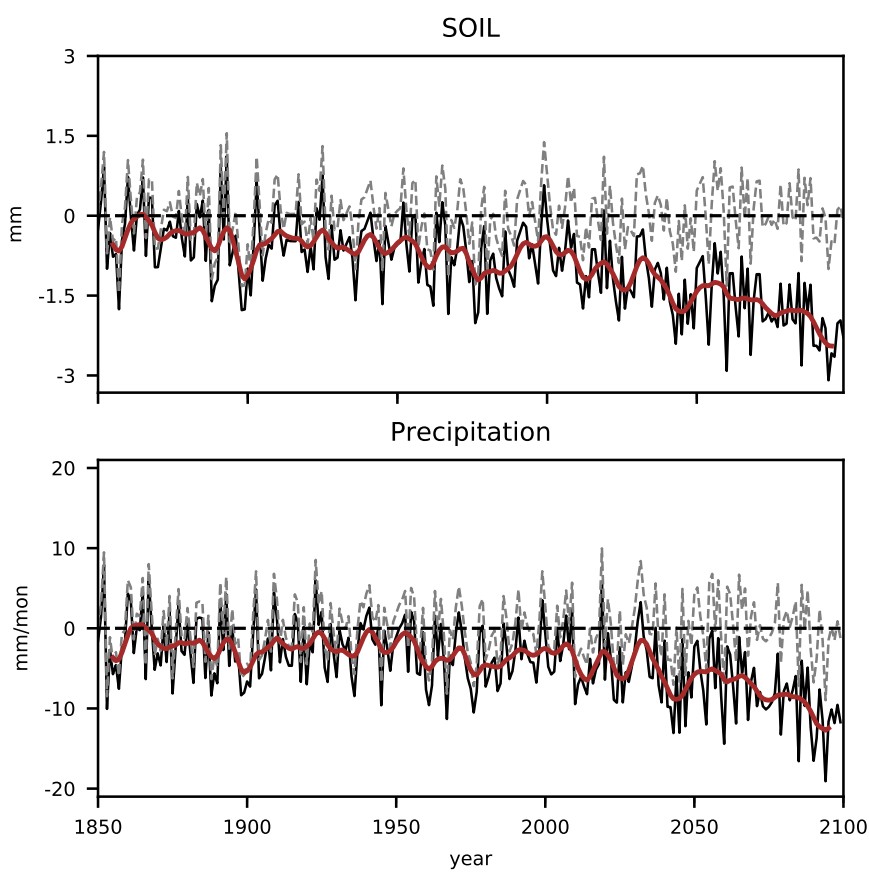

**Figure 13.** Time series of annual soil moisture (SOIL) and precipitation anomalies from 1850 to 2099 AD with respect to the 1000 - 1849 AD means. Brown lines indicate a 10-year running means, and dashed lines the detrended time series. The values of the removed trends is presented in Fig.A1

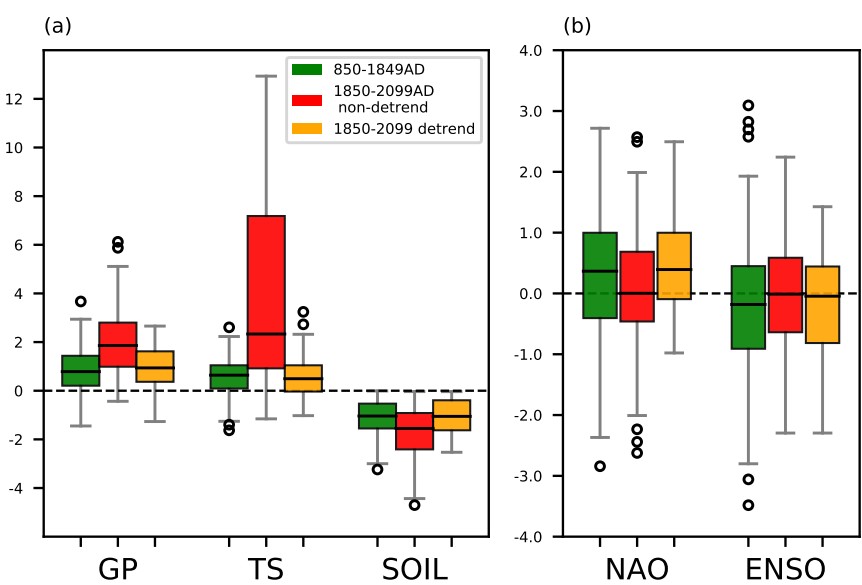

**Figure 14.** (a) Standardized regional variables: anomalies of geopotential height at 850 hPa (GP), surface temperature (TS) and soil moisture (SOIL) over the region of study, and (b) indices of large scale circulation patterns: NAO and ENSO during Mediterranean droughts for the period of 850 - 1849 AD (green), non-detrended 1850 - 2099 AD (red) and detrended 1850 - 2099 AD (yellow). The GP, TS and SOIL between the detrended 1850 – 2099 AD and the 850 – 1849 AD periods present the p-values from the Mann-Whitney U-tests of 0.09, 0.02 and 0.29, respectively. For NAO and ENSO, the p-values between these two periods are 0.19 and 0.29 for each.

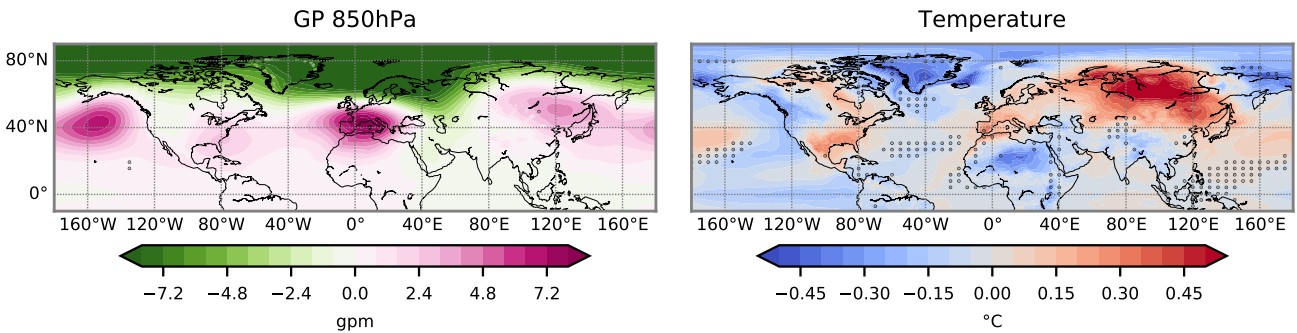

**Figure 15.** Detrended mean geopotential height anomaly at 850 hPa and surface temperature anomaly during Mediterranean droughts for the 1850 - 2099 AD. Black dots indicate the regions where the distributions between the detrended 1850 – 2099 AD and 850 – 1849 AD are statistically not similar at 5% confidence level (Fig. 8) from the Mann-Whitney U-tests.

**Appendix A**

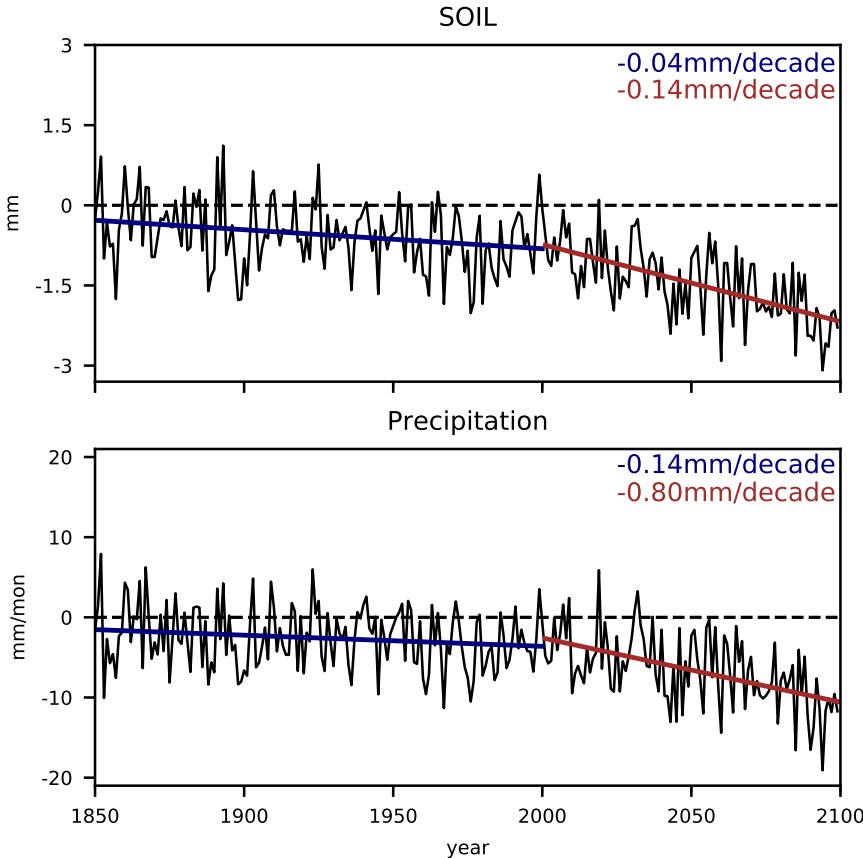

**Figure A1.** Time series of annual soil moisture (SOIL) and precipitation anomalies from 1850 to 2099 AD with respect to the 1000 - 1849 AD means for the detrending method. The least-square method is applied to two separate time period separately: 1850 - 2000 AD and 2001 - 2099 AD. The trends and their values for each period are also shown.

**Table A1.** P-values from the Mann-Whitney U-test between the drought indices in the control and transient simulations

|  | Control (400-yr) vs Transient (850 - 1849 AD) simulations |
|---|---|
| scPDSI-summer | 0.31 |
| scPDSI-annual | 0.35 |
| SPEI | 0.5 |
| SPI | 0.1 |
| SOIL | 0.43 |

**Table A2.** Years and total stratospheric sulphate aerosol injection of 16 largest and 16 smallest volcanic eruptions in 850 - 1849 AD from Gao et al. (2008)

| Largest eruptions | | Smallest eruptions | |
|---|---|---|---|
| year | total stratospheric sulphate aerosol loading (Tg) | year | total stratospheric sulphate aerosol loading (Tg) |
| 1258 | 257.91 | 1050 | 2.79 |
| 1452 | 137.5 | 1045 | 2.77 |
| 1815 | 109.72 | 1060 | 2.57 |
| 1783 | 92.96 | 1150 | 2.12 |
| 1227 | 67.52 | 1132 | 2.03 |
| 1275 | 63.72 | 987 | 1.92 |
| 1600 | 56.59 | 1794 | 1.88 |
| 1284 | 54.69 | 1316 | 1.83 |
| 1809 | 53.74 | 1503 | 1.72 |
| 1167 | 52.12 | 1158 | 1.56 |
| 1641 | 51.60 | 1213 | 1.56 |
| 1176 | 45.76 | 939 | 1.30 |
| 1835 | 40.16 | 1307 | 1.18 |
| 933 | 31.83 | 1358 | 1.09 |
| 1719 | 31.48 | 1142 | 0.82 |
| 1341 | 31.14 | 945 | 0.58 |