# Peer review of "Dynamics of the Mediterranean droughts from 850 to 2099 AD in the Community Earth System Model"

_Climate of the Past, 2020_

## Referee Comment (RC1) · Anonymous Referee #1 · 3 Aug 2020

Review of the manuscript

Dynamics of the Mediterranean droughts from 850 to 2099 AD in the Community Earth System Model

by Woon Mi Kim and Christoph C. Raible

submitted for publication in Climate of the Past

**General comments**

The authors investigate drought dynamics in a simulation with a comprehensive Earth System model during the last millennium and for a future GHG emissions scenario over the Mediterranean area. Their main results of the analysis of the climate simulation relate to a dominating influence of internal climate variability controlling drought in the pre-industrial times and hypothesize a stronger influence of external forcing factors, i.e. the increase in GHG concentrations for the historical period and even more pronounced for the GHG emission scenario.

The manuscript is well organized and written. In addition, the investigations include a comparison of different methods and what I really would like to emphasize, include the analysis of driving mechanisms controlling drought over the Mediterranean area, for instance related to large scale modes of atmospheric and coupled atmosphere-ocean dynamics. To my knowledge it's also the first study addressing drought dynamics specifically over the European Mediterranean area for the last millennium which makes it an important contribution for both, the modelling and the proxy community.

Below I listed a number of issues to improve the manuscript and to assess the model results in comparison with observational evidence.

*Abstract*

The abstract is brief and concise and summarizes the main conclusions of the study – maybe the authors can add some additional sentences on the uncertainties and limitations of the model-only study and include some basic statements on the suitability of the CCSM model to be used for drought studies over the Mediterranean realm.

*1 Introduction*

The authors should include a chapter on a more detailed description of the mean climatic characteristics of the Mediterranean area, especially during the winter half year when most of hydroclimatic variability plays a role. Moreover, it would be illustrative to elaborate in greater detail the spatial differences in hydroclimatic variability between the western and eastern Mediterranean area concerning the annual cycle (cf. references at the end of review by Dünkeloh and Jacobeit (2003), Luterbacher et al. (2006), Trigo et al. (1999), Peyron et al., (2017)).

A second point that might be also motivated in the introduction is why only a single model simulation with PMIP3-like forcings is investigated. Admittedly, the spatial resolution is one of the biggest advantages of the simulation, but also other simulations could have been addressed,

especially when large-scale areal averages are analyzed. Authors should try to motivate why CCSM4 in this version is outstanding and suited for drought investigations over the Mediterranean area. (cf also Coats et al. (2015) for a model-only studies over North American droughts)

*2.1 Description of the model and simulations*

The CCSM model has a very high spatial resolution, but I was wondering why the vertical resolution is quite low, consisting of only 26 levels. A number of PMIP3 models use a lower spatial resolution but with a considerably higher vertical resolution. I mention this issue because it might have important implications for the atmospheric dynamics, controlling precipitation variability, both spatially and temporally, over the Mediterranean area. Hence, a realistic simulation of those processes is pivotal for a realistic simulation of drought (or non-drought) dynamics.

The authors mention the orbital forcing is set to 1990 AD conditions for the control simulation – I guess this also applies for the transient simulation. Which effects could this have when the orbital parameters are not varying concerning the radiation changes, especially during the summertime in the course of the last millennium ?

Also, as the Mediterranean area in the northern region has a very vulnerable vegetation cover that is also important for hydrological dynamics, some words on the reconstruction and potential changes in land cover over the area would be informative for the readership. Likewise, the authors mention the soil model consisting of 15 layers, which is quite extensive for a global Earth System model. As soil dynamics also play a central role in the investigations carried out at a later stage, authors should add some more information on the soil model and highlight its importance, especially over the Mediterranean area.

*2.3 Drought definitions*

As I mentioned previously, I like the approach addressing several drought-related and quantifying indices, as results might be dependent on the respective metric used. I missed however a comparison of the general hydrological cycle for present-day climate in comparison with observational and/or re-analysis data sets. I suggest to at least perform a validation for i) the winter season for precipitation spatially resolved over the Mediterranean area and ii) the annual cycle separated over the western and eastern and northern and southern Mediterranean (cf. links for data sources at the end of this review) in the 2$^{nd}$ half of the 20$^{th}$ century. This is important to test whether the model is capable to reproduce the main climatic features in important on investigations in the context of drought (cf. López-Moreno et al., (2009) for present-day situation).

A second issue here is the question why the authors do not present a spatially resolved analysis for their study region. The areal extent of their region is quite large and planetary wave train structures might affect the area at the same time with different impacts. For instance, a ridge structure over the western Mediterranean can be accompanied by a trough structure at the same time over the Eastern Mediterranean with profound differences related to the hydrological impacts. A consequence might be that in situations with non pan-Mediterranean droughts, those dipole structures between east/west and north/south are cancelled out and the respective areal averages only contain a residual component that is not related to atmospheric circulation dynamics. Maybe the authors could at least mention how the usage of areal average might affect their conclusions.

*3.1 Quantification of droughts events over the Mediterranean: Selection of a drought index.*

A general issue investigating droughts over semi-arid regions like the European-Mediterranean area with a pronounced annual cycle relates to the high (mulit-annual) temporal and spatial variability of the availability of water resources. Therefore, drought or periods with scarcity of water are an intrinsic part of the climatic conditions over those regions. Likewise, this also applies for the opposite case with strong torrential rains leading to flooding and disastrous destructions over the respective areas. I think those points should be mentioned here or earlier in the introduction to put the drought terminology into context, underpinning that dry conditions are an integral part of the climate over those areas. Other, non climatic factors, for instance related to geology in terms of limestone with a high potential to effectively store water during winter and release it during summer could be mentioned. In addition, human impacts with steadily increasing demand for water resources play an important role interfering with the direct climatic driven changes in drought dynamics.

*3.3 Dynamics of multi-year droughts*

I liked this part because it links the (regional) drought dynamics over the Mediterranean area with large scale modes of atmospheric (NAO) and atmosphere-ocean (ENSO) variability. However, especially in terms of ENSO I suggest to use a more objective test metric, because in my opinion the numbers are not really convincing for a robust inference which state of the ENSO precedes Mediterranean droughts. The authors should motivate their definition of a positive NAO / ENSO state that should considerable deviate from mean or neutral conditions. For instance, the threshold values of the SST anomaly over the tropical Pacific is set to ±0.5 K. Authors might use a metric based on percentiles of the according index-PDFs and investigate the situation separated into full period and drought prone years to test the robustness of the according conclusion. This could eventually also allow a quantitative differentiation in moderate/strong events for NAO and ENSO and their impacts on Euro-Mediterranean droughts.

*3.4 Historical and Future conditions on droughts: 1850 to 2099 AD*

The authors use a very strong GHG scenario – I wonder how results change in simulations with less pronounced increase in GHG. Moreover, how can changes in vegetation cover and/or human water consumption play into drought dynamics purely based on climatic considerations ?

In this context it is again important to ask about the consequences if the main controlling factors (e.g. atmospheric circulation, Trigo et al., (1999)) are not realistically simulated. Are the models really able to realistically mimic the (change) of atmospheric circulation over the past and the following years ? This is especially important, given the fact that Mediterranean precipitation is characterized by very short-lived and very intense precipitation events initiated by meso-scale circulation patterns (e.g. Genoa low) that might not be represented well enough in those models.

*4 Conclusions*

The conclusions are a good summary – what I think is important to add one or two chapters with more critical comments and insights on the limitations and uncertainties involved in the study (e.g.

only single model used, validation of atmospheric circulation dynamics, importance of non-climatic events for drought dynamics), and also the implications in the context of model-proxy comparisons.

**Minor comments:**

Figure caption: If possible, please add below the technical description of the Figure a short sentence what are the main contents of the Figures for a better overview for the reader on the main conclusion of the respective plot(s).

Figure 1: Please include latitudes and longitudes in the figure – why is the eastern Mediterranean region not completely included into the analysis ?

**Additional data for comparison**

https://crudata.uea.ac.uk/cru/data/hrg/

https://www.dwd.de/EN/ourservices/gpcc/gpcc.html

**Additional references**

Brewer, S., Alleaume, S., Guiot, J., and A. Nicault, (2007): Historical droughts in Mediterranean regions during the last 500 years: a data/model approach, Clim. Past, 3, 355–366, https://doi.org/10.5194/cp-3-355-2007.

Coats, S., Cook, B.I., Smerdon, J.E.and R. Seager (2015); North American pancontinental droughts in model simulations of the last millennium. J Clim, 28, 2025-2043.

Dünkeloh A. and J. Jacobeit (2003) Circulation dynamics of Mediterranean precipitation variability 1948–1998. Int J Climatol, 23, 1843–1866.

López-Moreno, J. I., S. M. Vicente-Serrano, L. Gimeno, and R. Nieto (2009), Stability of the seasonal distribution of precipitation in the Mediterranean region: Observations since 1950 and projections for the 21st century, Geophys. Res. Lett., 36, L10703, doi:10.1029/2009GL037956.

Luterbacher, J., Xoplaki, E, Casty. C. et al. (2006): Mediterranean climate variability over the last centuries: A review, in: The Mediterranean Climate: an overview of the main characteristics and issues, In: Lionello, P., Malanotte-Rizzoli, P., and Boscolo R. (Eds.), Amsterdam, Elsevier, pp. 27-148.

Peyron, O., Combourieu-Nebout, N., Brayshaw, D., Goring, S., Andrieu-Ponel, V., Desprat, S., Fletcher, W., Gambin, B., Ioakim, C., Joannin, S., Kotthoff, U., Kouli, K., Montade, V., Pross, J., Sadori, L., and M. Magny (2017): Precipitation changes in the Mediterranean basin during the Holocene from terrestrial and marine pollen records: a model–data comparison, Clim. Past, 13, 249–265, https://doi.org/10.5194/cp-13-249-2017.

Trigo, I. F., Davies, T.D. and Bigg, G. R. (1999): Objective Climatology of Cyclones in the Mediterranean Region, J Clim, 12, 1685–1696.

---

## Referee Comment (RC2) · Anonymous Referee #2 · 13 Aug 2020

Review of manuscript

Dynamics of the Mediterranean droughts from 850 to 2099 AD in the Community Earth System Model

Authors: Woon Mi Kim and Christoph C. Raible

submitted for publication to Climate of the Past

General comments:

The authors present an interesting analysis of Mediterranean drought in the NCAR CESM- the work has relevance for both paleo drought dynamics as well as current/future drought in the region. Specifically, they use several drought metrics (which is a much appreciated comparison) to compare background drought frequency in a CESM Last Millennium simulation to variability in the Old World Drought Atlas then go on to use the CESM last millennium and historical/RCP8.5 extensions to examine ocean-atmosphere conditions associated with drought in past and present/future climate conditions. This is interesting work with valuable implications. However, in my opinion the authors have based their analysis and conclusions on the ability of the CESM to simulate pre-instrumental drought occurrence/frequency (and do not sufficiently prove that the CESM can do this) and draw several conclusions that appear to be based on visual comparisons of data in figures that I found hard to believe (and in some cases appeared simply incorrect) without further quantitative support.

Below I have listed my main concerns:

(1) CESM simulation data are not easily accessible without contacting the researcher who ran the simulations. (As noted below, I wanted to try replicating the authors' analysis by comparing the CESM data to the OWDA data, but the CESM data are not publicly available)

(2) Choice of geographic region for the Southern Mediterranean regional mean time series. I created a regional mean time series using the same region the authors used, but for the GPCCv2018 instrumental precipitation data and plotted the correlation coefficient (r-squared) for each grid point in the region- most of the grid points in the box only share  $\sim$ 20-30% of variance with the regional mean time series.

I wonder if perhaps some parts of the Mediterranean region experience drought at different times/magnitudes in the instrumental data (also in the CESM data)? Why did the authors choose this region?

Please provide more evidence that drought/precipitation in the region varies coherently (e.g., suggest showing more information in Figure 1 other than a map and a box).

CPD
(3) The authors gloss over a critical comparison of the paleo to the model data (lines 179-182) - they conclude the background drought statistics (occurrence/frequency) in the CESM are similar to the OWDA. Yet, an examination of figure 1c suggests to me that the drought occurrence in the model and paleo data are quite dissimilar- the bulk of droughts in the CESM are centered around 6-10 years in length, and in the OWDA the distribution is centered around ~1-4 year drought lengths. This discrepancy is quite striking to me, and I was surprised when the authors claim these distributions are comparable.

If the authors want to make this claim, I suggest using some sort of metric (e.g., something like a Mann-Whitney or Wilcoxon rank-sum test or some sort of distribution comparison metric) to show these two drought occurrence distributions are statistically similar. Even a report of the median, mean, and range would be more helpful than the visual comparison. I also suggest the authors use other metrics such as showing average drought occurrence per century (e.g., see Figure 3 in Parsons et al., 2018, J. Clim.).

Other suggestions include comparing the power spectra (PSD) of the OWDA and CESM PDSI. For example, I made Southern Mediterranean regional mean time series of PDSI from the OWDA and from the CESM1 LME run2 (this is an admittedly lower resolution version of CESM1; Otto-Bliesner et al., 2015; but the background drought statistics in the CESM LME and higher resolution versions of CESM are quite similar, at least in SW North America -e.g., Parsons and Coats, 2019, JGRA) over the 850-1849 CE time period. I found the power spectra show quite dissimilar behavior for the CESM and OWDA PDSI variables, with varying discrepancies as varying frequencies depending on how I standardize them.

(4) Comparison of CESM with instrumental/reanalysis data: the authors missed an opportunity to validate the performance of the CESM in the historical/instrumental era against instrumental/reanalysis data. The authors show (e.g., Figures 3,4,6,7) background geopotential height, SST, etc. anomaly patterns associated with drought, but

**CPD**
they have not used instrumental-based data to show the model can accurately simulate the observed climate, and I remain unconvinced the background drought statistics are similar to the OWDA (see Main Concern (3) above).

Authors could compare patterns associated with drought (using a metric such as 2D pattern correlation) in the model to observed/reanalysis geopotential height (ERA5 or 20th Century Reanalysis) and SST (NOAA ERSSTv5, HadSST, etc.), as well as drought occurrence in the model to instrumental data (GPCCv2018 precipitation, Dai PDSI, CRU precipitation).

Example of how other authors have made these comparisons among model and instrumental/reanalysis data: Figure 2 in Parsons et al., 2018, J Clim., Figure 2 in Coats et al. 2013, GRL, Figure 2 in Stevenson et al., 2015, J Clim.

Importantly, as the submitted paper is written without this comparison, I am left unsure/unconvinced the analysis presented in the paper is not just based on a model that can't simulate the relevant parts of the climate system for the study.

(5) The authors do not address several of the known shortcomings in the CESM model (e.g., frequency/strength of ENSO events; Parsons et al., 2017, J. Clim, Figure 6; Bellenger et al., Clim. Dyn, 2015 for a comparison of ENSO characteristics among models and instrumental data) and what the implications of these shortcomings could be for their study, especially because the authors make claims about likelihood of ENSO events before/during/after droughts.

I suggest the authors consider the findings of Ault et al. (2014, J Clim), who show that the background power spectra/statistical characteristics of drought/precipitation (e.g., white noise, power law, etc.) are critical for drought magnitude and duration, and many CMIP5-class models do not show the same background variability as instrumental/paleo data in many regions.

(6) Especially when future warming is considered, it is important to focus on metrics
of drought that don't just focus on 'atmospheric centric' supply and demand, especially if ecosystem/water resource drought impacts are important. See Swann et al., 2016, PNAS, and Swann (2018) who note that drought severity/impacts in a warming climate can be grossly overestimated by use of variables/metrics such as PDSI. I appreciate that the authors included 10cm soil moisture, but given that surface soil water content can basically just follow precipitation variability in many regions, and thus not really reflect full depth soil moisture trends (e.g., Berg et al., 2016), I think it would be helpful for the authors to show that they are analyzing variables actually relevant for plants/ecosystems/water resources in a warming climate, and not just supply/demand from the atmosphere. At least a discussion of some of these points could really strengthen the paper.

Č Specific comments:

Lines 13-14: the authors just list one or two types of drought (meteorological), but what about hydrological, agricultural/ecosystem, socioeconomic types of drought?

Line 22: 'climate hot-spot'- please cite a paper that shows this

Line 23: 'increase in drought episodes' - again, please cite a paper supporting this

Lines 45-46: 'attributed to the increase in the atmospheric greenhouse gases (GHG) concentration, which causes ... decrease in precipitation over the region' - citation?

Line 52: 'unprecedent intense drought projections' - citation?

Lines 63-64: The separation of ocean-atmosphere conditions during various drought stages has been done before- nice to acknowledge previous work (e.g., Parsons and Coats, 2019; Namias, 1960).

Lines 76-77: 'warm-dry temperature-hydroclimate co-variability at multidecadal timescales' confusing wording

Line 92: 'high horizontal resolution' is subjective (and now closer to 'average resolution'

**CPD**
in many CMIP6 models)

Line 102: Why not use the CESM LME (Otto-Bliesner et al., 2015)? There are more iterations, with several RCP8.5 extensions (and a much longer 1000 yr piControl run that is easier to compare w the last millennium runs given the similar length of simulations), allowing for a more complete analysis of internal variability. Is the background climate state that much better in the  $\sim$ 1 degree vs the  $\sim$ 2 degree version of the model?

I ask because the authors explicitly state on lines 119-120 that they are interested in studying internal vs externally forced variability, and multi-model ensembles provide an ideal experimental framework for doing this.

Line 107: the years 2001-2020 AD/CE are not the future

Lines 103-112: Suggest just citing Lehner et al. for the model description

Line 127: As in Main Concern (2), please show the region varies coherently in instrumental/paleo and the version of CESM used here

Lines 131-132: removing a linear trend over the 1850-2099 time period looks quite problematic to me (e.g., Figure 9)- removing a linear trend over this time period will add in non-climatic variability artifacts from the trend removal. It looks to my eye like there is a trend  $\sim$ 1900-2000, then a separate trend  $\sim$ 2000-2099.

Line 149-150: linear temperature trend is removed, but then authors study the impacts of warming using this drought metric, which includes temperature...so have the authors removed temperature changes, then try to study the impacts of warming on drought? This reasoning doesn't make sense to me. Perhaps a more clear explanation of trend removal would help (?).

Lines 140-155: As in Main Concern (6): I think all of these drought metrics/variables, with the exception of upper 10cm soil moisture, do NOT reflect actual impact on plants/ecosystems in a warming climate. Also, upper soil water content can diverge from deeper soil water – authors should show that this is a useful metric here that is
distinct from precipitation alone if they want to argue that their study has relevance for ecosystem impacts.

Lines 161-164: This drought counting method appears similar to Herweijer et al. 2007; Coats et al. 2013b- did the authors come up with this metric, or can they use a similar metric to previously published work (if so, please cite) to maintain consistency across the literature?

Lines 168-170: see above note about similar methods in Parsons and Coats as well as Namias.

Lines 179-183: As in Main Concern (3): Please be more quantitative. To my eye, these distributions do not appear similar- the OWDA shows droughts that are mostly 1-4 yrs, and the CESM shows droughts centered around 8 yrs. Please use a more quantitative method to compare drought time series power spectrum and/or drought frequency in paleo and model data.

Lines 187-188: difficult to visually compare these different drought metrics in lower panels in Figure 2 because the x axis limits are different.

Lines 204-205: 'no noticeable changes in occurrence of droughts' - is this to the eye? Can you use a more quantitative method to show this (e.g., running counts of droughts in 50 yr windows or something like that)?

Lines 205-206: 'not driven by external forcing': again, this conclusion appears to be drawn based on a visual comparison, which seems insufficient to me. Lehner et al. (2015, ESD, Figure 5) use running correlation to compare model output, which I imagine could be applied here, as could some sort of wavelet/coherence analysis between volcanic forcing time series and the OWDA and CESM data. Also, Superposed Epoch Analysis or Composite Analysis could be used with volcanic forcing time series/large eruptions. At minimum, it would be great to see a time series showing the external forcing to be able to compare to the drought time series in Figure 2.

CPD
Line 209-210: sentence wording is confusing/complicated

Lines 211-215: So if the r value is 0.78, doesn't this imply that only  ${\sim}60\%$  of variance is shared by the two time series?

Lines 218-220: 'control simulation presents 29 droughts'- this comparison with the transient simulation is non-sensical/misleading given the two simulation lengths are different. Can the authors instead present the average numbers of droughts of various lengths per century (e.g., Parsons et al., 2018; Coats et al., 2015, Figure 5). This gets around the issue of having different length time series and gives more meaningful information about drought risk standardized to a given time window (e.g., number of droughts per 100 or 500 years).

222-224: Is this the first time these patterns have been presented? Seems that a paper like Markonis et al. 2018 (Nature Communications) or other similar papers have previously presented similar patterns associated with hydroclimatic variability.

Lines 229-236: Similar to the point I raise in Main Concern (5)- It is well documented that this model simulates ENSO events that are too strong and too frequent (e.g., Bellenger et al., Clim. Dyn., among others)- how does that impact these results? For example, if the model simulates too strong, too regular ENSO events that unrealistically influence global climate, then is it surprising that a signal from ENSO is apparent in European drought/climate? And is this finding meaningful if it's based on model bias?

Figure 3 caption: the caption states 'means are not statistically significant'- unclear. Please be more specific. Also please clarify if data are annual, JJA, etc. in figure caption. Additionally, the significance dots are nearly impossible to see on the dark red/blue background

Lines 246-250: Are these % changes in drought/rainfall meaningful (e.g., for agriculture, ecosystems), or do these changes fall well within normal climate variations that
don't have a large impact? Also, is the background variability (e.g., standard deviation, mean) of rainfall in the CESM realistic, or can we chalk this up to model bias?

Lines 254-255: similar to Main Concern (4), what about in 20th century reanalysis, ERA5, or some similar reanalysis product vs GPCCv2018 or CRU precip? Or Dai PDSI?

Lines 257-260: 'The starting point is...to one or both of them'- confusing wording

Lines 262-269: So in other words, there is about equal odds of being in a drought during various NAO or ENSO phases? This seems important because the authors claim on lines 294-295 that a certain combination of NAO and ENSO conditions are important for initiating drought. . .but it appears to me as though there are nearly equal odds of this happening (~60%) based on the phase of NAO/ENSO. Is this interpretation incorrect?

Lines 298-310: I don't see how Fig 8 proves the point. Basically, it looks to me as though drought starts off dry and then transitions to less dry conditions at end of drought, and this is distinct from wet years.

Lines 325-327: Similar to Main Concern (4); I have not been shown how the model performs compared to instrumental/reanalysis for the relevant variables over Europe/Mediterranean, so these conclusions don't mean a lot to me.

Lines 337-340: 1) I see no major changes in distribution of drought in Figure 10- are these distributions distinct? Please see previous comments related to statistically distinguishing distributions (and not visually distinguishing), especially when they appear to overlap. 2) Any future changes in ENSO in this model should be interpreted with caution as most CMIP5 models, including this one as far as I can remember, struggle to reproduce the observed trends in the tropical Pacific (see Coats and Karnauskas, 2017, GRL as well as Seager et al., 2019, Nature Climate Change).

Lines 344-345: As figure 9 shows, trends in the region are not linear 1850-2100, so

CPD
trend removal is problematic over this time period. Perhaps it makes sense to remove the trend 2000-2099, but otherwise the authors could be adding an artifact of trend removal into the analysis.

Lines 358-359: 'our analysis shows that the overall similarities': as stated above, the authors never actually showed this statistically, just a visual comparison.

Line 383: 'climate over the region to a drier climate have started earlier than reported in the modern observational era': to back up a statement like this, I'd again like to see that the model is simulating instrumentally observed climate during the relevant temporal overlap in the historical run (e.g., show Mediterranean precip./PDSI time series in model and instrumental data) before claiming that any drying has happened earlier than reported.

Technical/typo errors and corrections:

Line 4: 'the internal variability' – perhaps a stylistic choice, but suggest removing 'the' Line 37: 'on ENSO' – do the authors mean 'to ENSO'? Line 40: change to 'autumn and spring' Line 74: during 'the' last millennium

References:

Ault, T. R., J. E. Cole, J. T. Overpeck, G. T. Pederson, and D. M. Meko (2014), Assessing the Risk of Persistent Drought Using Climate Model Simulations and Paleoclimate Data, J. Clim., 27, 7529-7549. Ault, T. R., J. E. Cole, J. T. Overpeck, G. T. Pederson, S. St George, B. Otto-Bliesner, C. A. Woodhouse, and C. Deser (2013), The Continuum of Hydroclimate Variability in Western North America during the Last Millennium, J. Clim., 26, 5863-5878. Bellenger, H., E. Guilyardi, J. Leloup, M. Lengaigne, and J. Vialard (2014), ENSO representation in climate models: from CMIP3 to CMIP5, Clim. Dyn., 42, 1999-2018. Berg, A., J. Sheffield, and P. C. Milly (2017), Divergent surface and total soil moisture projections under global warming, Geophys. Res. Lett., 44(1), 236-244. Coats, S. and K. B. Karnauskas (2017), Are simulated and observed twentieth
century tropical Pacific sea surface temperature trends significant relative to internal variability?, Geophys. Res. Lett., 44, 9928-9937. Coats, S., B. I. Cook, J. E. Smerdon, and R. Seager (2015), North American pancontinental droughts in model simulations of the last millennium, J. Clim., 28, 2025-2043. Coats, S., J. E. Smerdon, R. Seager, B. I. Cook, and J. F. Gonzalez-Rouco (2013), Megadroughts in Southwestern North America in ECHO-G Millennial Simulations and Their Comparison to Proxy Drought Reconstructions, J. Clim., 26, 7635-7649. Coats, S., J. E. Smerdon, B. I. Cook, and R. Seager (2013), Stationarity of the tropical pacific teleconnection to North America in CMIP5/PMIP3 model simulations, Geophys. Res. Lett., 40, 4927-4932. Coats, S., J. E. Smerdon, B. I. Cook, and R. Seager (2015), Are simulated megadroughts in the North American Southwest forced?, J. Clim., 28, 124-142. Herweijer, C., R. Seager, E. R. Cook, and J. Emile-Geay (2007), North American droughts of the last millennium from a gridded network of tree-ring data, J. Clim., 20, 1353-1376. Markonis, Y., M. Hanel, P. Máca, J. KyselÃi, and E. R. Cook (2018), Persistent multi-scale fluctuations shift European hydroclimate to its millennial boundaries. Nature communications. 9(1), 1-12. Namias, J. (1960), Factors in the initiation, perpetuation and termination of drought, IASH Commission of Surface Waters Publication, 51, 81-94. Otto-Bliesner, B., E. C. Brady, J. Fasullo, A. Jahn, L. Landrum, S. Stevenson, N. Rosenbloom, A. Mai, and G. Strand (2015), Climate Variability and Change since 850 C.E.: An Ensemble Approach with the Community Earth System Model (CESM), Bull. Amer. Meteor. Soc. Parsons, L. A. and S. Coats (2019), Ocean-atmosphere trajectories of extended drought in Southwestern North America, Journal of Geophysical Research: Atmospheres, 124(16), 8953-8971. Parsons, L. A., S. Coats, and J. T. Overpeck (2018), The continuum of drought in Southwestern North America, J. Clim. Parsons, L. A., G. R. Loope, J. T. Overpeck, T. R. Ault, R. Stouffer, and J. E. Cole (2017), Temperature and precipitation variance in CMIP5 simulations and paleoclimate records of the last millennium, J. Clim. Seager, R., M. Cane, N. Henderson, D. Lee, R. Abernathey, and H. Zhang (2019), Strengthening tropical Pacific zonal sea surface temperature gradient consistent with rising greenhouse gases. Nature Climate Change, 9, 517. Stevenson,

**CPD**
S., A. Timmermann, Y. Chikamoto, S. Langford, and P. DiNezio (2015), Stochastically generated north american megadroughts, J. Clim., 28, 1865-1880. Swann, A. L. (2018), Plants and drought in a changing climate, Current Climate Change Reports, 4(2), 192-201. Swann, A. L., F. M. Hoffman, C. D. Koven, and J. T. Randerson (2016), Plant responses to increasing CO2 reduce estimates of climate impacts on drought severity, Proceedings of the National Academy of Sciences, 113(36), 10019-10024.

---

## Author Comment (AC1) · 9 Oct 2020

We would like to thank the referee 1 for the constructive feedbacks and insightful comments. We appreciate the time and effort the referee dedicated to review our manuscript, which helped us to improve our presentation.

Our response to the referee's comments is attached as supplement.

Please also note the supplement to this comment:
https://cp.copernicus.org/preprints/cp-2020-79/cp-2020-79-AC1-supplement.pdf

---

## Author Comment (AC2) · 9 Oct 2020

We would like to thank the referee 2 for the constructive feedbacks and insightful comments. We appreciate the time and effort the referee dedicated to review our manuscript, which helped us to improve our presentation.

Our response to the referee's comments is attached as a supplement.

Please also note the supplement to this comment:
https://cp.copernicus.org/preprints/cp-2020-79/cp-2020-79-AC2-supplement.pdf

---

## Author Response (AR1)

Dr. Hugues Goosse, editor
Climate of the Past

Woon Mi Kim
Physics Institute and Oeschger Centre for Climate Change Research, University of Bern
Sidlerstrasse 5, 3012 Bern, Switzerland

25[th] November

Dear editor Dr. Hugues Goosse,

We would like to thank you for the opportunity to resubmit our revised manuscript. We would also like to thank again the reviewers for their constructive feedbacks and insightful comments that helped us to improve our presentation. We have addressed the reviewers' comments and you can find our point-to-point responses to each of the reviewers below in order (Response to the referee 1 and 2) and the mark-up revised manuscript at the end of this file.

Sincerely,

Woon Mi Kim[1] and Christoph C. Raible

[1]Corresponding author: woonmi.kim@climate.unibe.ch

Manuscript cp-2020-79
**Response to the referee 1**

We would like to thank again the referees for their constructive feedbacks and insightful comments. We appreciate the time and effort the referee dedicated to review our manuscript, which helped us to improve our presentation. We have incorporated the suggestions made by them, and below you find our responses to the referees' comments (in blue).

*Major comments:*

*Abstract*

*1. The abstract is brief and concise and summarizes the main conclusions of the study – maybe the authors can add some additional sentences on the uncertainties and limitations of the model-only study and include some basic statements on the suitability of the CCSM model to be used for drought studies over the Mediterranean realm.*

**Thanks for the comment. As the model related limitations and uncertainties are quite ample in topics (the modes of variability, mid-latitude dynamics and precipitation, comparison to proxies and uncertainties in future scenario), we provided more extensive discussion on these issues in the conclusion (Sect. 4) and added a new section that validates the model (Sect. 3.1.) in the revised manuscript.**

 *Introduction*

*2. The authors should include a chapter on a more detailed description of the mean climatic characteristics of the Mediterranean area, especially during the winter half year when most of hydroclimatic variability plays a role. Moreover, it would be illustrative to elaborate in greater detail the spatial differences in hydroclimatic variability between the western and eastern Mediterranean area concerning the annual cycle (cf. references at the end of review by Dünkeloh and Jacobeit (2003), Luterbacher et al. (2006), Trigo et al. (1999), Peyron et al., (2017)).*

**We agree with the referee's point, thus, we included a more detailed description on the mean climate of the Mediterranean in the introduction (Sect. 1. lines 37 – 56) mentioning about the spatial differences between west and east side during the winter and influence of large scale circulation patterns in different regions of Mediterranean:**

**Sect. 1, lines 37 – 46) "Overall, the region shows mild and wet winters, and hot and dry summers (Lionello et al., 2006). The variability of precipitation is not uniform across the entire Mediterranean area. The western and eastern regions show different precipitation regimes during the winter. A regional mode of circulation that explains a large percentage of the variability in winter is characterized by opposite pressure and precipitation patterns between the west-central and eastern Mediterranean regions, known as Mediterranean Oscillation (Dunkeloh and Jacobeit, 2003). Besides, the regional precipitation is strongly influenced by the mid-latitude storm tracks and cyclones, which become stronger during**

the winter (Lionello et al., 2016, Raible et al., 2007; Raible et al., 2010; Ulbrich et al., 2006), regional cyclones (Alpert et al., 1990), and large-scale modes of variability, such as the North Atlantic Oscillation (NAO), East Atlantic - West Russian pattern (EA - WR) and El Niño-Southern Oscillation (ENSO) (Lionello et al., 2006; Raible, 2007)."

*3. A second point that might be also motivated in the introduction is why only a single model simulation with PMIP3-like forcings is investigated. Admittedly, the spatial resolution is one of the biggest advantages of the simulation, but also other simulations could have been addressed, especially when large-scale areal averages are analyzed. Authors should try to motivate why CCSM4 in this version is outstanding and suited for drought investigations over the Mediterranean area. (cf. also Coats et al. (2015) for a model-only studies over North American droughts).*

As we said in the first response phase, we provided more explanations on the reason and benefit of using this single model in the introduction (Sect. 1.) and conclusion (Sect. 4.).

Sect. 1. lines 123 – 129) "The spatial resolution of the model is a clear advantage for our study on a relatively small confined area. The precipitation of the region is strongly influenced by extratropical cyclones and in general, GCMs have difficulties in reproducing in full degree the dynamics and precipitation associated with these meso-scale phenomena (Raible et al. 2007; Watterson et al., 2006). Nevertheless, these atmospheric dynamics and precipitation are better represented in GCMs with finer spatial resolution (Champion et al., 2011; Watterson et al., 2006). Hence, using a model that provides a seamless simulation for period 850 – 2099 AD guarantees an improved representation of precipitation related processes, thus, drought associated mechanisms over the region."

Sect. 4. lines 561 – 568) "Fifthly, it is important to mention that our analysis is based on a single model output and this raises questions related to single model studies, such as boundary condition problems and model-dependent biases and physics (PAGES Hydro2kConsortium, 2017). Nevertheless, for a small confined area that surrounds a large body of water mass, the Mediterranean Sea, and the land coverage is limited, we found the necessity to use a simulation with a finer resolution to represent the regional climate better. In the end, our study provides a useful understanding on the long-term variability and mechanisms of Mediterranean droughts by analyzing the entire last millennium. We addressed the influences of external and internal variability on Mediterranean droughts and distinctly different roles of the large-scale modes of variability and regional circulation during the different stages of multi-year droughts."

Additionally, in order to assess more clearly the role of volcanic forcing on droughts, the wavelength coherence analysis is included in the result section (Sect. 3.2) in lines 333 - 341 and figure 6 in the revised manuscript.

*Description of the model and simulations*

*4. The CCSM model has a very high spatial resolution, but I was wondering why the vertical resolution is quite low, consisting of only 26 levels. A number of PMIP3 models use a lower*

*spatial resolution but with a considerably higher vertical resolution. I mention this issue because it might have important implications for the atmospheric dynamics, controlling precipitation variability, both spatially and temporally, over the Mediterranean area. Hence, a realistic simulation of those processes is pivotal for a realistic simulation of drought (or non-drought) dynamics.*

**We provide here the same response as the first response phase: one hint that the vertical resolution is sufficient is given by the comparison of the simulation with the reanalysis data. We found that the correlations between geopotential heights (at 850 and 500 hPa) and the scPDSI in the model during droughts for the period of 1901-2000 seem to be in range with the correlation fields of the reanalysis data (NOAA 21th Century Reanalysis V3; Compo et al., 2011), which you can see the figure-a below. Thus, the model mimics reasonably well the atmospheric dynamics associated with droughts.**

[Figure]

**Figure-a. Fields of correlation between the monthly scPDSI and geopential height at (a-above) 850 hPa and (b-below) 500 hPa for the period of 1901-2000 in (left) the NOAA 21th century reanalysis V3 data and (right) CESM. Regions where the correlations are not statistically significant at 5% level by Mann-Whitney U test are dashed.**

**In the revised manuscript, we only included the correlation at 850 hPa (figure 4 and Sect. 3.1. in lines 269 - 278 in the revised manuscript), as the statement derived from these maps of different heights are the same.**
**Note also that changing the vertical resolution is not an easy task, as then the model parameterizations need to be partly retuned. So, we follow the NCAR suggestions and used a version which was officially released by the NCAR (i.e. a version which is rigorously tested).**

*5. The authors mention the orbital forcing is set to 1990 AD conditions for the control simulation – I guess this also applies for the transient simulation. Which effects could this have when the orbital parameters are not varying concerning the radiation changes, especially during the summertime in the course of the last millennium?*

**We provide here the same response as the first response phase: certainly, the change in orbital parameters from 850CE led to a progressive change in insolation in the Northern Hemisphere, such as the increase in summer insolation and a shift of the maximum month**

of insolation at high latitudes (Schmidt, 2011). However, the impact of these changes on the climate during the last millennium is rather small compared to the other forcings, such as the total solar irradiance and volcanic eruptions (PAGES Hydro2k Consortium, 2017). Clearly, on longer time scales such as the last 6000 years, the orbital forcing has a stronger effect and cannot be ignored.

*6. Also, as the Mediterranean area in the northern region has a very vulnerable vegetation cover that is also important for hydrological dynamics, some words on the reconstruction and potential changes in land cover over the area would be informative for the readership. Likewise, the authors mention the soil model consisting of 15 layers, which is quite extensive for a global Earth System model. As soil dynamics also play a central role in the investigations carried out at a later stage, authors should add some more information on the soil model and highlight its importance, especially over the Mediterranean area.*

We included in the conclusion (Sect. 4.) a paragraph discussing on complex interaction among vegetation, soil and atmosphere. Instead of focusing only on Mediterranean vegetations, we elaborated this part more in general way, as the roles of plants and their interaction with climate in extreme hydrological conditions are still not fully understood.

 Sect. 4. lines 569 – 579) "Lastly, we emphasize again the importance of assessing different drought indices in the paleoclimate context, also in the present and future warming scenario. Assessing different drought metrics is important for drought studies, as the most commonly used drought indices are based on water balance only from the atmospheric moisture supply and demand, and they tend to magnify drought risks in the future warming scenario (Berg and Sheffield, 2018; Mukherjee et al., 2018; Swann et al., 2016). For a more comprehensive picture, droughts need be quantified with indices that can also reflect water stress on plants and ecosystem, and complex interactions among soil, atmosphere and vegetation. We used the upper 10 cm soil moisture anomaly to partially tackle this issue and derived spatial patterns associated with Mediterranean droughts based on this index. Still, the upper 10 cm soil moisture do not fully involve a complex interaction in the soil and atmosphere occurring during droughts. As vegetation is known to have more complex responses to the changing climate and droughts (Swann 2018; Swann et al., 2016), the role of vegetation in extreme hydrological events can provide a more comprehensive view on drought mechanisms."

Regarding the description on the soil model, the referee 2 commented that we can shorten the model description just by citing Lehner et al. (2015), as more details are already explained in that literature. Therefore, we did not mention more in detail about each components model, including the soil model, and the readers can directly refer to Lehner et al. (2015), and related literature.

*Drought definitions*

*7. As I mentioned previously, I like the approach addressing several drought-related and quantifying indices, as results might be dependent on the respective metric used. I missed however a comparison of the general hydrological cycle for present-day climate in comparison*

*with observational and/or re-analysis data sets. I suggest to at least perform a validation for i) the winter season for precipitation spatially resolved over the Mediterranean area and ii) the annual cycle separated over the western and eastern and northern and southern Mediterranean (cf. links for data sources at the end of this review) in the 2nd half of the 20th century. This is important to test whether the model is capable to reproduce the main climatic features in important on investigations in the context of drought (cf. López-Moreno et al., (2009) for present-day situation).*

**As we said in the first response phase, we included a new section that compare the observation, OWDA and model: "Sect 3.1. Validation of CESM: comparison among observation, proxy and model (1901 – 2000 AD).". In the section, we performed the comparison of mean seasonal spatial and annual cycle of precipitation between the observation and model over the region of study in the validation section.**

*8. A second issue here is the question why the authors do not present a spatially resolved analysis for their study region. The areal extent of their region is quite large and planetary wave train structures might affect the area at the same time with different impacts. For instance, a ridge structure over the western Mediterranean can be accompanied by a trough structure at the same time over the Eastern Mediterranean with profound differences related to the hydrological impacts. A consequence might be that in situations with non pan-Mediterranean droughts, those dipole structures between east/west and north/south are cancelled out and the respective areal averages only contain a residual component that is not related to atmospheric circulation dynamics. Maybe the authors could at least mention how the usage of areal average might affect their conclusions.*

**As we said in the first response phase, we elaborated better our motivation to focus on this specific averaged region in the introduction (Sect. 1.), and the reason for selecting the extent of the region in the model description and methods section (Sect. 2.)**

**Sect. 1. lines 116 - 121) "We choose this specific area, as this region has been affected by recent large scale droughts (Garcia et al., 2019; Spinoni et al., 2017), which can be seen as pan-west-central Mediterranean droughts. Moreover, the region shows coherent desiccation in the future scenario (Dubrovsky et al., 2014). From now on, for simplicity, we refer to the west-central Mediterranean region in our study simply as the Mediterranean region. We focus on understanding the dynamics that induce past persistent pan-regional multi-year droughts, and whether the dynamics that induce droughts in the past will change in the historical and future periods with the anthropogenic increase in GHG."**

**Sect. 2.2. lines. 153 – 157) "The focus area of the study is the western and central Mediterranean region confined to 15°- 28°E and 33°- 45°N (Fig. 01). The extent of the region is selected based on Empirical Orthogonal Function analysis on the monthly precipitation from the observation (gridded station precipitation from U.Delaware v5.01; Willmott and Matsuura, 2001). The region shares overall a similar variability in the first EOF (13.28%) and second EOF (11.01%) (Fig. 1), in which this similarity also can be attributed to the influence of North Atlantic Oscillation in precipitation over the region (Dunkeloh and Jacobeit, 2003)"**

**Sect 2.3. lines 217 – 221)** "Lastly, in order to define droughts with pan-west-central Mediterranean characteristic, we select only drought events where more than 70% of the region of study is occupied by negative indices (Fig. 1). Though, for multi-year droughts, we allow that this condition does not need to be fulfilled for the initiation and termination years but only for the transition years: a drought can start weak and with a more local characteristic, then expand to a larger proportion of the Mediterranean, and weaken again in the termination years."

*Quantification of droughts events over the Mediterranean: Selection of a drought index.*

*9. A general issue investigating droughts over semi-arid regions like the European Mediterranean area with a pronounced annual cycle relates to the high (multi-annual) temporal and spatial variability of the availability of water resources. Therefore, drought or periods with scarcity of water are an intrinsic part of the climatic conditions over those regions. Likewise, this also applies for the opposite case with strong torrential rains leading to flooding and disastrous destructions over the respective areas. I think those points should be mentioned here or earlier in the introduction to put the drought terminology into context, underpinning that dry conditions are an integral part of the climate over those areas. Other, non climatic factors, for instance related to geology in terms of limestone with a high potential to effectively store water during winter and release it during summer could be mentioned. In addition, human impacts with steadily increasing demand for water resources play an important role interfering with the direct climatic driven changes in drought dynamics.*

**We agree with the referee that droughts are intrinsic part of the climate over the regions. The tree-ring-based reconstruction support this fact too, by showing a drought variability of multidecadal frequency over the western Mediterranean region during the last millennium (Cook et al. 2016a). We included this point and the corresponding literature in the introduction (Sect. 1. lines 73 - 78). Regarding torrential rainfall and flooding, as our focus is more on droughts with longer time scales (year to multiyear), we assume that short-lived events are already included as averages in drought calculations. Instead, the importance of small spatial scale (meso-scale) events associated to mid-latitude storm tracks on droughts was mentioned in introduction (Sect. 1, lines 124 – 127). The non-climatic factor, for example, related to anthropogenic influences was added in the conclusion (Sect. 4. lines 580 – 581).**

**Sect. 1. lines 73 - 78)** "Using the summer self-calibrated Palmer Drought Severity Index (scPDSI) based on tree ring reconstructions (also known as the Old World Drought Atlas; OWDA; Cook et al., 2015), Cook et al. (2016) showed that there have been several dry periods during the last 900 years over the region, some with persistent pan-Mediterranean characteristics. The region has experienced drought variability with frequencies of not only interannual, but multidecadal timescales."

**Sect. 1. lines 124 – 127)** "The precipitation of the region is strongly influenced by extratropical cyclones and in general, GCMs have difficulties in reproducing in full degree the dynamics and precipitation associated with these meso-scale phenomena (Raible et al, 2007; Watterson, 2006). Nevertheless, these atmospheric dynamics and precipitation are

**better represented in GCMs with finer spatial resolution (Champion et al., 2011; Watterson, 2006)."**

**Sect. 4. lines. 580 – 581) "[…] human impacts can modify the natural mechanisms and propagation of droughts (van Loon et al., 2016) increasing droughts risks and water shortage issues over the region."**

*Dynamics of multi-year droughts*

*10. I liked this part because it links the (regional) drought dynamics over the Mediterranean area with large scale modes of atmospheric (NAO) and atmosphere-ocean (ENSO) variability. However, especially in terms of ENSO I suggest to use a more objective test metric, because in my opinion the numbers are not really convincing for a robust inference which state of the ENSO precedes Mediterranean droughts. The authors should motivate their definition of a positive NAO / ENSO state that should considerable deviate from mean or neutral conditions. For instance, the threshold values of the SST anomaly over the tropical Pacific is set to ±0.5 K. Authors might use a metric based on percentiles of the according index-PDFs and investigate the situation separated into full period and drought prone years to test the robustness of the according conclusion. This could eventually also allow a quantitative differentiation in moderate/strong events for NAO and ENSO and their impacts on Euro-Mediterranean droughts.*

**We agree with the referee, thus, this part is updated because of the modified thresholds to discern positive/negative NAO and ENSO: The phases of NAO and ENSO are defined with respect to the non-drought periods: the values below 25 (above 75) percentile of NAO and ENSO during the non-droughts periods are considered as negative (positive) phases of NAO and ENSO respectively. We updated the texts (Sect. 3.4. lines 407 - 433) and respective plot (figure 9 in the manuscript or at the end of the responses) according to the modified values. As the text is long, we do not include here the modified paragraph.**

**We set the extreme positive NAO as the 95 percentiles and extreme negative ENSO as the 5 percentiles. We found that strong extreme positive NAO are more frequent during droughts, mainly during the initiation years. However, for extreme negative ENSO, the changes in frequencies in different stages of droughts are not observed (Sect. 3.4. lines 417 – 421).**

*Historical and Future conditions on droughts: 1850 to 2099 AD*

*11. The authors use a very strong GHG scenario – I wonder how results change in simulations with less pronounced increase in GHG. Moreover, how can changes in vegetation cover and/or human water consumption play into drought dynamics purely based on climatic considerations?*

**We mentioned briefly a complex interaction between plants and atmosphere (same as the response #6) and also the anthropogenic influences on droughts with the respective paper in the conclusion (Sect. 4. lines. 580 - 581, same as the response #9).**

**Regarding the different GHG scenarios, we provide the same response as the first response phase: we chose the RCP8.5 scenario as we could see more pronounced changes in the climate compared to the past condition. Additionally, this scenario is a part of the continuous run of this CESM simulation from 850 to 2099 AD (Lehner et al., 2015). Regarding the impacts of different GHG scenarios on the Mediterranean climate and droughts, among many others, Lehner et al. (2017) performed analysis to assess drought risks using the same model. They show that the drought risk over the Mediterranean increases in all GHG scenarios. We cited this paper in the introduction (Sect. 1.).**

*12. In this context it is again important to ask about the consequences if the main controlling factors (e.g. atmospheric circulation, Trigo et al., (1999)) are not realistically simulated. Are the models really able to realistically mimic the (change) of atmospheric circulation over the past and the following years? This is especially important, given the fact that Mediterranean precipitation is characterized by very short-lived and very intense precipitation events initiated by meso-scale circulation patterns (e.g. Genoa low) that might not be represented well enough in those models.*

**We addressed this issue regarding the atmospheric circulation associated with Mediterranean droughts in the response #7.**
**We observed that some differences exist between the observation and model, mainly in terms of mean scPDSI, and we attribute this difference to the model ability to perform dynamics and precipitation associated with meso-scale phenomena. We introduced this issue in the introduction (Sect. 1. lines 123 – 129, also see our response #3) and discussed in detail in the result section (Sect. 3.1. lines 259 – 261).**

**Sect. 3.1. lines 259 - 261) "One of the reason for the difference in the overall scPDSI between the model and observation is potentially related to the model performance on meso-scale phenomena, which play an important role for the regional precipitation during the wet season (Alpert et al., 1990; Ulbrich et al., 2009; Champion et al, 2007; Watterson, 2006)."**

*Conclusions*

*13. The conclusions are a good summary – what I think is important to add one or two chapters with more critical comments and insights on the limitations and uncertainties involved in the study (e.g. only single model used, validation of atmospheric circulation dynamics, importance of non-climatic events for drought dynamics), and also the implications in the context of model-proxy comparisons.*

**We included these points that the referee mentions in the conclusion (Sect. 4.):**

**- Limitations of the single model (lines 561 – 564): "[…] it is important to mention that our analysis is based on a single model output and this raises questions related to single model**

studies, such as boundary condition problems and model-dependent biases and physics (PAGES Hydro2k Consortium, 2017). Nevertheless, for a small confined area that surrounds a large body of water mass, the Mediterranean Sea, and the land coverage is limited, we found the necessity to use a simulation with a finer resolution to represent the regional climate better."

- Limitations of the model performance (lines 545 – 552): "The model inherent biases in representing ENSO and NAO (Bellenger et al., 2014; Fasullo et al., 2020) can have some implications in our results on the frequencies of ENSO and NAO at different stages of droughts. The model may produce too frequent and strong La Niña conditions and positive NAO during droughts due to its amplified, decadal for ENSO and seasonal for NAO, variability.  Moreover, due to the uncertainty associated with the changes in these modes in the future scenario, caution is required when interpreting the connection between droughts and modes of variability in the future warming scenario. Nonetheless, this problem does not affect our conclusion: the roles of ENSO and NAO become weaker with the longevity of droughts, while the regional circulation and feedback become more dominant at maintaining the persistence of droughts, which is also found in the future warming scenario."

- Anthropogenic influence on droughts (lines 580 – 581): "[…] human impacts can modify the natural mechanisms and propagation of droughts (van Loon et al., 2016) increasing droughts risks and water shortage issues over the region."

The validation, thus, the comparison among observation, proxy and model is also extensively discussed in the newly added validation section (Sect. 3.1).

Minor comments:

*14. Figure caption: If possible, please add below the technical description of the Figure a short sentence what are the main contents of the Figures for a better overview for the reader on the main conclusion of the respective plot(s).*

**We updated the captions.**

*15. Figure 1: Please include latitudes and longitudes in the figure – why is the eastern Mediterranean region not completely included into the analysis?*

**The respective figure is updated. Concerning the choice of the region, we addressed this issue in the response #8.**

**Figures**

**For the comment #10:**

[Figure]

**Fig 9. (a) Box plots of NAO and ENSO during multi-year Mediterranean droughts. The black crosses indicate 95 percentile value for NAO, and 5 percentile value for ENSO. Dashed red lines indicate the 25 and 75 percentiles of non-drought periods, which are the thresholds to discern negative/positive states of NAO and ENSO. Dashed brown lines indicate 5 and 95 percentile of non-drought periods. (b) Frequencies of occurrences of positive and negative states of NAO (left) and ENSO (right) for annual and winter total, and each stage of droughts. Black crosses indicate the frequencies of occurrence of extreme positive NAO and extreme negative ENSO.**

Manuscript cp-2020-79
**Response to the referee 2**

We would like to thank again the referees for their constructive feedbacks and insightful comments. We appreciate the time and effort the referee dedicated to review our manuscript, which helped us to improve our presentation. We have incorporated the suggestions made by them, and below you find our responses to the referees' comments (in blue).

**Major comments:**

*This is interesting work with valuable implications. However, in my opinion the authors have based their analysis and conclusions on the ability of the CESM to simulate pre-instrumental drought occurrence/frequency (and do not sufficiently prove that the CESM can do this) and draw several conclusions that appear to be based on visual comparisons of data in figures that I found hard to believe (and in some cases appeared simply incorrect) without further quantitative support.*

**We updated the figures in order to represent better our result and added the statistical tests where are necessary. We also included more details on the ability of CESM to simulate the present and past climates, and the limitations associated with the model.**

*Main concerns:*

*1. CESM simulation data are not easily accessible without contacting the researcher who ran the simulations. (As noted below, I wanted to try replicating the authors' analysis by comparing the CESM data to the OWDA data, but the CESM data are not publicly available.)*

**We provide some of the data used for our analysis (precipitation, temperature, geopotential heights) on anonymous FTP server:**
**https://fileserver.climate.unibe.ch/public/woonkim/**

*2. I wonder if perhaps some parts of the Mediterranean region experience drought at different times/magnitudes in the instrumental data (also in the CESM data)? Why did the authors choose this region?*
*Please provide more evidence that drought/precipitation in the region varies coherently*
*(e.g., suggest showing more information in Figure 1 other than a map and a box).*

**As we said in the first response phase, we elaborated better our motivation to focus on this specific averaged region in the introduction (Sect. 1.), and the reason for selecting the extent of the region in the model description and methods section (Sect. 2.)**

**Sect. 1. lines 116 - 121) "We choose this specific area, as this region has been affected by recent large scale droughts (Garcia et al., 2019; Spinoni et al., 2017), which can be seen as pan-west-central Mediterranean droughts. Moreover, the region shows coherent desiccation in the future scenario (Dubrovsky et al., 2014). From now on, for simplicity, we refer to the west-central Mediterranean region in our study simply as the Mediterranean**

region. We focus on understanding the dynamics that induce past persistent pan-regional multi-year droughts, and whether the dynamics that induce droughts in the past will change in the historical and future periods with the anthropogenic increase in GHG."

Sect. 2.2. lines. 153 – 157) "The focus area of the study is the western and central Mediterranean region confined to 15°- 28°E and 33°- 45°N (Fig. 01). The extent of the region is selected based on Empirical Orthogonal Function analysis on the monthly precipitation from the observation (gridded station precipitation from U.Delaware v5.01; Willmott and Matsuura, 2001). The region shares overall a similar variability in the first EOF (13.28%) and second EOF (11.01%) (Fig. 1), in which this similarity also can be attributed to the influence of North Atlantic Oscillation in precipitation over the region (Dunkeloh and Jacobeit, 2003)"

Sect 2.3. lines 217 – 221) "Lastly, in order to define droughts with pan-west-central Mediterranean characteristic, we select only drought events where more than 70% of the region of study is occupied by negative indices (Fig. 1). Though, for multi-year droughts, we allow that this condition does not need to be fulfilled for the initiation and termination years but only for the transition years: a drought can start weak and with a more local characteristic, then expand to a larger proportion of the Mediterranean, and weaken again in the termination years."

*3. The authors gloss over a critical comparison of the paleo to the model data (lines 179-182) - they conclude the background drought statistics (occurrence/frequency) in the CESM are similar to the OWDA. Yet, an examination of figure 1c suggests to me that the drought occurrence in the model and paleo data are quite dissimilar- the bulk of droughts in the CESM are centered around 6-10 years in length, and in the OWDA the distribution is centered around 1-4 year drought lengths. This discrepancy is quite striking to me, and I was surprised when the authors claim these distributions are comparable.*

As we responded in the first response phase, we agree with the referee that the absolute numbers of drought occurrence are dissimilar. We modified the respective texts and updated the plots in order to visualize better our results on the Sect. 3.2.

Sect 3.2. lines 299 - 306) "[…] the discrepancy between CESM and OWDA is clear, with OWDA presenting the mean duration of 5.38 years and CESM with 7.89 years. These means are statistically different to each other (p-value from Mann-Whitney U test of 0.003). This seems to be consistent with the result in the previous section (Sect. 3.1.) that shows that OWDA has droughts with shorter durations compared to the present-day observation and CESM over the Mediterranean region. Thus, this characteristic is still present during the entire last millennium. The variability of droughts over time in OWDA and CESM is also different to each other (Fig. 5.c), without a specific common period of increase in droughts. This gives us a first hint that the occurrence of droughts over the region are not mainly driven by the external natural forcings. Both time series present a common period of decrease in droughts around 1600 AD."

*4. If the authors want to make this claim, I suggest using some sort of metric (e.g., something like a Mann-Whitney or Wilcoxon rank-sum test or some sort of distribution comparison metric) to show these two drought occurrence distributions are statistically similar. Even a*

*report of the median, mean, and range would be more helpful than the visual comparison. I also suggest the authors use other metrics such as showing average drought occurrence per century (e.g., see Figure 3 in Parsons et al., 2018, J. Clim.).*

**We provided a new representation of the duration of droughts through a box plot that visualizes duration, number of droughts, means and medians of duration (see Figure 5 in the revised manuscript or at the end of the responses). We also performed Mann-Whitney U test between the summer CESM scPDSI and OWDA, which shows a p-value of 0.003. Also refer to our response #3.**

*5. Other suggestions include comparing the power spectra (PSD) of the OWDA and CESM PDSI. For example, I made Southern Mediterranean regional mean time series of PDSI from the OWDA and from the CESM1 LME run2 (this is an admittedly lower resolution version of CESM1; Otto-Bliesner et al., 2015; but the background drought statistics in the CESM LME and higher resolution versions of CESM are quite similar, at least in SW North America -e.g., Parsons and Coats, 2019, JGRA) over the 850- 1849 CE time period. I found the power spectra show quite dissimilar behavior for the CESM and OWDA PDSI variables, with varying discrepancies as varying frequencies depending on how I standardize them.*

**Refer to our response #3 and #4. We performed Mann Whitney U test and found a statistical dissimilarity between both time series and discussed about this point in the Sect. 3.2. in lines 299 - 306.**

*6. Comparison of CESM with instrumental/reanalysis data: the authors missed an opportunity to validate the performance of the CESM in the historical/instrumental era against instrumental/reanalysis data. The authors show (e.g., Figures 3,4,6,7) background geopotential height, SST, etc. anomaly patterns associated with drought, but they have not used instrumental-based data to show the model can accurately simulate the observed climate, and I remain unconvinced the background drought statistics are similar to the OWDA (see Main Concern (3) above).*

**As we said in the first response phase, we included a new section that compare the observation, OWDA and model: "Sect 3.1. Validation of CESM: comparison among observation, proxy and model (1901 – 2000 AD)."**

*7. Authors could compare patterns associated with drought (using a metric such as 2D pattern correlation) in the model to observed/reanalysis geopotential height (ERA5 or 20th Century Reanalysis) and SST (NOAA ERSSTv5, HadSST, etc.), as well as drought occurrence in the model to instrumental data (GPCCv2018 precipitation, Dai PDSI, CRU precipitation).Example of how other authors have made these comparisons among model and instrumental/ reanalysis data: Figure 2 in Parsons et al., 2018, J Clim., Figure 2 in Coats et al. 2013, GRL, Figure 2 in Stevenson et al., 2015, J Clim.)*

**The pattern correlation between the scPDSI and SST, and the scPDSI and geopotential height at 850 hPa in the CESM and reanalysis - observational data for the period of 1901-2000 are added in the validation section (Sect. 3.1.) in lines 269 – 278. As the text is long,**

**we do not include the paragraph here. You can find the corresponding correlation maps in the figure 4 in the revised manuscript or at the end of the responses.**

*8. The authors do not address several of the known shortcomings in the CESM model (e.g., frequency/strength of ENSO events; Parsons et al., 2017, J. Clim, Figure 6; Bellenger et al., Clim. Dyn, 2015 for a comparison of ENSO characteristics among models and instrumental data) and what the implications of these shortcomings could be for their study, especially because the authors make claims about likelihood of ENSO events before/during/after droughts. I suggest the authors consider the findings of Ault et al. (2014, J Clim), who show that the background power spectra/statistical characteristics of drought/precipitation (e.g., white noise, power law, etc.) are critical for drought magnitude and duration, and many CMIP5-class models do not show the same background variability as instrumental/ paleo data in many regions.*

**We included a discussion about the limitation of model regarding ENSO and NAO and possible implications of it in our result in the result section (Sect. 3.1.) and conclusion (Sect. 4).**

**Sect 3.1. lines 279 – 284) "It is known that the variability of ENSO is too strongly represented in CESM (Parsons et al., 2017; Stevenson et al., 2018). Nevertheless, the model is able to capture relatively well the hydroclimate condition associated with the ENSO teleconnection (Stevenson et al., 2018). In case of NAO, the seasonal variability of NAO seems to be amplified in many CMIP models (Fasullo et al., 2020). The CESM, however, resembles the present-day NAO pattern well and the spatial precipitation and temperature associated with this mode of variability in Europe (Deser et al., 2017). These inherent biases with respect to modes of variability (in particular for ENSO) can partially explain the differences we observe here between the model and observation."**

**Sect. 4. lines 545 – 552) "The model inherent biases in representing ENSO and NAO (Bellenger et al., 2014; Fasullo et al., 2020) can have some implications in our results on the frequencies of ENSO and NAO at different stages of droughts. The model may produce too frequent and strong La Niña conditions and positive NAO during droughts due to its amplified, decadal for ENSO and seasonal for NAO, variability. Moreover, due to the uncertainty associated with the changes in these modes in the future scenario, caution is required when interpreting the connection between droughts and modes of variability in the future warming scenario. Nonetheless, this problem does not affect our conclusion: the roles of ENSO and NAO become weaker with the longevity of droughts, while the regional circulation and feedback become more dominant at maintaining the persistence of droughts, which is also found in the future warming scenario."**

*9. Especially when future warming is considered, it is important to focus on metrics of drought that don't just focus on 'atmospheric centric' supply and demand, especially if ecosystem/water resource drought impacts are important. See Swann et al., 2016, PNAS, and Swann (2018) who note that drought severity/impacts in a warming climate can be grossly overestimated by use of variables/metrics such as PDSI.*

**We included a paragraph discussing about the problems associated with atmospheric centric drought indices in the result section (Sect. 3.2.) and conclusion (Sect. 4).**

**Sect. 3. lines 347 – 352) "Moreover, using the SOIL helps us to avoid the overestimation of drought risk and severity that occurs with many offline drought indices. The offline estimation of droughts by some common drought metrics, such as scPDSI, tends to magnify the impact of increase in temperature, therefore in potential evapotranspiration, on drought-associated atmosphere - surface feedback (Seneviratne et al., 2010). Hence, in the warming scenario, the indices that are constructed based on atmospheric supply and demand of moisture, such as scPDSI and SPEI, strongly overestimate future drought risks (Berg and Sheffield, 2018; Cook et al., 2018; Swann et al., 2016)."**

**Sect. 4. lines 573 – 579) "For a more comprehensive picture, droughts need be quantified with indices that can also reflect water stress on plants and ecosystem, and complex interactions among soil, atmosphere and vegetation. We used the upper 10 cm soil moisture anomaly to partially tackle this issue and derived spatial patterns associated with Mediterranean droughts based on this index. Still, the upper 10 cm soil moisture do not fully involve a complex interaction in the soil and atmosphere occurring during droughts. As vegetation is known to have more complex responses to the changing climate and droughts (Swann et al., 2016), the role of vegetation in extreme hydrological events can provide a more comprehensive view on drought mechanisms."**

*10. I appreciate that the authors included 10cm soil moisture, but given that surface soil water content can basically just follow precipitation variability in many regions, and thus, not really reflect full depth soil moisture trends (e.g., Berg et al., 2016), I think it would be helpful for the authors to show that they are analyzing variables actually relevant for plants/ecosystems/water resources in a warming climate, and not just supply/ demand from the atmosphere. At least a discussion of some of these points could really strengthen the paper.*

**We included a discussion about the drought index related issues and possible role of vegetation (and its interaction with soil and atmosphere) in the conclusion (Sect. 4) in lines 569 – 579. Also refer to our response #9.**

**"Lastly, we emphasize again the importance of assessing different drought indices in the paleoclimate context, also in the present and future warming scenario. Assessing different drought metrics is important for drought studies, as the most commonly used drought indices are based on water balance only from the atmospheric moisture supply and demand, and they tend to magnify drought risks in the future warming scenario (Berg and Sheffield, 2018; Mukherjee et al., 2018; Swann et al., 2016). For a more comprehensive picture, droughts need be quantified with indices that can also reflect water stress on plants and ecosystem, and complex interactions among soil, atmosphere and vegetation. We used the upper 10 cm soil moisture anomaly to partially tackle this issue and derived spatial patterns associated with Mediterranean droughts based on this index. Still, the upper 10 cm soil moisture do not fully involve a complex interaction in the soil and atmosphere occurring during droughts. As vegetation is known to have more complex responses to the changing climate and droughts (Swann, 2018; Swann et al., 2016), the role of vegetation in extreme hydrological events can provide a more comprehensive view on drought mechanisms."**

*Specific comments:*

*11. Lines 13-14: the authors just list one or two types of drought (meteorological), but what about hydrological, agricultural/ecosystem, socioeconomic types of drought?*

**As we responded in the first phase, here, we mostly used the drought metrics that reflect the meteorological (SPI), and agricultural (scPSDI, SPEI, SOIL) droughts. Though, it is important to mention that when the time scales of droughts become longer (like 1 year as our study), the differentiation among types of droughts becomes more difficult. We included more explanation on different types of droughts in the introduction of the revised manuscript.**

**Sect. 1. lines 18 – 24) "[…] it can be classified in four types: meteorological drought, associated with the decrease in precipitation; agricultural drought, associated with the depletion of soil moisture and impacts on crops and plants; hydrological drought, characterized by the depletion of streamflow and water reservoirs, and lastly socio-economic drought, that occurs when the other types of droughts cause impacts on society, in a way that the water supply cannot meet the demand from society (Mishra and Singh, 2010). If a drought lasts for a longer time period, meteorological drought is transformed to other kind of droughts, agricultural and/or hydrological, and different types of droughts become interconnected to each other (Wang et al., 2016; Zhu et al., 2019)."**

*12. Line 22: 'climate hot-spot'- please cite a paper that shows this*
*Line 23: 'increase in drought episodes' – again, please cite a paper supporting this*
*Lines 45-46: 'attributed to the increase in the atmospheric greenhouse gases (GHG) concentration, which causes . . . decrease in precipitation over the region' - citation?*
*Line 52: 'unprecedent intense drought projections' – citation?*

**We updated and corrected the citations.**

*13. Lines 63-64: The separation of ocean-atmosphere conditions during various drought stages has been done before- nice to acknowledge previous work (e.g., Parsons and Coats, 2019; Namias, 1960).*

**We included Parsons and Coats (2019) and Namias (1960) in the introduction (Sect. 1.), lines 85 – 91.**

*14. Lines 76-77: 'warm-dry temperature-hydroclimate co-variability at multidecadal timescales' confusing wording.*

**We reformulated the paragraph that includes the sentence to make the content of the paragraph clearer.**

**Lines 98 – 104) "[…] Ljungqvist et al. (2019) examined a long-term covariability between the summer temperature and hydroclimate during the Common Era. By comparing the instrumental records, tree ring-based reconstructions, and model simulations, they found**

that a warm-dry relationship with multi-decadal variability is more dominant in the southern Europe. Though, all datasets share some common leading modes of covariability across different time frequencies, there are some discrepancies among instrumental records, proxies, and model. The proxies present a stronger positive temperature-hydroclimate relationship, while the model exhibits a stronger negative relationship than the instrumental records."

*15. Line 92: 'high horizontal resolution' is subjective (and now closer to 'average resolution') in many CMIP6 models).*

**We removed "high" and replaced it by the resolution of the model (1.25° x 0.9°).**

*16. Line 102: Why not use the CESM LME (Otto-Bliesner et al., 2015)? There are more iterations, with several RCP8.5 extensions (and a much longer 1000 yr piControl run that is easier to compare w the last millennium runs given the similar length of simulations), allowing for a more complete analysis of internal variability. Is the background climate state that much better in the 1 degree vs the 2 degree version of the model? I ask because the authors explicitly state on lines 119-120 that they are interested in studying internal vs externally forced variability, and multi-model ensembles provide an ideal experimental framework for doing this.*

**As we said in the first response phase, we provided more explanations on the reason and benefit of using this single model in the introduction (Sect. 1.) and conclusion (Sect. 4.).**

**Sect. 1. lines 123 – 129) "The spatial resolution of the model is a clear advantage for our study on a relatively small confined area. The precipitation of the region is strongly influenced by extratropical cyclones and in general, GCMs have difficulties in reproducing in full degree the dynamics and precipitation associated with these meso-scale phenomena (Raible et al. 2007; Watterson, 2006). Nevertheless, these atmospheric dynamics and precipitation are better represented in GCMs with finer spatial resolution (Champion et al., 2011; Watterson, 2006). Hence, using a model that provides a seamless simulation for period 850 – 2099 AD guarantees an improved representation of precipitation related processes, thus, drought associated mechanisms over the region."**

**Sect. 4. lines 561 – 568) "Fifthly, it is important to mention that our analysis is based on a single model output and this raises questions related to single model studies, such as boundary condition problems and model-dependent biases and physics (PAGES Hydro2kConsortium, 2017). Nevertheless, for a small confined area that surrounds a large body of water mass, the Mediterranean Sea, and the land coverage is limited, we found the necessity to use a simulation with a finer resolution to represent the regional climate better. In the end, our study provides a useful understanding on the long-term variability and mechanisms of Mediterranean droughts by analyzing the entire last millennium. We addressed the influences of external and internal variability on Mediterranean droughts and distinctly different roles of the large-scale modes of variability and regional circulation during the different stages of multi-year droughts."**

**Additionally, as we mentioned in the first response phase, in order to assess more clearly the role of volcanic forcing on droughts, the wavelength coherence analysis is included in the result section (Sect. 3.2) in lines 331 - 339 and figure 6.**

*17. Line 107: the years 2001-2020 AD/CE are not the future*

**We modified the sentence as : "We use the Community Earth System model version 1.0.1 and its continuous transient simulation of 1250 years (850 - 2099 AD) where the 2005 – 2099 AD is run with the RCP 8.5 scenario, and control simulation of 400 years at perpetual 850 AD conditions (Lehner et al., 2015)" , lines 140 – 142.**

*18. Lines 103-112: Suggest just citing Lehner et al. for the model description*

**We removed the sentences about each component model, and just cited Lehner et al. (2015), in line 142.**

*19. Line 127: As in Main Concern (2), please show the region varies coherently in instrumental/paleo and the version of CESM used here*

**We addressed this issue in the response #6.**

*20. Lines 131-132: removing a linear trend over the 1850-2099 time period looks quite problematic to me (e.g., Figure 9)- removing a linear trend over this time period will add in non-climatic variability artifacts from the trend removal. It looks to my eye like there is a trend 1900-2000, then a separate trend 2000-2099.*

**We applied the detrending method to two time periods separately:  the 1850-2000 and the 2001-2099. We mentioned these steps in the model description and method section (Sect 2.2), but we reformulated the respective paragraph (lines 170 – 174) to clarify the procedure.**

**Sect. 2.2, lines 170 – 174) "For the second period (1850 - 2099 AD), the anomalies are linearly detrended in order to examine the background mechanisms associated with dryness during this period without the anthropogenic influence on climate. This is performed by splitting the time period into two and applying the least squares method to the each of the period separately: from 1850 to 2000 and from 2001 to 2099 AD.  Then, the detrended anomalies are compared against the non-detrended anomalies for the same period and also for the first period."**

*21. Line 149-150: linear temperature trend is removed, but then authors study the impacts of warming using this drought metric, which includes temperature. . .so have the authors removed temperature changes, then try to study the impacts of warming on drought? This reasoning doesn't make sense to me. Perhaps a more clear explanation. of trend removal would help (?).*

**Here, we aim to see whether during this period, there is a natural change in the mechanisms of droughts or the changes in droughts are due to the anthropogenic influences.**

**We reformulated the paragraph about the detrending procedure in the Sect. 2.2. (refer to our response #20) and also rearranged the sentences about this result in Sect. 3.5. in order to clarify our observation.**

**Sect. 3. 5. lines 505 – 509) "This result shows that the natural mechanisms associated with droughts remain the same as it is in the past period, thus, no natural changes on drought mechanisms occur for the period of 1850 – 2099 AD. This means that the intensification of Mediterranean droughts is clearly due to the anthropogenic influences: in the future scenario, the intensity of atmosphere - soil feedback is magnified, due to the increase in GP and TS, and this mechanism becomes the dominant one at controlling the desiccation over the region."**

*22. Lines 140-155: As in Main Concern (6): I think all of these drought metrics/variables, with the exception of upper 10cm soil moisture, do NOT reflect actual impact on plants/ecosystems in a warming climate. Also, upper soil water content can diverge from deeper soil water – authors should show that this is a useful metric here that is distinct from precipitation alone if they want to argue that their study has relevance for ecosystem impacts.*

**We addressed this issue in the responses #9 and #10.**

*23. Lines 161-164: This drought counting method appears similar to Herweijer et al. 2007; Coats et al. 2013b- did the authors come up with this metric, or can they use a similar metric to previously published work (if so, please cite) to maintain consistency across the literature?*

**Here, we provided same response as the first response phase: the drought counting method we used differs from the counting methods from the literature mentioned here including the one by Herweijer et al (2007). In our work, one drought cluster has to have only negative or zero anomalies with at least one year that the drought index falls below its 10 percentiles of distribution, and any wet year would stop the continuity of drought. By defining a drought cluster in this way, we make sure we only take strong events, also, assuring that a dry condition persists throughout the entire year. We explained our counting method in the 2.3 section of our manuscript.**
**We have already checked that the other counting methods (for example, the method by Coats et al. (2013) that a drought starts with two continuous years of negative anomalies and stops with two continuous positive anomalies) are not appropriate for our region of study, where slight dry conditions are more frequent.**

*24. Lines 168-170: see above note about similar methods in Parsons and Coats as well as Namias.*

**Refer to our response #13.**

*25. Lines 179-183: As in Main Concern (3): Please be more quantitative. To my eye, these distributions do not appear similar- the OWDA shows droughts that are mostly 1-4 yrs, and the CESM shows droughts centered around 8 yrs. Please use a more quantitative method to compare drought time series power spectrum and/or drought frequency in paleo and model data.*

**We modified the plots and performed some statistical tests. Refer to our response #3 and #4.**

26. Lines 187-188: difficult to visually compare these different drought metrics in lower panels in Figure 2 because the x axis limits are different.

**We modified the respective plot and texts. Refer to our response #3.**

27. Lines 204-205: 'no noticeable changes in occurrence of droughts' - is this to the eye? Can you use a more quantitative method to show this (e.g., running counts of droughts in 50 yr windows or something like that)?

**We agree with the referee that the text is misleading. We changed the text as "at first glance, similarly to the summertime OWDA and CESM, no noticeable coherent changes among all indices, such as a common period of increase in the occurrence of droughts over time, is noticed." in lines 329 – 331.**

**Also, as we said in the first response phase, we changed the plot to show the occurrence and duration of droughts to 100-yr running mean of duration of Mediterranean droughts (figure 5 in the manuscript or at the end of the responses), which can better show the point we mention.**

28. Lines 205-206: 'not driven by external forcing': again, this conclusion appears to be drawn based on a visual comparison, which seems insufficient to me. Lehner et al. (2015, ESD, Figure 5) use running correlation to compare model output, which I imagine could be applied here, as could some sort of wavelet/coherence analysis between volcanic forcing time series and the OWDA and CESM data. Also, Superposed Epoch Analysis or Composite Analysis could be used with volcanic forcing time series/large eruptions. At minimum, it would be great to see a time series showing the external forcing to be able to compare to the drought time series in Figure 2.

**The duration plots are updated (figure 5 in the manuscript or at the end of the responses), and we believe that this change makes it easier to visually compare the drought indices. Additionally, a wavelet analysis is performed to address more clearly the role of volcanic forcing on droughts (Sect. 3.2,lines 333 - 341 and figure 6). Also refer to the last paragraph of our response #16.**

29. Line 209-210: sentence wording is confusing/complicated.

**We changed the sentence as: "In the next sections, we investigate the underlying dynamics using only the SOIL as the drought indicator in order to understand the role of the internal variability in Mediterranean droughts.", lines 342 – 343.**

30. Lines 211-215: So if the r value is 0.78, doesn't this imply that only 60% of variance is shared by the two time series?

**We provided the same response as the first response phase: yes, in which we think it is a good percentage of variance shared by two time series.**

*31. Lines 218-220: 'control simulation presents 29 droughts'- this comparison with the transient simulation is non-sensical/misleading given the two simulation lengths are different. Can the authors instead present the average numbers of droughts of various lengths per century (e.g., Parsons et al., 2018; Coats et al., 2015, Figure 5). This gets around the issue of having different length time series and gives more meaningful information about drought risk standardized to a given time window (e.g., number of droughts per 100 or 500 years).*

**We changed these numbers by mentioning the number of events/100 years, lines 362 – 363.**

*32. 222-224: Is this the first time these patterns have been presented? Seems that a paper like Markonis et al. 2018 (Nature Communications) or other similar papers have previously presented similar patterns associated with hydroclimatic variability.*

**This is the same response as the first response phase: we mentioned in our manuscript that more similar patterns are found in Xoplaki et al.(2003), and we discussed about the similarity in detail in lines 381 - 390.**

*33. Lines 229-236: Similar to the point I raise in Main Concern (5)- It is well documented that this model simulates ENSO events that are too strong and too frequent (e.g., Bellenger et al., Clim. Dyn., among others)- how does that impact these results? For example, if the model simulates too strong, too regular ENSO events that unrealistically influence global climate, then is it surprising that a signal from ENSO is apparent in European drought/climate? And is this finding meaningful if it's based on model bias?*

**Refer to our response #8. About the response of ENSO in European climate has been addressed in some literature before (Mariotti et al., 2002; 2008; Brönnimann, 2007; Brönnimann et al., 2007), which we mentioned in the introduction of our manuscript.**

*34. Figure 3 caption: the caption states 'means are not statistically significant'- unclear. Please be more specific. Also please clarify if data are annual, JJA, etc. in figure caption. Additionally, the significance dots are nearly impossible to see on the dark red/blue background*

**We updated the captions and the plots.**

*35. Lines 246-250: Are these % changes in drought/rainfall meaningful (e.g., for agriculture, ecosystems), or do these changes fall well within normal climate variations that don't have a large impact?*

**We included sentences explaining about these rates.**

**Sect. 3.3. lines 392 – 395) "These changes in precipitation are in the range of the rates of expected decrease in annual precipitation over the Mediterranean region in the future scenario. In the future, the regional precipitation is expected to reduce by 5 – 30 % from its present-day value and this will cause water shortage related issues in the region (Dubrovsky et al., 2014, Mariotti et al., 2008)."**

*36. Also, is the background variability (e.g., standard deviation, mean) of rainfall in the CESM realistic, or can we chalk this up to model bias?*

**To address this point, we included the comparison of mean seasonal spatial and annual cycle of precipitation between the observation and model in the validation section (Sect. 3.1., lines 239 – 244). Also refer to our response #6.**

*37. Lines 254-255: similar to Main Concern (4), what about in 20th century reanalysis, ERA5, or some similar reanalysis product vs GPCCv2018 or CRU precip? Or Dai PDSI?*

**Refer to our responses #6 and #7.**

*38. Lines 257-260: 'The starting point is. . .to one or both of them'- confusing wording*

**We changed the sentence as: "In the following, we investigate the origin and the evolution of Mediterranean long droughts associated to NAO, ENSO-like condition and drought high using the transient simulation up to 1849 AD." in lines 404 – 406.**

*39. Lines 262-269: So in other words, there is about equal odds of being in a drought during various NAO or ENSO phases? This seems important because the authors claim on lines 294-295 that a certain combination of NAO and ENSO conditions are important for initiating drought. . .but it appears to me as though there are nearly equal odds of this happening (60%) based on the phase of NAO/ENSO. Is this interpretation incorrect?*

**This part is updated because we modified the thresholds to discern positive/negative NAO and ENSO: The phases of NAO and ENSO are defined with respect to the non-drought periods: the values below 25 (above 75) percentile of NAO and ENSO during the non-droughts periods are considered as negative (positive) phases of NAO and ENSO respectively. We updated the texts (Sect. 3.4. lines 407 - 433) and respective plot (figure 9 in the manuscript or at the end of the responses) according to the modified values. As the text is long, we do not include here the modified paragraph. Note that the occurrence of NAO and ENSO in different stages of droughts are more distinguishable.**

*40. Lines 298-310: I don't see how Fig 8 proves the point. Basically, it looks to me as though drought starts off dry and then transitions to less dry conditions at end of drought, and this is distinct from wet years.*

**Here, we provide the same response as the first response phase: we agree that droughts start off dry and become less dry with time. Here, we want to emphasize the importance of development of anticyclonic center associated with the high geopotential height anomaly which would be the driver that maintains the dry condition over the region for long time while the roles of the large-scale circulation patterns are decreasing from the transition to the last stage of droughts.**

**We reframed the related paragraph in the Sect. 3.4. in order to clarify this question.**

**Sect. 3.4. lines 443 - 447) "The mean circulation in each stage shows that the development of the high pressure system, namely the drought high in the Fig. 7, takes place in the transition years. This indicates that some mechanisms associated with this circulation is possibly important at determining the longevity of droughts after the initiation years. A possible candidate of an important process for the transition stage of Mediterranean droughts is the interaction among regional atmospheric and soil variables initiated by the anticyclonic circulation system over the region."**

**Also, we found that the figures 8 and 9 in the previous manuscript were quite repetitive to show our main point (that the drought high is developed in the transition years), thus, we only provide one figure for the mean circulation in different stages of droughts without separating NAO and ENSO (figure 10 in the revised manuscript).**

*41. Lines 325-327: Similar to Main Concern (4); I have not been shown how the model performs compared to instrumental/reanalysis for the relevant variables over Europe/Mediterranean, so these conclusions don't mean a lot to me.*

**Refer to our responses #6 and #7.**

*42. Lines 337-340: 1) I see no major changes in distribution of drought in Figure 10 - are these distributions distinct? Please see previous comments related to statistically distinguishing distributions (and not visually distinguishing), especially when they appear to overlap. 2) Any future changes in ENSO in this model should be interpreted with caution as most CMIP5 models, including this one as far as I can remember, struggle to reproduce the observed trends in the tropical Pacific (see Coats and Karnauskas, 2017, GRL as well as Seager et al., 2019, Nature Climate Change).*

**1) We performed Mann Whitney U-test and the corresponding paragraph is updated accordingly:**

**Sect. 3.5. lines 498 – 501) "For the detrended variables, the means of GP and SOIL during droughts show that they are statistically indifferent to the 850-1849 AD values (p-value of 0.09 for GP and 0.44 for SOIL). The same is also true for the NAO and ENSO during droughts**

(p-values of 0.19 and 0.29 for each). The detrended TS over the region is statistically similar from the 850-1849 AD value but only at 2% confidence level (p-value of 0.02)."

2) We included a brief discussion on the uncertainties associated with the ENSO and NAO in models and possible implication in our result in the conclusion (Sect. 4), lines 543 – 550. Also refer to our response #8.

Sect. 4. lines 545 – 552) "The model inherent biases in representing ENSO and NAO (Bellenger et al., 2014; Fasullo et al., 2020) can have some implications in our results on the frequencies of ENSO and NAO at different stages of droughts. The model may produce too frequent and strong La Niña conditions and positive NAO during droughts due to its amplified, decadal for ENSO and seasonal for NAO, variability. Moreover, due to the uncertainty associated with the changes in these modes in the future scenario, caution is required when interpreting the connection between droughts and modes of variability in the future warming scenario."

*43. Lines 344-345: As figure 9 shows, trends in the region are not linear 1850-2100, so trend removal is problematic over this time period. Perhaps it makes sense to remove the trend 2000-2099, but otherwise the authors could be adding an artifact of trend removal into the analysis.*

We addressed this issue in our response #20.

*44. Lines 358-359: 'our analysis shows that the overall similarities': as stated above, the authors never actually showed this statistically, just a visual comparison.*

We updated the texts according to the added analysis. Refer to our response #3.

*45. Line 383: 'climate over the region to a drier climate have started earlier than reported in the modern observational era': to back up a statement like this, I'd again like to see that the model is simulating instrumentally observed climate during the relevant temporal overlap in the historical run (e.g., show Mediterranean precip./PDSI time series in model and instrumental data) before claiming that any drying has happened earlier than reported.*

We modified the sentence to "This means that the intensification of droughts and the shift of the mean climate over the region to a drier climate have already started since the pre-industrial era.", lines 554 – 555.

**Figures**

**For the comments #4, #27 and #28:**

[Figure]

**Fig 5. (a) Distribution of duration of droughts for the summer scPDSIs in OWDA and CESM, and for (b) the annual drought indices in CESM during 850 – 1849 AD. Red points indicate individual drought events, black lines on the boxes are the medians and blue crosses are the means of duration. (c) 100-years running means of duration of droughts for the summer scPDSIs and (d) the annual drought indices. The indices are the scPDSI from Old World Drought Atlas (OWDA), summer scPDSI from CESM (CESM-scPDSI), annual Standardized Precipitation Index (SPI), Standardized Precipitation Evapotranspiration Index (SPEI), soil moisture anomaly (SOIL), and annual scPDSI. Note that the annual scPDSI (brown line in (d)) has a separate y-axis for its duration. The p-value from Mann-Whitney U-test between the duration of summer scPDSI in OWDA and CESM is 0.003, which indicates the means are statistically different. For the annual indices, the means among each other are also statistically different, except between the SPEI and SOIL (p-value of 0.87).**

**For the comment #7:**

[Figure]

**Fig 4. Maps of Pearson correlation between the scPDSI and anomalies of (a) Sea Surface Temperature from ERSST v5 and (b) Geopotential height at 850 hPa from the CR21 for the observation (left) and CESM (right) during the period of 1901-2000 AD. The linear trends of variables are removed before applying the correlation. Black dots on the maps show the regions where correlations are statistically not significant at 5% confidence level.**

**For the comment #39:**

[Figure]

**Fig 9. (a) Box plots of NAO and ENSO during multi-year Mediterranean droughts. The black crosses indicate 95 percentile value for NAO, and 5 percentile value for ENSO. Dashed red lines indicate the 25 and 75 percentiles of non-drought periods, which are the thresholds**

to discern negative/positive states of NAO and ENSO. Dashed brown lines indicate 5 and 95 percentile of non-drought periods. (b) Frequencies of occurrences of positive and negative states of NAO (left) and ENSO (right) for annual and winter total, and each stage of droughts. Black crosses indicate the frequencies of occurrence of extreme positive NAO and extreme negative ENSO.

**Dynamics of the Mediterranean droughts from 850 to 2099 AD in the Community Earth System Model**

Woon Mi Kim[1,2] and Christoph C. Raible[1,2]

[1]Climate and Environmental Physics, University of Bern, Switzerland
[2]Oeschger Centre for Climate Change Research, University of Bern, Switzerland

**Correspondence:** Woon Mi Kim (woonmi.kim@climate.unibe.ch)

**Abstract.**

In this study, we analyze the dynamics of multi-year long droughts over the western and central Mediterranean region for the period of 850 - 2099 AD using the Community Earth System model version 1.0.1. Our study indicates that Mediterranean droughts during the period of 850 - 1849 AD are mainly driven by the internal variability of the climate system. A barotropic high pressure system together with a positive temperature anomaly over central Europe and the Mediterranean region is the prominent pattern that occurs in all seasons with droughts. Also, the modes of variability, i.e. the North Atlantic Oscillation (NAO) and El Niño Southern Oscillation (ENSO), are associated with Mediterranean multi-year droughts, showing that droughts occur more frequently with positive NAO and La Niña-like conditions. These modes of variability play a more dominant role during the initial stage of droughts. However, their role diminishes with the evolution of droughts, and after the initial stage, the persistence of multi-year droughts is determined by the interaction between the regional atmospheric and soil moisture variables. This atmosphere – soil interaction becomes stronger during the 1850 – 2099 AD period, reducing the importance of modes of variability on droughts and inducing a constant dryness over the Mediterranean region. Additionally, the discrepancy among diverse drought metrics in representing duration and frequencies of past droughts is presented, re-affirming the necessity of assessing a variety of drought indices even in the paleoclimate context.

**1 Introduction**

Drought is an extreme weather and climate event characterized by a prolonged period with persistent depletion of atmospheric moisture and surface water balance from its mean average condition. Drought is also characterized by a slow onset and devastating impacts on society, the economy and the environment (Wilhite, 1993; Dai, 2011; Mishra and Singh, 2010), and it can be classified in four types: meteorological drought, associated with the decrease in precipitation; agricultural drought, associated with the depletion of soil moisture and impacts on crops and plants; hydrological drought, characterized by the depletion of streamflow and water reservoirs, and lastly socio-economic drought, that occurs when the other types of droughts cause impacts on society, in a way that the water supply cannot meet the demand from society (Mishra and Singh, 2010). If a drought lasts for a longer time period, meteorological drought is transformed to other kind of droughts, agricultural and/or hydrological, and different types of droughts become interconnected to each other (
[revised manuscript text omitted]
 background mechanisms associated with dryness during this period without the anthropogenic influence on climate. This is performed by splitting the time period into two and applying the least squares method to the each of the period separately: from 1850 to 2000 and from 2001 to 2099 AD. Then, the detrended anomalies are compared against the non-detrended anomalies for the same period and also for the first period.

Composites of positive and negative phases of two modes of variability are also investigated: the NAO and ENSO. The NAO is taken as the difference in the sea level pressure anomalies between the regions confined to 33° - 21°W / 35° - 39°N and 25° - 13°W / 63° - 67°N, which reflects the Azores high and the Iceland low respectively (Wallace and Gutzler, 1981; Trigo et al., 2002). The ENSO is characterized by the annual mean sea surface temperature anomalies over Niño 3.4 region in the Tropical Equatorial Pacific (170° - 120°W and 5°S - 5°N) (Trenberth, 1997).

We further perform wavelet coherence analysis (Grinsted et al., 2004; Gouhier et al., 2018) in order to find a possible time-varying association between droughts and volcanic eruptions, using the time series of volcanic eruptions (Gao et al., 2008) and of drought indices (Sect. 2.3). The time series are normalized to have a zero mean and one standard deviation.

**2.3 Drought definitions**

[revised manuscript text omitted]

In terms of the droughts (Fig. 3.(a)), the means from the summer scPDSI between the model and OWDA are statistically similar to each other (p-value from the t-student test of 0.28). This is also the case between the OWDA and observation but with much lower confidence level of 1% (p value of 0.01). However, both summer and annual scPDSI from the model are statistically different to those from the observation (p-values of 0.001). In terms of the number of droughts (Fig. 3.(b)), the observation presents 4, OWDA 7 and CESM 3 events during the last century. For the duration of droughts, the mean (5.43 years) and median (3 years) from OWDA are different to those from the observation (11.50 and 12.50 years respectively), with OWDA exhibiting lower values. CESM also presents a lower median in the duration of droughts (6 years) compared to the observation, though, its mean (9.67 years) resembles better the observation than the one from OWDA. The discrepancies in the means and medians of droughts between the observation and CESM are still present in annual droughts.

We observe that the model tends to underestimate the duration of present-day droughts than those from the observation. However, the model still shows to a certain extent its ability to reproduce persistent long droughts, simulating droughts of few years long, and with longer duration than those from OWDA. Still, the analysis on annual-scale extreme dry events based on a relatively short present period of 100 years is not sufficient to draw comprehensive conclusions of present-day multi-years droughts, due to the limited number of events.

One of the reason for the difference in the overall scPDSI between the model and observation is potentially related to the model performance on meso-scale phenomena, which play an important role for the regional precipitation during the wet

season (Alpert et al., 1990; Ulbrich et al., 2009; Champion et al., 2011; Watterson, 2006). Additionally, the model performance on internal variability may contribute to this discrepancy, which will be discussed in the following paragraphs. The same can also explain the case of OWDA and observation, that shows a p-value at the limit of 1% confidence level. The annually resolved OWDA is based on tree rings, which are known to be biased towards the growing season, so that the full annual signal might be not fully preserved in such a reconstruction. Moreover, proxy model observation comparison studies show that tree ring-based reconstructions tend to deviate in their spectral behavior (Franke et al., 2013), and the distribution of tree rings used to generate the gridded reconstruction over Mediterranean (Cook et al., 2015) may not be enough to capture precipitation events associated with regional-scale cyclones and to fully explain dry/wet variability for the entire region (Babst et al., 2018).

After all, the model and observation share many common patterns associated with the scPDSI. The ocean and atmospheric condition associated with the variability of scPDSI in the observation and model are presented in Fig. (4) through the correlation patterns between the scPDSI and SST, and the scPDSI and geopotential height at 850hPa. The observation and model exhibit significant positive correlations over the central Equatorial Pacific, though in the observation, the region with statistically significant correlation is located more on the north-central Pacific than in the model. The observation and model share a significant wave-like pattern over the extratropical latitudinal belt, that extends from the North Pacific to Siberia. Within this wave-like pattern, a bipolar pattern with a significant negative correlation centered over Mediterranean region and a positive correlation over the northern high latitudes are prominent. In the observation, the area of negative correlation over Europe is larger and the positive correlation of the bipolar pattern is shifted more to the Scandinavian region than in the model. The observation and model present some common patterns occurring over the regions of ENSO and NAO.

It is known that the variability of ENSO is too strongly represented in CESM (Parsons et al., 2017; Stevenson et al., 2018). Nevertheless, the model is able to capture relatively well the hydroclimate condition associated with the ENSO teleconnection (Stevenson et al., 2018). In case of NAO, the seasonal variability of NAO seems to be amplified in many CMIP models (Fasullo et al., 2020). The CESM, however, resembles the present-day NAO pattern well and the spatial precipitation and temperature associated with this mode of variability in Europe (Deser et al., 2017). These inherent biases with respect to modes of variability (in particular for ENSO) can partially explain the differences we observe here between the model and observation.

Overall, the model is able to reproduce the climate condition associated with the variability of present-day scPDSI, despite the fact presents some discrepancy to the observation exist. Moreover, the model does not significantly underestimate the persistence of multi-year droughts. As it also shows statistical similarity to OWDA over the region, we consider that the model can be used for the analysis on past Mediterranean droughts. We take into account these differences of model to the observation and biases on modes of variability to discuss the implications of them in our results more in detail in the Sect. 4.

**3.2 Variability of Mediterranean droughts during the 850 - 1849 AD and their connection to the volcanic forcing**

To gain an overview of drought conditions in the Mediterranean, we assess the indices defined in the Sect. 2.3 using the period 850 to 1849 AD and focusing on the drought events and their duration. In the beginning, we compare the variability and duration of droughts of the summertime scPDSI from CESM with those from OWDA. We do not aim to make a direct comparison between the proxies and the model simulation, as this cannot be made due to the different initial conditions

295  between the proxies and model (PAGES Hydro2k Consortium, 2017; Xoplaki et al., 2018). Here, we rather focus on comparing the simulated summer drought variability with the one of the OWDA. Additionally, we assess whether the simulated and reconstructed droughts respond similarly to the same external forcing, i.e. the volcanic eruptions.

The Fig. 5.(a) and (c) exhibits the distribution and the 100 years running mean of duration of summer (June-July-August mean) in OWDA and CESM. In terms of duration of summer droughts (Fig. 5.(a)), the discrepancy between CESM and OWDA

300  is clear, with OWDA presenting the mean duration of 5.38 years and CESM with 7.89 years. These means are statistically different to each other. This seems to be consistent with the result in the previous section (Sect. 3.1) that shows that OWDA has droughts with shorter duration compared to the present-day observation and CESM over the Mediterranean region. Thus, this characteristic is still present during the entire last millennium. The variability of droughts over time in OWDA and CESM is also different to each other (Fig. 5.(c)), without a specific common period of increase in droughts. This gives us a first hint

305  that the occurrence of droughts over the region are not mainly driven by the external natural forcings. Both time series present a common period of decrease in droughts around 1600 AD.

Similarly, the Fig. 5.(b) and (d) show the distributions and running means of duration of annual Mediterranean droughts in CESM quantified using different indices. As expected, different indices do not exactly behave similarly in terms of the occurrence, number of events, and duration (Raible et al., 2017; Mukherjee et al., 2018). However, the indices coincide for

310  some years: in total 89 years of the 850 - 1849 AD period, all indices indicate the same overlapped drought periods. In terms of duration, the scPDSI is the one which shows more longer lasting droughts than other indices, with a mean duration of 9.1 years. Then, the SPEI, SOIL and SPI follow it with the mean durations of 2.9, 2.8, and 2.3 years, respectively. The SPI presents more events than other indices, but with shorter duration. All these means are statistically different among each other, except the means between SPEI and SOIL, which are statistically indifferent (p-value of 0.87).

315  The difference in duration of droughts among indices can be attributed to the water balance variables involved in the computation of each index. For instance, the SPI only takes precipitation as its input variable. Thus, it does not consider the atmospheric evaporative demands, which can be intensified during dry periods. Therefore, we expect that the SPI shows a reduced duration of droughts compared to the scPDSI and SPEI, which include the potential evapotranspiration in their water balance. The same holds true for the SOIL index. Though, the soil moisture in the model is closely connected to the hydrological cycle

320  reflecting the balance between the precipitation and actual evapotranspiration, the magnitude of actual evapotranspiration over the region is smaller than the potential evapotranspiration derived from the Thornthwaite method. Hence, the water balance involved in SOIL is affected in such a way that the drought duration is reduced compared to the scPDSI and SPEI. Lastly, droughts with relatively longer duration in the scPDSI can be explained by the memory effect embedded in the calculation scheme of scPDSI (Palmer, 1965; Wells et al., 2004), which other indices that are obtained by being normalized with respect to

325  certain statistical distribution families do not present. The scPDSI is an accumulating index; therefore, during the calculation process, the weighted value of preceding months is used to estimate the index for the current month, implying a persistence of the events. Hence, with the scPDSI, an intense yearly drought would likely induce a drought in the following year and this effect can be exacerbated in the context of intense multi-year droughts.

In terms of variability over the period of 850 - 1849 AD (Fig. 5.(d)), at first glance, similarly to the summertime OWDA and CESM, no noticeable coherent changes among all indices, such as a common period of increase in the occurrence of droughts over time, is noticed. Although each index captures slightly different aspects of water balance, we expect to see a similar common response to external forcing among all the indices, if droughts are influenced by the same externally forced variability, e.g., the volcanic forcing. However, this is not the case. The wavelet coherence analysis between the drought indices and volcanic eruptions corroborates the absence of a connection between these two variables (Fig. 6). The signals of statistically significant coherent variability between drought indices (SOIL and scPDSI) in CESM and volcanic eruptions are not uniform across the period-year frequency bands. Before 1100 AD, the anti-phase relationship between the two variables dominates, while after 1100 AD is the opposite, an in-phase relationship is more visible. Moreover, the leading variables of association also change over the time and frequency band. This is the same for OWDA. Besides, OWDA does not show strong significant signal during the 1257 Samala eruption, which was the strongest eruption in the last millennium (Gao et al., 2008). Hence, the analysis confirms that the occurrence of yearly and multi-year Mediterranean droughts are not driven by the volcanic eruptions, rather, the driver can be attributed to the internal variability.

In the next sections, we investigate the underlying dynamics using only the SOIL as the drought indicator in order to understand the role of the internal variability in Mediterranean droughts. The focus on one indicator is motivated by the fact that soil moisture reflects the regional hydrological balance associated with the precipitation and evapotranspiration, and it is also an indicative of water stress on plants and ecosystem (Berg et al., 2017; Swann, 2018). Another advantage of this index is that the variable is a direct output from the model, thus, it does not require any further step, except for calculating the anomalies, and statistical assumptions as other indices do. Moreover, using the SOIL helps us to avoid the overestimation of drought risk and severity that occurs with many offline drought indices. The offline estimation of droughts by some common drought metrics, such as scPDSI, tends to magnify the impact of increase in temperature, therefore in potential evapotranspiration, on drought-associated atmosphere - surface feedback (Seneviratne et al., 2010). Hence, in the warming scenario, the indices that are constructed based on atmospheric supply and demand of moisture, such as scPDSI and SPEI, strongly overestimate future drought risks (Berg et al., 2017; Cook et al., 2018; Swann et al., 2016).

[revised manuscript text omitted]
 by comparing the non-detrended and detrended drought related variables. The detrending process is performed following the steps mentioned in the Sect. 2.2. In this way, we exclude the trends caused by the anthropogenic changes during these 250 years to observe the background climate during droughts.

480 Previously, it is shown that past Mediterranean droughts are associated with the intense positive geopotential height and temperature anomalies over central Europe and the Mediterranean. These features are also observed during droughts in the non-detrended 1850 - 2099 AD condition, but with more intense positive geopotential height (GP) and temperature (TS) anomalies than during the period 850 - 1849 AD (Fig. 13.(a)). The variances of GP, TS and SOIL are also enlarged compared to the past; therefore, their medians and extreme tails are also magnified, which imply that the dryness and its associated

485 atmospheric conditions become more frequent and severe in 1850 - 2099 AD. The increases in GP and TS clearly intensify the above mentioned interaction among regional atmospheric and soil variables, i.e the positive temperature – soil moisture feedback. This intensification aids the longevity and intensity of droughts, which is also reflected as a reduction in the surface soil moisture anomaly (Fig. 9). Additionally, the precipitation - soil moisture interaction is involved, i.e. a continuous reduction in precipitation decreases soil moisture, thus, inducing less evapotranspiration, which leads again to a reduction in precipitation

490 (Seneviratne et al., 2010).

Related to the modes of variability, the frequencies of positive NAO and La Niña-like conditions during droughts also seem to be affected by the overall change in global temperature (Fig. 13.(b)). Compared to 850 – 1849 AD, the non-detrended 1850 – 2099 AD period shows a slight reduction in the preference toward the positive NAO and La Niña-like conditions during droughts. This result is in line with the previously mentioned regional circulation associated with droughts: in this situation

495 where the regional atmospheric variables have a more dominant role in the regional desiccation aided by the intense GP and

TS, the importance of modes of variability is reduced, even during the initial stages of droughts. Hence, the role of positive NAO and La Niña-like through different stages of droughts is diminished.

For the detrended variables, the means of GP and SOIL during droughts show that they are statistically indifferent to the 850-1849 AD values (p-value of 0.09 for GP and 0.44 for SOIL). The same is also true for the NAO and ENSO during droughts (p-values of 0.19 and 0.29 for each). The detrended TS over the region is statistically similar from the 850-1849 AD value but only at 2% confidence level (p-value of 0.02). This indicates that the detrending method is not enough to fully exclude the strong effects of anthropogenic changes on temperature in future droughts. Nevertheless, the mean spatial composites of the detrended surface temperature and geopotential height at 850 hPa during Mediterranean droughts (Fig. 14) support the indifference between these two periods over the large portion of the Mediterranean region, exhibiting the circulation patterns which are similar to those during the 850 - 1849 AD period (Fig. 7). This result shows that the natural mechanisms associated with droughts remain the same as it is in the past period, thus, no natural changes on drought mechanisms occur for the period of 1850 – 2099 AD. This means that the intensification of Mediterranean droughts is clearly due to the anthropogenic influences: in the future scenario, the intensity of atmosphere - soil feedback is magnified, due to the increase in GP and TS, and this mechanism becomes the dominant one at controlling the desiccation over the region.

**4    Conclusions**

We have investigated the variability and mechanisms of multi-years droughts over the western and central Mediterranean region with pan-regional characteristic for the period of 850 - 1849 AD and whether these mechanisms associated with Mediterranean droughts have changed after the pre-industrial period from 1850 to 2099 AD with the anthropogenic increase in GHG. We have performed our analysis by using CESM simulations.

[revised manuscript text omitted]

coverage is limited, we found the necessity to use a simulation with a finer resolution to represent the regional climate better.

565 In the end, our study provides a useful understanding on the long-term variability and mechanisms of Mediterranean droughts by analyzing the entire last millennium. We addressed the influences of external and internal variability on Mediterranean droughts and distinctly different roles of the large-scale modes of variability and regional circulation during the different stages of multi-year droughts.

Lastly, we emphasize again the importance of assessing different drought indices in the paleoclimate context, but also in
570 the present and future warming scenario. Assessing different drought metrics is important for drought studies, as the most commonly used drought indices are based on water balance only from the atmospheric moisture supply and demand, and they tend to magnify drought risks in the future warming scenario (Berg et al., 2017; Mukherjee et al., 2018; Swann et al., 2016). For a more comprehensive picture, droughts need be quantified with indices that can also reflect water stress on plants and ecosystem, and complex interactions among soil, atmosphere and vegetation. We used the upper 10 cm soil moisture anomaly
575 to partially tackle this issue and derived spatial patterns associated with Mediterranean droughts based on this index. Still, the upper 10 cm soil moisture does not fully involve a complex interaction in the soil and atmosphere occurring during droughts. As vegetation is known to have more complex responses to the changing climate and droughts (Swann, 2018; Swann et al., 2016), the role of vegetation in extreme hydrological events can provide a more comprehensive view on drought mechanisms.

The Mediterranean region is considered as one of the most vulnerable regions under the current climate change scenario
580 (e.g. Giorgi and Lionello, 2008; Lehner et al., 2017) and human impacts can modify the natural mechanisms and propagation of droughts (Van Loon et al., 2016), increasing droughts risks and water shortage issues over the region. Hence, more studies on the topics related to droughts and permanent future desiccation in the Mediterranean region including the role of vegetation are necessary to develop a better preparedness for upcoming changes.

[revised manuscript text omitted]
 0.28. The p-value between the OWDA and observation is 0.01, and between the CESM and Observation for the summer and annual, both p-values are 0.001. (b) Distribution of duration of annual-scale droughts in different datasets. Red points on each box shows the data points, thus number of droughts, black lines are the medians and crosses are the mean values of duration.

[Figure]

**Figure 4.** Maps of Pearson correlation between the scPDSI and anomalies of (a) Sea Surface Temperature from ERSST v5 and (b) Geopotential height at 850 hPa from the CR21 for the observation (left) and CESM (right) during the period of 1901-2000 AD. The linear trends of variables are removed before applying the correlation. Black dots on the maps show the regions where correlations are statistically not significant at 5% confidence level.

[Figure]

**Figure 5.** (a) Distribution of duration of droughts for the summer scPDSIs in OWDA and CESM, and for (b) the annual drought indices in CESM during 850 – 1849 AD. Red points indicate individual drought events, black lines on the boxes are the medians and blue crosses are the means of duration. (c) 100-years running means of duration of droughts for the summer scPDSIs and (d) the annual drought indices. The indices are the scPDSI from Old World Drought Atlas (OWDA), summer scPDSI from CESM (CESM-scPDSI), annual Standardized Precipitation Index (SPI), Standardized Precipitation Evapotranspiration Index (SPEI), soil moisture anomaly (SOIL), and annual scPDSI. Note that the annual scPDSI (brown line in (d)) has a separate y-axis for its duration. The p-value from Mann-Whitney U-test between the duration of summer scPDSI in OWDA and CESM is 0.003, which indicates the means are statistically different. For the annual indices, the means are also statistically different among each other, except between the SPEI and SOIL (p-value of 0.87).

[Figure]

**Figure 6.** Wavelength coherence between drought indices (OWDA, summer and annual CESM scPDSI, and SOIL) and volcanic eruptions for 850 – 1849 AD. The red shaded regions indicate where the coherence of two time series are statistically significant at 5% confidence level. The directions of arrow provide information on the association between two variables, whether they are in-phase (to the right) or anti-phase (to the left), the first variable (right-down or left-up) or the second variable (right-up or left-down) plays the causal role.

[Figure]

**Figure 7.** (a) Mean geopotential height anomaly at 850 hpa, and (b) mean surface temperature anomaly, for the control (left) and transient (right) simulations during Mediterranean droughts in 850 – 1849 AD. Black dots on the composites of the transient simulation indicate the regions where the means between the control and transient simulations are statistically not significant at 5% confidence level.

[Figure]

**Figure 8.** Mean geopotential height anomaly at 850 hpa (color shaded) and surface temperature anomaly (contours every $0.2C^\circ$, positive in red and negative in blue) during Mediterranean droughts for each season in the transient simulation.

[Figure]

**Figure 9.** (a) Box plots of NAO and ENSO during multi-year Mediterranean droughts. The black crosses indicate 95 percentile value for NAO, and 5 percentile value for ENSO. Dashed red lines indicate the 25 and 75 percentiles of non-drought periods, which are the thresholds to discern negative/positive states of NAO and ENSO. Dashed brown lines indicate 5 and 95 percentile of non-drought periods. (b) Frequencies of occurrences of positive and negative states of NAO (left) and ENSO (right) for annual and winter total, and each stage of droughts. Black crosses indicate the frequencies of occurrence of extreme positive NAO and extreme negative ENSO.

[Figure]

**Figure 10.** Evolution of atmospheric conditions in each stages of droughts. Anomalies of (above) specific humidity, and (below) temperature, both at 925 hPa during initiation, transition and termination years. Arrows indicate winds at 925 hPa.

[Figure]

**Figure 11.** Same boxplot as Fig. 9 but for standardized regional atmospheric and soil variables over the region of study during Mediterranean droughts: anomalies of geopotential height at 850 hPa (GP), surface temperature (TS), soil moisture (SOIL), sensible heat flux (SH), latent heat flux (LH) and evapotranspiration (EV). (b) Frequencies of occurrences of positive and negative anomalies in each stage of droughts in order: initiation, transient and termination years.

[Figure]

**Figure 12.** Time series of annual soil moisture (SOIL), and precipitation anomalies from 1850 to 2099 AD with respect to the 1000 - 1849 AD means. Brown lines indicate smoothed 10 years running mean, and dashed lines the detrended time series.

[Figure]

**Figure 13.** (a) Standardized regional variables: anomalies of geopotential height at 850 hPa (GP), surface temperature (TS) and soil moisture (SOIL) over the region of study, and (b) indices of large scale circulation patterns: NAO and ENSO during Mediterranean droughts for the period of 850 - 1849 AD (green), non-detrended 1850 - 2099 AD (red) and detrended 1850 - 2099 AD (yellow). The GP, TS and SOIL between the detrended 1850 – 2099 AD and the 850 – 1849 AD periods present p-values from Mann-Whitney U test of 0.09, 0.02 and 0.29, respectively. For NAO and ENSO, the p-values are 0.19 and 0.29 for each.

[Figure]

**Figure 14.** Detrended mean geopotential height anomaly at 850 hpa and surface temperature anomaly during Mediterranean droughts for the 1850 - 2099 AD. Black dots indicate the regions where the means between the detrended 1850 – 2099 AD and 850 – 1849 AD are statistically not significant at 5% confidence level (Fig. 7).

---

## Author Response (AR2)

Dr. Hugues Goosse, editor
Climate of the Past

Woon Mi Kim
Physics Institute and Oeschger Centre for Climate Change Research, University of Bern
Sidlerstrasse 5, 3012 Bern, Switzerland

16th February 2021

Dear editor Dr. Hugues Goosse,

We would like to thank you again for another opportunity to resubmit our manuscript and also to the reviewers for their detailed and insightful comments. We greatly appreciate the time that the reviewers dedicated to our manuscript. We have addressed all the reviewers' comments carefully and please find below our responses to each of the reviewers.

Sincerely,

Woon Mi Kim[*] and Christoph C. Raible

[*]Corresponding author: woonmi.kim@climate.unibe.ch

Minor comments

Just some minor comments on wavelet analysis: I would have suggested superimposed epoch analysis, focusing on the largest volcanic eruptions of the last millennium. If authors use wavelet, it might be helpful to give some more background information how to interpret the highly temporally irregularly distributed volcanic forcing time series in conjunction with droughts in a wavelet concept. For instance, the very large eruptions in 1258 (Samalas), 1452 (Kuwae), 1600 (Huaynaptina) and 1815 (Tambora) are reflected as clear peaks in the decadal bands for the CESM drought Indices vs. volcanic eruption wavelet coherence figures. The peak prior to 1100 AD might just be an artefact, since no large volcanic eruption in the period 850 – 1200 AD are evidenced in the reconstructed volcanic time series of GAO et al (2008) that was used as volcanic forcing for CESM. An argument for the physical implausibility of an inversion of the lead-lag droughts vs. volcanic eruptions relationship (of the pre-1100 AD peaks) is that droughts typically do not lead to large tropical eruptions.

In addition, the null hypothesis for the test is not clear to me (i.e. are the confidence intervals based on white or red noise?). At least the authors should mention those issues in an additional paragraph in the methods and respective discussion section 3.2.

**Thanks for your feedback and comments. As the referee mentioned, we noted the problem with applying the wavelet coherence analysis to the time series of discrete events and we are aware of it. The process of filtering frequencies of low variability in the time series of eruptions can enlarge the effects of pre- and post-eruptions, therefore, adding some non-physical artefacts in the filtered time series.**

**In the revised manuscript, we still included the wavelet coherence analysis in Fig 6., as the analysis is useful to show the effects of large and small eruptions on drought indices. We now mentioned the problem associated with the wavelet coherence analysis with the time series of eruptions in lines 360 - 365, then, as suggested by the reviewer, we performed a superposed epoch analysis on the drought indices and included it in Fig. 7. The superposed epoch analysis supports a connection between the drought indices and large eruptions indicating that eruptions cause wet conditions rather than dry conditions over the Mediterranean region.**

**The discussion on these new results is presented in lines 350 – 373.**

**Lines 350 – 373: 'To further assess a potential connection between the drought indices and volcanic eruptions, a wavelet coherence analysis is applied (Fig. 6). The analysis shows that significant co-variability between the simulated drought indices (SOIL and scPDSI) and eruptions are found during periods with strong and frequent volcanic eruptions, for instance, around 1257 Samala and 1600 Huaynaputina eruptions. For small eruptions, the signals of co-variability are not uniform among the eruptions, some showing significant co-variability while other not. In addition, the phase relationships between the eruptions and the drought indices also vary among the eruptions (not shown). This non-uniformity of co-variability between the small eruptions and the drought indices is a strong indication that no physical connection between both exists, i.e., the significant co-variability is merely due to statistical artefacts. OWDA does not show a strong significant co-variability during the 1257 Samala eruption, which was the strongest eruption in the last millennium**

(Gao et al., 2008). Still, OWDA shows a significant co-variability during the period of Little Ice Age around 1400 - 1600 AD, similarly to CESM.

The wavelet coherence analysis is clearly useful to distinguish the effects of strong and small eruptions on the variability of drought indices. However, the analysis poses some problems in handling discontinuous time series with a sporadic occurrence of the events, such as the volcanic eruptions. Filtering certain frequency bands from this kind of discontinuous time series smears out the eruption (i.e., an eruption starts earlier and last longer than in reality), therefore, adding some non-physical artefacts in the time series. Some of these are also reflected in Fig. 6 showing that the significant co-variability occur earlier that the actual eruption years.

Hence, for a more detailed analysis on causal effects of volcanic eruptions on drought variability, the superposed epoch analysis is applied to the 16 largest eruptions and the 16 smallest eruptions (Fig. 7; for the list of the eruption years, see table A2). The analysis shows that the increases in drought indices are followed after large eruptions, and this positive association lasts up to 3 years in CESM. On the other hand, no significant response of drought indices to small eruptions is noted. This finding is in line with Rao et al. (2017) and Mcconnell et al. (2020) that demonstrated wetter conditions in the Mediterranean region after strong eruptions.

Thus, the analysis shows that large eruptions are associated with an increase in drought indices, i.e., wet periods. In other words, the occurrence of yearly and multi-year Mediterranean droughts are not driven by the volcanic eruptions, but the internal variability.'

Response to the Referee 2

(1) The comparison of CESM with instrumental/reanalysis data: there is much improvement here (e.g., comparison of geopotential height, SST anomalies in simulations and instrumental/reanalysis), but the authors still don't show that the model is simulating the timing and magnitude of trends in local rainfall and SOIL/PDSI (e.g., Figure 12 – if the drying trend is in fact a forced signal that is as strong as the CESM appears to indicate it is, it should show up in the instrumental data)- given that the conclusions about ongoing/future forced changes hinge on the model's ability to do this, I suggest addition of instrumental vs CESM time series data in Fig 12 or a panel c in Figure 3 that shows time series from the model and observations on top of each other (I know the paper/figures are already quite long, so I don't want the authors to have to add extra figures).

**We included the time series of summer scPDSI from the observation, CESM, and OWDA in Fig. 3.(c), also indicating the values of their trends on the upper left in the same figure.**

**We added a sentence about these trends in:**

**Lines 254 - 257: 'Nevertheless, all three scPDSI show negative trends during 1901-2000 AD (Fig. 3.(c)), also in each sub-period (1901 - 1950 and 1951 - 2000 AD), indicating a continuous increase of drying over the region, which is in line with the previous studies (Mariotti et al. (2008); Sousa et al. (2011); Spinoni et al. (2015)). Based on the Mann-Kendall tests, these trends are all statistically significant at 95% confidence level.'**

(2 - a) Potentially flawed use of statistical tests: the authors claim to use a Mann-Whitney test to determine if the 'means' of the OWDA and CESM PDSI variables are distinguishable and find 'the means are statistically similar to one another' - this is a non-sensical test as far as I can tell. First, it looks like the two PDSI time series are normalized to similar time periods (the last millennium), so they should both be centered about a long-term mean of zero, so testing if two time series' means are statistically different if they are centered at zero is not a test that the model is doing well- the real test seems to me to be in Figure 5a (are drought distributions similar).

**It is true that the time series of drought indices (scPDSI, SPI, SPEI) are normalized with respect to their long-term means. However, note that what we have compared here are the time series of duration of drought events, and not the entire time series which contain wet/dry fluctuations.**

**Statistically, drought episodes are located at the lower tails of the probabilistic distributions of drought indices. Hence, the time series with only droughts are clearly not centered at a long-term mean of zero and they do not necessarily follow a normal distribution. Thus, for these time series, we can use a non-parametric Mann-Whitney U test.**

(2 - b) Finally, the Mann-Whitney test should be applied to test the similarity of distributions of data, not means (as far as I know), but the authors' wording suggests they are using this test to determine if means of distributions are significantly different throughout the manuscript.

**We admit that we used a wrong word 'means' to describe what is tested in a Mann-Whitney U test (M-W). As far as we know, a M-W test can be interpreted as an analog to the conventional t-test for location parameters (means or medians) of two samples with unknown probabilistic distributions (Wilks, 2001). The null hypothesis of M-W test states equal distributions of two samples, and statistically equal distributions in a non-parametric test imply equal location parameters between two samples. However, as the assumption in a non-parametric test is that the location parameters of samples are still unknown, we agree that here we cannot state explicitly that the means or medians of two time series are compared to each other. Therefore, as the referee points out, we corrected the term 'means' to 'distributions' where the M-W tests are involved in our manuscript and mentioned what are tested in M-W in the method section (Sect. 2.2).**

**Lines 174 – 175: 'A Mann-Whitney U significance test is performed to statistically compare the distributions between the transient and control simulations.'**

*Wilks, Daniel S. Statistical methods in the atmospheric sciences. Vol. 100. Academic press, 2011.*

(3) The reported finding that Mediterranean droughts are mainly driven by internal variability in the climate system: the wavelet coherence figure seems to demonstrate that Mediterranean PDSI shows significant coherence with the volcanic forcing time series after large eruptions, so the timing of variability doesn't look like it's only due to internal variability. Perhaps there are wet periods following volcanic eruptions, but there is not enough information in the figure caption to determine the phase of the relationship. Additionally, I would think the authors would want to show that the magnitude/duration of droughts in the control run overlap with the magnitude/duration in the forced run (and statistically test if these drought duration distributions are distinguishable), but I don't see a figure or analysis that shows this (e.g., additional box/whisker plot w CESM control drought durations in Fig. 5a)- the authors do present the mean number of droughts and durations in the control and forced runs in Section 3.3, but showing the actual drought distributions and testing this difference could be much more informative and actually test/support the authors' assertion that drought (duration) is driven by internal variability.

**It is true that the wavelet coherence analysis shows some co-variability between the drought indices and eruptions, mostly after some large eruptions. However, we noted that the wavelet coherence analysis cannot fully demonstrate in detail the associations between these two variables as the usage of a discrete time series like the volcanic forcing initiates some problems. The process of filtering the frequencies of low variability in the time series of eruptions seems to enlarge the effects of pre- and post-eruptions, therefore, adding some non-physical artefacts in the filtered time series. This problem was also pointed out by the referee 1.**

**To show this point, we applied the low pass filters to the summer scPDSI and volcanic eruptions for some periods where the covariability between these two timeseries are significant in the wavelet**

coherence map (Fig 1 below). The filtered time series of eruptions shows that the peak of the eruptions is smeared out, so that the eruptions occur much earlier than their actual years.

[Figure]

*Fig 1. Low-pass filtered time series of volcanic eruptions (brown) and summer scPDSI (navy blue). The filtered time series are scaled with respect to their maximum values and standardized to make easier the visual comparison between two time series. Dots indicate the actual eruptions.*

This fact is reflected in the wavelet coherence plot (Fig. 6 before the revision) that exhibits some significant signals of covariability already some years before the peaks of the eruptions. This issue also can explain why sometimes scPDSI leads the causal association with the volcanic eruptions.

However, we still found the wavelet coherence analysis useful to distinguish the effects of large and small eruptions on the variability of drought indices. Most significant covariability are found during the periods of large eruptions, while for small eruptions, the significant signals are low or random among eruptions.

Hence, we decided to still include the wavelet coherence analysis in Fig 6. but without showing the phase relationship to concentrate on discerning the effects of large and small eruptions on drought

indices. In addition, we mentioned the problem related to the wavelet coherence analysis with this specific time series in lines 360 - 365. For more robust analysis to see the association between the drought indices and volcanic eruptions, we performed the superposed epoch analysis on the drought indices. It is found that the superposed epoch analysis (Fig. 7) supports the positive association between the drought indices and large eruptions indicating that the eruptions cause wet conditions but not dry conditions. We included the plot on this analysis in Fig 7, and modified the discussion on this part in lines 350 – 373.

Also, as suggested, we included the duration of droughts from the control simulation in Fig 5. (a) and (b), and applied the M-W test to these variables between the transient and control simulations (p-values in table A1 in the appendix). Here, we noted that the variability of duration of droughts in the transient simulation is in the range of the variability found in the control simulation. The text about this result is added in lines 343 - 345.

Lines 350 - 374: 'To further assess a potential connection between the drought indices and volcanic eruptions, a wavelet coherence analysis is applied (Fig. 6). The analysis shows that significant co-variability between the simulated drought indices (SOIL and scPDSI) and eruptions are found during periods with strong and frequent volcanic eruptions, for instance, around 1257 Samala and 1600 Huaynaputina eruptions. For small eruptions, the signals of co-variability are not uniform among the eruptions, some showing significant co-variability while other not. In addition, the phase relationships between the eruptions and the drought indices also vary among the eruptions (not shown). This non-uniformity of co-variability between the small eruptions and the drought indices is a strong indication that no physical connection between both exists, i.e., the significant co-variability is merely due to statistical artefacts. OWDA does not show a strong significant co-variability during the 1257 Samala eruption, which was the strongest eruption in the last millennium (Gao et al., 2008). Still, OWDA shows a significant co-variability during the period of Little Ice Age around 1400 - 1600 AD, similarly to CESM.

The wavelet coherence analysis is clearly useful to distinguish the effects of strong and small eruptions on the variability of drought indices. However, the analysis poses some problems in handling discontinuous time series with a sporadic occurrence of the events, such as the volcanic eruptions. Filtering certain frequency bands from this kind of discontinuous time series smears out the eruption (i.e., an eruption starts earlier and last longer than in reality), therefore, adding some non-physical artefacts in the time series. Some of these are also reflected in Fig. 6 showing that the significant co-variability occur earlier that the actual eruption years.

Hence, for a more detailed analysis on causal effects of volcanic eruptions on drought variability, the superposed epoch analysis is applied to the 16 largest eruptions and the 16 smallest eruptions (Fig. 7; for the list of the eruption years, see table A2). The analysis shows that the increases in drought indices are followed after large eruptions, and this positive association lasts up to 3 years in CESM. On the other hand, no significant response of drought indices to small eruptions is noted. This finding is in line with Rao et al. (2017) and Mcconnell et al. (2020) that demonstrated wetter conditions in the Mediterranean region after strong eruptions.

**Thus, the analysis shows that large eruptions are associated with an increase in drought indices, i.e., wet periods. In other words, the occurrence of yearly and multi-year Mediterranean droughts are not driven by the volcanic eruptions, but the internal variability.'**

**Lines 343 - 345: 'Importantly, for the same indices, the distributions of duration of droughts are statistically similar to the distributions in the control simulation at 99% confidence interval (Fig. 5. (a) and (b) and. Table A1). This implies that the variability of droughts in the transient simulation is within the range given by the internal variability in the control simulation.'**

(4) The drought initiation/termination is quite interesting, but I don't know what to make of the indices (NAO/ENSO) after they have been redefined to percentiles during non-drought periods- is this standard practice? And how is this information meaningful for interpreting year to year NAO/ENSO as it relates to drought predictability/trajectories? For example, I am unsure as to what the 'extreme positive' NAO now means- how much is this index now changed if it is redefined only during non-drought years?

**The motivation for this part was that the positive NAO and negative ENSO also occur frequently during the non-drought period as one can observe in Fig. 10. (a). Then our question was, how the NAO and ENSO during droughts differ in terms of their occurrences and magnitudes from those during non-droughts. For this, it was necessary to set a reference period to be a "drought-free" period without any perturbation introduced by droughts to discern clearly what happens during droughts with respect to the non-drought periods. Thus, defining the thresholds based on the non-drought period as a reference facilitates the comparison between these two periods.**

**Regarding the predictability/trajectories, we do not think that using these relative thresholds would lead to a different conclusion about the evolution of NAO/ENSO in different stages of droughts from using the absolute thresholds. The reason is simply that the values of the threshold for positive NAO (0.58) and negative ENSO (-0.56) (see the table 1 below) do not differ much from the absolute thresholds (0.5 and -0.5). Thus, the conclusion that the role of the large-scale circulation patterns becomes weaker through the stages of droughts remains unchanged.**

**The extreme NAO and ENSO are defined relative to the non-drought period with the same reason as mentioned in the first paragraph. Again, the 5th and 95th percentiles of each mode during the non-drought period are not too different from those during the total period (Table 1). These 5th and 95th extreme values differ much from those during the drought period, and this supports our approach that the condition during droughts can be characterized better by comparing with respect to the "drought-free" condition.**

**Regarding our approach of taking a specific period relative to another as a reference is rather common, e.g., the superposed epoch analysis to extract volcanic forcing imprints is such an example. This estimates the mean departure of climate conditions during volcanic events relative to other years.**

|  | NAO | | | ENSO | | |
|---|---|---|---|---|---|---|
|  | total | Non-drought | Drought | total | Non-drought | Drought |
| 5th | -1.69 | -1.74 | -1.22 | -1.59 | -1.53 | -1.72 |
| 25th | -0.64 | -0.69 | -0.40 | -0.66 | -0.56 | -0.89 |
| 75th | 0.69 | 0.58 | 0.98 | 0.56 | 0.61 | 0.44 |
| 95th | 1.55 | 1.43 | 1.76 | 1.77 | 1.84 | 1.40 |

*Table 1. Percentiles of the distributions of NAO and ENSO*

**In the revised manuscript, we clarified that the thresholds we use for the analysis are relative to the non-drought period, including the extreme thresholds:**

**Lines 433 – 438: 'The extreme NAO and ENSO are also defined in a similar way by taking the values below 5th percentile for negative extremes and above 5th percentile for positive extremes. Defining thresholds relative to the non-drought period facilitates the comparison between the two periods, showing whether and how these modes of variability during droughts differ from those during the opposite hydroclimate conditions. For simplicity, we call these relative negative and positive phases simply as positive and negative NAO and ENSO without referring constantly that they are defined relative to the non-drought period.'**

(5) I still find many sentences/phrases hard to decipher, as they are either grammatically incorrect or just hard to interpret- I have noted in line numbers where this is most apparent.

**We went through the manuscript and tried to correct ambiguous sentences and also modified the parts and sentences noted by the referee.**

Specific comments in sections/figures/by lines:

(6) Abstract: I see the authors have responded to my previous comment about model limitations and added to the main text, but again, I am left with no sense in the Abstract about any model limitations/bias – given that many readers may only look at figures and the Abstract, it would be informative to insert a phrase/sentence about model strengths and limitations. If the list of model shortcomings as it relates to simulation of drought over the Mediterranean is so long that it can't be easily summarized in the abstract, that's a problem I think.

**We included some sentences on the model limitation in the last part of the abstract.**

**Lines 2 – 8: 'Overall, the model is able to realistically represent droughts over this region, although it shows some biases in representing El Niño Southern Oscillation (ENSO) variability and mesoscale phenomena that are relevant in the context of droughts over the region.'**

**Also note that we added some more information in the abstract.**

(7) Lines 3-4 'Our study indicates…mainly driven by internal variability' – as I mentioned in the general comments, after seeing the wavelet coherence plots, I think this statement is flawed and needs to be qualified- perhaps the overall duration/severity of droughts is statistically indistinguishable in the control and forced simulations (which isn't shown as far as I can determine), but PDSI variability and large volcanic eruptions appear to vary coherently at interannual-decadal timescales. To really be able to make the claim that the duration/severity of drought is purely internal, the authors need to actually compare the duration/severity of control run droughts and forced simulation droughts, but I don't see this comparison anywhere. As it stands, it looks like the CESM shows a sensitivity to volcanic eruptions that the OWDA doesn't show.

**We addressed this issue in the response (3).**

(8) Lines 20-26: I appreciate that the authors acknowledged there are various types of drought, but I disagree with the idea that a meteorological drought that is long enough just becomes the other types of droughts. For example, couldn't an area become warmer (and receive the same amount of precipitation, experience the same drought frequency), but experience earlier spring melt, more evaporation, and thus hydrologic or agricultural drought? This ag drought would not just be caused be a persistent meteorological drought.

**We agree that there are other causes of agricultural and hydrological droughts and they are not always caused by meteorological droughts. Here we wanted to state a connection among different types of droughts we mentioned, mainly when a drought is initiated by a deficit of precipitation, thus a meteorological drought. Therefore, we modified the sentence to:**

**Lines 30 - 34: 'If a meteorological drought lasts for a longer period, it has the potential to propagate to other types of droughts, such as agricultural or hydrological drought. In this sense, different types of droughts can become connected to each other. Thus, meteorological drought is one of the causes of other types of droughts, among other processes such as seasonal changes of run-off or an increase in evapotranspiration demand.'**

(9) Lines 34-35: 'The climate of the Mediterranean is characterized as semi-arid with a pronounced annual cycle, thus, high temporal and spatial variability of the availability of water resources'- does a semi-arid climate imply spatial variability? Or temporal variability? I thought it just had to do with the mean climatologic conditions (winter wet, summer dry on average), but I could be wrong.

**As the referee said, the temporal variability is related to a pronounced annual cycle. We corrected the sentence as:**

**Lines 43 - 44: 'The climate of the Mediterranean is characterized as semi-arid with a pronounced annual cycle, which means a high temporal variability of the availability of water resources.'**

(10) Lines 38-39: 'The western and eastern regions show different precipitation regimes' this makes it sound like the region should be split into two separate regimes for the analysis.

**We focus on the western-central region. The eastern region with a different precipitation regime is not included in the analysis. We mentioned our region of study and validated the choice of the region in the introduction lines 120 - 125 and in the method section (Sect. 2.2) lines 161 - 165.**

(11) Line 44/49: authors define EA-WR and use it, but then ER-WR is used on line 77 (?)

**Thanks for the point. That was a typo and we corrected it to Eastern Atlantic pattern (EA).**

(12) Line 51: suggest 'response of the Mediterranean climate to ENSO'

**We corrected the word as suggested.**

(13) Line 71: 'multi-years long desiccation' is a bit of an odd phrase- suggest 'multi-year drought' or multi-year dry periods'.

**We changed the word to 'multi-year dry periods'.**

(14) Lines 112-113: I understand that other cited studies have shown that parts of Europe may be drying using different data sources, but it would be helpful to show the actual time series in instrumental and CESM data earlier in the manuscript, both to illustrate this visually for the reader and to validate the model's trends/drought sensitivity to warming.

**We addressed this issue in the response (1).**

(15) Line 132-133: suggest 'validate the model simulation used here' in place of 'our model'

**We modified the sentence to 'validate the model simulation'.**

(16) Line 166-167: 'The statistical tests to compare the transient to the control simulations are performed with the Mann-Whitney U significance test for the means at a 5% confidence level' – the authors go on to use this test in the text/figure captions to show that the 'means are statistically different'- similar to my general comment above, doesn't this test ask if the distributions are likely different?

**We addressed this issue in (2-b) and modified the text accordingly.**

(17) Lines 168-169: I'm a bit confused about which years are used and why 5 sets of 89 years are chosen for the transient simulation- the control run is 400 years, so the total number of years doesn't match in the random draw of the transient vs the control. Also, are the draws of years contiguous (e.g., years 1-89 in a row) or randomly drawn? If they are randomly drawn, what is the sense in using different sets of draws? Please clarify because this approach doesn't make immediate sense to me.

**We picked 5 sets of random 89 years from the 225 drought years in the transient simulation (not in the control simulation). We performed this step to see whether the length of the simulation, therefore the years with droughts, is what forces the transient simulation to have the similar geopotential and temperature patterns to those in the control simulation during droughts (in other words, whether performing a mean over more drought years smooths some possible external signals involved during droughts). If droughts in both simulations are affected by the same drivers, then when we pick some random years with droughts from the transient simulation, we would expect to see similar circulation patterns in both transient and control simulations. In the end, the result shows that regardless the length of the simulation, the circulation patterns during droughts in both simulations are statistically similar. However, with Section 3.2 and after the response (3) to compare the duration of droughts between the transient and control simulation in the manuscript, we noted that this is a redundant analysis to affirm the same conclusion as Section 3.2. Thus, in the revised manuscript, we excluded the sentence about this step.**

(18) Lines 171-174: After my last comment in the previous round of reviews, I appreciate the authors tried to split up the time periods from which they remove trends, but I still am not sure why linear trends are removed over these time periods as the figure showing the time series isn't shown until Figure 12 - it may help to show the reader these time series when discussing the time series and trend removal.

**We included the figure of time series of soil moisture we used with the magnitudes of trends for each period (the 1850 – 2000 and the 2001 – 2099) in the appendix Fig. A1. and mentioned about this figure this in the line 178.**

(19) Line 184: 'some drought metrics'- suggest 'various'? Or state exactly how many- 'some' sounds strange

**We changed this to 'We use four drought metrics [...]'.**

(20) Lines 242-243: If the model underestimates/simulates 30-50% lower precip than observations, does this bias mean anything for simulation of drought in the region? I ask because the authors mention land-atmosphere feedbacks - if the land is already 'too dry' in the model, are there implications for the life cycle of droughts (e.g., is the model somehow unrealistically 'on the edge' of setting off land-atmosphere drought feedbacks that wouldn't occur if the model just simulated a slightly wetter background climate regime that was more realistic?)

**Although the mean background climate is drier than the observed value, droughts and the variables for land-atmosphere feedbacks (anomalies of temperature, geopotential height and soil moisture) are**

**defined with respect to the long-term mean climate. If the model showed slightly wetter climate, these droughts and land-atmosphere feedbacks would be re-defined based on the new mean condition. Thus, the result would not change much, unless the model simulates more wetter episodes that frequently deviate from the mean climate. However, in that case, these continuous wetter episodes would be reflected to a change in the annual cycle. In the end, when we compare the model with the observation (Section 3.1), we see that the difference in the mean climate between both datasets does not implicate more drought events in the model simulation (Fig 3).**

(21) Line 245-246: 'the means are statistically similar to one another'- as I mention in the general comments, this appears to be a non-sensical test- PDSI should be centered around zero, so testing the means are statistically different is not a test that the model is doing well. I am also not sure if the instrumental- model-OWDA comparison was done on time series that were re-normalized to the same time period, or if the distributions from the data normalized to the last millennium are compared to distributions of data normalized to the last 100 years- if the time series are normalized to different time periods, I would expect there could be differences in the mean, etc.

**We also addressed this issue in the response (2). For the last 100 years, the reference period of all datasets is the 1950 – 1979. As the time series are calibrated with respect to that 30-yr period, we do not expect them to have a mean zero over entire time period, even less when we consider only drought events.**

(22) Figure 3 caption:' The p-value from the t-student test between the summer scPDSI from CESM and OWDA is 0.28' – what exactly is being tested here? I assume it's the statistical tests mentioned in the Methods, but please clarify in figure caption as I'm not sure (e.g., that distributions are significantly different or what?

**Before the sentence 'The p-value […]', we added a sentence about the null hypothesis of the test: ' The t-tests are applied under the null hypothesis of an equal mean between two time series.'. As in this case, we used the t-tests and not the Mann-Whitney test (as we compare the entire time series of annual drought indices), the null hypothesis of comparing means between two sample distributions is correct.**

(23) Separately, the figure caption states the 'red points on each box show the data points' – so there are only 3-5 data points? If there are <5 data points, then what is the sense in showing box plots that are intended to summarize large numbers of data points (don't the boxplots just show the minimum, first quartile, median, third quartile, and maximum? So how can 3 points of data meaningfully produce a boxplot?). Please clarify.

**We agree that the boxplot is not appropriate for this case with few data points. As we added it simply for a visualization purpose, we eliminated the boxes on the data points in Fig. 3 in the revised manuscript.**

(24) Also, in panel (a) the observations are black, then in (b) the colors show different information- this seems unnecessary confusing as there's no figure legend in the boxplot panel. Can the authors use consistent colors in both panels?

**We changed the colors of points in the (b) to be the same as in the (a).**

(25) Finally, what time periods are shown in the different parts of panel (b)? Please label on figure and/or describe in caption- for example, Figure 5a is immediately much more clear with the labels above the different sections of boxplots.

**We added the labels and titles for the Fig. 3.**

(26) Lines 255-256: confusing wording: 'simulating droughts of few years long, and with longer duration than those from OWDA.'

**We modified the sentence to:**

**Lines 263 - 265: 'However, the model still shows to a certain extent its ability to reproduce persistent droughts of multi-year long duration. Additionally, the mean duration of droughts in CESM is longer than in OWDA.'**

(27) Line 266-267: 'tree ring based reconstructions tend to deviate in their spectral behavior' – what does this mean?

**We clarified the sentence by changing it to:**

**Lines 274 – 276: ' Moreover, tree ring-based reconstructions for droughts tend to overestimate low-frequency variability compared to the instrumental observations.'**

(28) Line 286: confusing wording: 'despite the fact presents some discrepancy to the observation exist'

**We changed the sentence to:**

**Lines 297 – 298: 'Although there are some discrepancies between the model simulation and the observation, the model is able to reproduce the climate conditions associated with the variability of present-day scPDSI.'**

(29) Lines 286-287: 'the model does not significantly underestimate the persistence of multi-year droughts' – yet my reading of Figure 3b in which the observations are compared to the CESM show that the CESM simulates droughts that are about half the duration as compared to obs (large differences in median, inner quartile range). Perhaps the authors mean CESM simulates longer droughts than the OWDA?

**Indeed, yes, it is associated with the temporal variability. Thus, we modified the sentence to:**

**Lines 298 – 299: 'In particular, the model is able to simulate multi-year droughts and these droughts have longer duration than those in OWDA.'**

(30) Line 291: suggest 'by focusing' (not and focusing)

**We changed this as suggested.**

(31) Lines 293-295: 'We do not aim to make a direct comparison between the proxies and the model simulation, as this cannot be made due to the different initial conditions' – this is fair, but according to the wavelet analysis shown later, the CESM PDSI variability appears to line up with large eruptions, whereas they don't in the OWDA – if we 'believe' the model simulation, the PDSI variability could be (partly) forced, meaning they should line up temporally in the proxy data and CESM, right?

Similar comment on lines 304-307: 'are not mainly driven by external natural forcings'

**We modified the analysis and interpretation of the results for this part according to (3).**

**The superposed epoch analysis (Fig. 7) and comparison of the transient and control simulations (Fig. 5) support that Mediterranean droughts are more associated to the internal variability than the volcanic frocings. Wet periods are followed after some large eruptions. Thus, the statement that in CESM, droughts are driven by the internal variability remains unaffected.**

(32) Line 314: suggest 'statistically indistinguishable', not 'indifferent' – both here and elsewhere (I think indifferent means 'unconcerned' or 'mediocre', not indistinguishable)

**Thanks for the correction. We corrected the word to 'statistically indistinguishable' or 'statistically similar'.**

(33) Lines 329-331- given the timescales of impacts of volcanic forcing (e.g., multi-year cooling, then recovery to 'normal', maybe with some impacts on AMO), would you expect to see imprints of volcanic eruptions visually on drought indices that have been smoothed with a 100-year running mean?

**We performed the analysis with the non-smoothed annual time series. The figure of 100-year running means is included in order to visualize better all the time series of drought indices together. We clarified the time series we used for the analysis in the caption of Fig 5.**

(34) Line 334: for figure 6, the authors have shown a wavelet coherence diagram here, with no time series (and no indication which variable is the 'first' or 'second' variable as they ask the reader to interpret for lead/lag/phase relationship in the caption)- I think it would be helpful to plot the time series of volcanic eruptions below or above the plots so the readers can 'see' when the volcanic eruptions occur- for example, I know there are large eruptions ~1257/1258, ~1450, ~1600-1650, and in the early 19th century. All of these time intervals/years happen to show significant coherence with simulated scPDSI variability at ~4-16 yr timescales in figure 6, but this information is not provided for the reader, so unless the reader regularly works with the last millennium forcing data, they may not notice that there are conspicuous areas of significant coherence around large eruptions.

**We added the time series of volcanic eruptions in the upper panel of Fig. 6.**

(35) Lines 334-336: 'are not uniform across the period frequency bands' - Would we expect the signals of statistically significant coherent variability between drought indices to be uniform across frequency bands? I think a lack of coherence across frequency bands does not show a lack of forcing response, it only shows a lack of forcing response at certain frequencies/periods. This relationship (at longer time periods/lower frequencies) could make sense given the nature of simulated responses to volcanic eruptions in the North Atlantic in the NCAR CESM model. For example, Otto-Bliesner et al. (2016, BAMS, CESM LME documentation paper)- shows that the AMO/AMV is clearly impacted at decadal timescales by volcanic eruptions (see Figures 11 and 12 in Otto-Bliesner).

**We modified all this part according to (3).**

(36) Lines 339-341: 'the analysis confirms…not driven by the volcanic eruptions' – I strongly disagree with this interpretation- the figure seems to show me that the CESM PDSI, and soil variable all show 'significant' coherence with the 1258 eruption (and even some coherence with the 15th century, 17th century, and 19th century eruptions for PDSI) – the 'red' regions of coherence surrounded by black lines to show significance at ~4-16 year periods is pretty hard to miss.

**The information I take away from this figure is that the OWDA does NOT show coherence with eruptions, but the CESM does, which suggests the forcing is implemented wrong in the model, or that OWDA data aren't picking up on volcanic eruptions.**

**We addressed this issue in the responses (3) and (31).**

(37) Lines 343-345: 'The focus on one indicator is motivated by the fact…' - I still don't see any indication that the 10cm soil variable actually reflects what is happening with deeper soil water in the model- as Berg et al. 2017 (GRL) show, 10cm soil water can basically just mimic what is happening in atmospheric-centric variables likes precipitation. For example, as Berg et al. (GRL, 2017) show in their Figure 1 and describe in the text: 'In contrast, projections of negative changes in total soil moisture are more muted, in both extent and amplitude. Regions of negative changes (e.g., southern U.S. and Central America, northern South America, Mediterranean region, and South Africa) display relative changes of reduced

amplitude compared to surface changes.' – so in fact, I would argue that unless the authors show that the deeper soil moisture column actually shows the same degree of water stress, this statement is not supported by these citations.

**We agree with the referee, that our statement is not supported by the citation. Thus, we modified the paragraph as:**

**Lines 375 - 379: 'We use SOIL as it reflects the regional hydrological balance associated with the precipitation and evapotranspiration. Another advantage of this index is that the variable is a direct output from the model, thus, it does not require any further step, except for calculating the anomalies, and statistical assumptions as other indices do. In addition, the SOIL overlaps full or a part of drought periods given by the other three indices, without significantly underestimating the multi-year duration of droughts.'**

**And we discuss more about this in the conclusion lines 600 - 609.**

**Lines 600 – 609: 'The reason is that most of the commonly used drought indices are based on a water balance that considers only the atmospheric moisture supply and demand, and these indices tend to overestimate drought risks in the future warming scenario (Berg et al., 2017, Mukherjee et al., 2018, Swann et al., 2016). Berg et al. (2017) found that surface-based indices indicate droughts, whereas the mean 3-meter soil moisture shows wet or relatively weak dry conditions compared to the surface level. In our study, we used the upper 10 cm soil moisture anomaly that partially reflects the water stress on plants. However, the upper 10 cm of soil level is not enough to fully assess the complex atmosphere-soil-vegetation interaction and the variability in the deeper levels of the soil. Thus, the upper 10 cm soil moisture used here also magnifies drought risks to some extent. However, the Mediterranean is one of the regions where still the depletion of soil moisture occurs both at the surface and in the mean 3-meter soil level, though the amplitude of the rate of decrease is reduced in the 3-meter soil moisture compared to the rate in the surface soil moisture (Berg et al., 2017).'**

(38) Line 362: suggest 'presents an average of 7.25 droughts per century'- because this is an average, right?

**We corrected the word as suggested.**

(39) Line 404: suggest 'in the following section' (not 'in the following')

**We changed the words to 'Here'.**

(40) Line 407: 'The phases of NAO and ENSO are defined with respect to the non-drought periods: the values below (above 75) percentile of NAO and ENSO during the non-droughts periods are considered as negative (positive) phases of NAO and ENSO respectively (Fig. 9).' – I am unsure of what to make of this- so the authors redefined standard indices based on non-drought years? What does this do to the time series/what is the reason to do this other than to maximize drought signal of NAO and ENSO? And how is this information meaningful for 'real world' NAO or ENSO (e.g., how can this information about NAO/ENSO

during drought be used if the indices have to be redefined during non-drought years? I can see how maps of mean differences in drought and non-drought years could make sense, but I don't know how to interpret the treatment of the indices).

**We addressed this issue in the response (4).**

(41) Line 418: Again, is this what the MW-U test is testing (difference in means?)

**We modified the MW-U test related texts according to (2).**

(42) Line2 425-426: 'The positive NAO occupies 49% in the initiation years, then it decreases throughout the development of droughts, falling to 29% in the termination years' – in terms of this being meaningful information for drought prediction or giving information about how these droughts initiate, it sounds like positive NAO occurs almost exactly half of the time at the start of drought, but not traditional NAO, but instead NAO as defined by NAO during non-drought years?

**Yes, 49% with respect to the relative threshold, meaning that there is more probability that the NAO with positive values (with more than 0.58 based on the table 1) will occur during droughts than during the non-drought period by 24% (as the occurrence of positive NAO during the non-drought period is always 25%, thus, 49% minus 25%). Analogously, zero to positive NAO will occur with 74% (49% + 25%) of probability during the initiation years of droughts, which reflects the shift of the distribution of NAO to positive values (as one can see in Fig. 10) compared to the distribution during the non-drought period.**

**In the revised manuscript, we included a sentence mentioning about this increase in the occurrence:**

**Lines 455 - 456: 'This shows that the occurrence of positive NAO almost doubles in the initiation years of droughts compared to the non-drought period (25% of the occurrence of positive NAO).'**

(43) Line 449 vs line 457: 'transition years' vs 'transient years'- suggest consistent usage of terminology, here it makes sense (transition years), but in other locations (e.g., line 457), 'transient' is used (and in figure captions I think?)- this change in wording can be confusing, especially because the transient forcing/simulations terminology is also used, so I suggest removing wording that refers to transient as transition drought years, and just consistently use transition. (unless I completely don't understand and the authors intended there to be a difference)

**Thanks for the point. That was a typo.  We corrected all the 'transient' years to 'transition' years.**

(44) Line 469-470: About the time series shown in Figure 12 – these would seem to suggest that the Mediterranean region is in a long-term drought relative to the last millennium- the smoothed time series for SOIL never reach pre-industrial moisture levels after ~1860 AD- does this mean that climate change has caused a long-term drought that has lasted for ~150 years? I am not a Mediterranean climate expert, so I plotted GPCCv2018 annual precipitation, as well as Dai NCAR PDSI over the relevant time periods for the Mediterranean region and see no noticeable long-term trend in precipitation, and either a drought or

a 'step function' in PDSI in the late 20th century (again, no long-term aridification trend as the CESM seems to simulate). Can the authors plot the instrumental time series in the background to show if the model exceeds the envelope of variability in the instrumental data and/or if the trends are present in instrumental data too?

**We added the plot of times series of scPDSI from the meteorological station data (U.Delaware v5.01), CESM and OWDA in Fig. 3.(c) according to the comment (3). The scPDSIs here are calculated relative to 1950 – 1979 AD. In the figure, all three scPDSIs shows decreases during the last 100 years and also during last 50 years. These negative trends are statistically significant at 95% based on Mann-Kendall trend tests (null hypothesis of the test is that there is a no monotonous trend in the time series). Obviously, the magnitudes of these trends cannot be compared to the magnitudes of the trends for SOIL and precipitation in Fig. 12, which are calculated relative to the 850 years before the pre-industrial period.**

**Long-term decreases in precipitation and drought indices over the west and central Mediterranean region have been already reported in several studies: with scPDSI, Sousa et al. (2011), with SPEI and SPI, Spinoni et al. (2017), and with precipitation datasets, Nunes and Lourenço (2015) over Portugal, Valdes-Abellan et al. (2017) over southeastern Spain, Caloiero et al. (2018) and Philandras et al. (2011) over the entire Mediterranean region among others.**

*Caloiero, Tommaso, Paola Caloiero, and Francesco Frustaci. "Long-term precipitation trend analysis in Europe and in the Mediterranean basin." Water and Environment Journal 32.3 (2018): 433-445.*

*Nunes, A. N., and L. Lourenço. "Precipitation variability in Portugal from 1960 to 2011." Journal of Geographical Sciences 25.7 (2015): 784-800.*

*Philandras, C. M., et al. "Long term precipitation trends and variability within the Mediterranean region." Natural Hazards and Earth System Sciences 11.12 (2011): 3235-3250.*

*Sousa, Paulo M., et al. "Trends and extremes of drought indices throughout the 20th century in the Mediterranean." Natural Hazards and Earth System Sciences 11.1 (2011): 33-51.*

*Spinoni, Jonathan, Gustavo Naumann, and Jürgen V. Vogt. "Pan-European seasonal trends and recent changes of drought frequency and severity." Global and Planetary Change 148 (2017): 113-130.*

*Valdes-Abellan, Javier, M. A. Pardo, and Antonio José Tenza-Abril. "Observed precipitation trend changes in the western Mediterranean region." International Journal of Climatology 37 (2017): 1285-1296.*

(45) For Figure 13, please define the acronyms ND and D (I assume this is Detrend and Non-Detrended, but this is not explicitly defined)

**We modified the 'ND' to non-detrend' and 'D' to 'detrend' in the legend of Fig. 14 and also defined them in the caption.**

(46) Line 504: suggest word other than 'indifference'

**As suggested in (32), we corrected this word in the manuscript.**

(47) Lines 505-507: 'this result shows that the natural mechanisms associated with droughts remain the same…' ok, in so far as they are defined by circulation patterns, but what about increases in evapotranspiration/aridification due to increasing temperatures? This sentence basically is contradicted by the next sentence, which states that the mechanisms are anthropogenically driven- can the authors distinguish/clarify? The authors have shown how different drought drivers progress in Figure 11, but couldn't EV change in the future, thus the droughts would not have 'natural mechanisms'?

**We agree that we were not very clear with this part. Here, our questions are how the mechanisms associated with droughts change during this period because of the anthropogenic influences and whether besides the anthropogenic effects, there are some natural effects that drive changes in droughts and drought associated mechanisms. The linear detrending was performed in order to see this point, considering that the linear trends in drought related variables are caused by the anthropogenic influences.**

**By the end, we confirmed that the dryness in the future is mainly due to the anthropogenic influences on the land – atmosphere feedback and not by changes related to circulation patterns.**

**We modified the paragraphs on introducing the motivation and discussing the result to clarify our point:**

**Lines 506 - 510: 'We examine whether the mechanisms associated with Mediterranean droughts described in the previous section are affected by the anthropogenic influences on climate and whether these changes contribute to the intensification of droughts and eventual aridification in the region occurring in this period. For this, the detrending method is applied to the simulation following the steps mentioned in Sect.2.2. First, we analyze the non-detrended drought related variables with the anthropogenic influences on them; then, the detrended variables to see the background climate during droughts excluding the linear trends.'**

**Lines 536 - 540: ' Hence, when the linear trends, thus the anthropogenic influence, are not taken into account, the mechanisms involved in droughts remain unchanged during this period which indicates that no other factor than the anthropogenic influences in temperature is the cause of the severe dryness in 1850 - 2099 AD. In the future scenario, the intensities of both land - atmosphere feedbacks are magnified due to the increases in GP and TS caused by the increases in GHG, and these feedbacks become the dominant one at controlling the desiccation over the region.'**

(48) Also, again, there is no precipitation and/or obs-based soil moisture/PDSI shown here- do the observations show the same general trends in terms of long-term aridification? If not, this an important thing to point out, if so, then great- the model is doing well, and this should be noted.

**We addressed this issue in the response (44).**

(49) For Figure 7 caption: 'the regions where the means between the control and transient simulations are statistically not significant at 5% confidence level'? - this seems like the wrong test here- are we interested in the means being the same, or are we interested in where the 'spread' from internal variability in the control run is different than the forced run spread?

**We modified this as 'distribution' now in Fig. 8. Also refer to our response (2).**

(50) Line 515: suggest changing wording to 'although our result shows' or the sentence is incomplete/comma splice

**We modified the sentence as suggested.**

(51) Lines 525-529: Authors conclude there is no 'causal connection between volcanic eruptions and dry conditions', but their wavelet figures indicate that volcanic eruptions are significantly coherent with PDSI and soil moisture variability at ~4-20yr periods around large eruptions. I agree that if we average all the geopotential height patterns during drought in the control and forced simulations, there may be minimal differences, but not all droughts/pluvials occur during eruptions, so any anomalous behavior after eruptions could be 'averaged out' by the large numbers of PDSI anomalies (droughts/pluvials) that do not occur after eruptions.

**We addressed this issue in the response (3).**

(52) Lines 553-554- These results would be much more believable if the authors showed GPCC/CRU/UDel precipitation on top of the model precipitation time series, and instrumental-based scPDSI on the CESM PDSI time series to show the model is getting the timing and magnitude of trends right: https://psl.noaa.gov/data/gridded/data.pdsi.html

**We addressed this issue in the response (1).**

(53) Lines 574-575: again, the authors are choosing to study a region that Berg et al. have shown has 10cm soil moisture that magnifies droughts and does not reflect what is happening in 'full column' soil moisture (to ~3m depth) – so bringing up that the authors have used 10cm soil water isn't really showing that they have got around this problem.

**We agree with the referee that the statement is not supported by the citation. Thus, we corrected the paragraph as:**

**Lines 602 - 609: 'Berg et al. (2017) found that the surface-based indices indicate droughts, whereas the mean 3-meter soil moisture shows wet or relatively weak dry conditions compared to the surface level. In our study, we used the upper 10 cm soil moisture anomaly that partially reflects the water stress on plants. However, the upper 10 cm of soil level is not enough to fully assess the complex atmosphere-soil-vegetation interaction and the variability in the deeper levels of the soil. Thus, the upper 10 cm soil moisture used here also magnifies drought risks to some extent. However, the Mediterranean is one of**

the regions where still the depletion of soil moisture occurs both at the surface and in the mean 3-meter soil level, though the amplitude of the rate of decrease is reduced in the 3-meter soil moisture compared to the rate in the surface soil moisture (Berg et al., 2017).'

---

## Author Response (AR3)

Dr. Hugues Goosse, editor
Climate of the Past

Woon Mi Kim
Physics Institute and Oeschger Centre for Climate Change Research, University of Bern
Sidlerstrasse 5, 3012 Bern, Switzerland

15th March 2021

Dear Dr. Hugues Goosse,

We would like to thank you again for this opportunity to resubmit our manuscript and also many thanks to the reviewer for the detailed feedbacks. We greatly appreciate the time that the editor and the reviewer dedicated to our manuscript. We have addressed all the reviewer's comments carefully. Please find below our response to the reviewer.

Sincerely,

Woon Mi Kim[*] and Christoph C. Raible

[*]Corresponding author: woonmi.kim@climate.unibe.ch

Response to the referee #2

*General Comments:*

*1) I commend the authors for improving their paper and conducting extra analyses that support their statements/conclusions. I think this paper will be an important contribution to the literature after some minor wording clarifications.*

**Thanks very much for your comment.**

*I have several relatively minor wording and figure clarification suggestions, listed by line number or Figure number below:*

*2) Line 3-5: thank you for adding some wording about potentially relevant model weaknesses*

*3) Line 21: 'induce a constant dryness' – suggest adding relative to what time period*

**We added 'with respect to the 1000 – 1849AD' in line 22:**

**Lines 21 - 22: 'These feedbacks are intensified during the period 1850 – 2099 AD due to the anthropogenic influence, thus reducing the role of modes of variability on droughts in this period. Eventually, the land - atmosphere feedbacks induce a constant dryness over the Mediterranean region for the late 21st century relative to the period 1000 – 1849AD.'**

*4) Introduction up to Line 60: nice introduction to Mediterranean drought and previous literature*

**Thanks very much.**

*5)Line 84: suggest 'paleoclimate' proxies instead of natural (as opposed to synthetic?)*

**We modified the word to 'paleoclimate' as suggested.**

*6) Line 89-95: 'The results show…in the middle years'- I initially asked the authors to acknowledge Parsons/Coats and Namias because they also examine drought trajectories. This description is a bit long given the paper is about Med. drought, not North American drought, so I would remove all of this summary of Parsons/Coats except to state that others have examined drought trajectories before (but in other locations or time periods).*

**We deleted the summary about Parson and Coats (2019) and Namias (1960). Instead, we briefly mentioned the work from Parson and Coats (2019) together with other literature on the U.S droughts in lines 86 - 88:**

**Lines 86 - 88: 'Modelling studies on the long-term variability of droughts are focused on the U.S continent, mainly to investigate the variability and mechanisms of South Western United States (SW) droughts (Coats et al., 2013, 2016; Parsons et al., 2018; Parsons and Coats, 2019) and North American pan-continental droughts (Coats et al., 2015; Cook et al., 2016b).**

*7) Line 134: 'guarantees' seems like a very strong word for the benefits of slightly higher model resolution I the CESM1- I suggest ''provides'' or something along those lines instead*

**We changed the word to 'can improve'. Now the sentence is:**

**Line 130 – 131: 'Hence, using a model that provides a seamless simulation for the period 850 – 2099 AD with a relatively finer spatial resolution can improve a representation of precipitation related processes.'**

*8) Line 143: 'we present a conclusions'- fix wording*

**We corrected the sentence to:**

**Lines 138 - 139: Finally, the conclusions are presented in Section 4.**

*9) Line 148:'where the 2005…RCP8.5'- what about the 150 yrs before this? Why specifically state it's RCP, then not say anything here about historical/last millennium forcing.*

*Lines 156-159: this paragraph seems to fit better with the phrase/description of rcp forcing in the first paragraph of section 2.1. suggest combining so forcing is described all together instead of in pieces*

**We modified the text as suggested. We removed 'where the 2005 …' and added the new text in lines 154 - 157.**

**Lines 142 - 143: 'Two simulations are used: a continuous transient simulation of 1250 years (850 - 2099 AD) and a control simulation of 400 years at perpetual 850 AD conditions'**

**Lines 153 - 156: '[…] Then, the period 2005 – 2099 AD is run with the RCP 8.5 scenario. The transient forcing follows the third Paleoclimate Modelling Intercomparison Project (PMIP4; Schmidt et al.,2012)– fifth Coupled Model Intercomparison Project (CMIP5; Taylor et al., 2012) protocols.'**

*10) Line 161: suggest removing 'confined to'*

**We removed 'confined to' and put in parenthesis the latitudes and longitudes of the region.**

**Line 159: 'The focus area of the study is the western and central Mediterranean region (15 W −28 E and 33 -45N; Fig. 1).'**

*11) Line 165: The authors used the EOF of instrumental data to define a region, then use this region to study dynamics in the model- this is fine, but note that when I have looked at the EOF of precipitation in other regions, the instrumental and cesm leading EOF modes/patterns don't always essentially show the same spatial distributions.*

**Thanks for your comment. We assume that the difference in precipitation between models (not only in CESM but also in many other GCMs) and observation are expected due to the model difficulties at representing the precipitation (mostly in regional scale) well. In our analysis, we also noted that there are some differences in spatial and temporal distributions of precipitation between the model and observation over our region of studies (Fig 2). We mentioned these differences and also statistical similarities between the model and observation in the manuscript to justify the usage of CESM (Sect. 3.1., lines 245 – 250.)**

*12) Line 174-175: I think it would be helpful to clarify what the MW U test is testing (e.g. null hypothesis)- authors later in paper present p-values, etc, but the reader is never told how to interpret these (e.g., does a low p value mean the distributions are or are not statistically distinguishable? In other words, is the null hypothesis that the two distributions are distinct or similar?)*

**We added in lines 172 - 173 a sentence mentioning about the null hypothesis of MW U test:**

**Lines 172 - 173: 'The null hypothesis of a Mann-Whitney U-test states that the distributions of both populations (in this case, the transient and the control simulations) are equal.'**

**Hence, a low p-value means rejecting the null hypothesis, indicating that the two distributions are different to each other.**

*13) Methods up to Line 187: thank you, nice overview of the Methods*

**Thanks very much for the comment.**

*14) Line 208: 'area weighted averaged' seems awkward– are the authors trying to state 'the area-weighted average is calculated'?*

**We changed the sentence to:**

**Line 206: 'Then, the area- weighted average of each index is calculated over the Mediterranean region.'**

*15) Line 212: '10 percentiles' – suggest '10th percentile'*

**We corrected the word as suggested.**

*16) Line 213-215: this sentence is long and seems to run several phrases together that need to be separated. Suggest: 'This method, which imposes…extreme percentiles,…drought. Thus, we only'*

**We corrected the sentence as suggested.**

*17) Line 221-222: 'This separation…Parsons and Coats (2019)'- see my comment above- I'd suggest just using this sentence/citation here, and eliminating most of the sentences giving details about North American drought in the Introduction.*

**We left this citation in the Method section as it is and modified the introduction according to the comment #6.**

*18) Line 233: what is 21th century? Suggest '20th'*

**Thanks for the point. It was 20th and we corrected it.**

*19) Line 235: change 'associated to droughts' to 'associated with droughts'*

**We corrected 'to' to 'with'.**

*20) Lines 245-250: thank you for assessing model performance*

**We would like to thank you very much for your detailed comments about the model performance during the previous revision phases.**

*21) Line 252-253: 'This is also the case…(p values of 0.01)'- please clarify 'this' and also see above note on p-value interpretation and null hypothesis clarification for reader. Also, the previous sentence discusses a student t-test, so is this sentence discussing the M-W U test, student t-test, both?*

**We noted that we did not mentioned that we used the t-test to compare the time series of scPDSI for the entire present period (1901 – 2000) in the Method section. Thus, we added in the Section 2.4 in lines 236 - 237, a sentence stating about t-tests we used.**

**Lines 236 - 237: 'The time series of scPDSI during this period are statistically compared to each other using the t-tests with the null hypothesis of equal means between two time series.'**

**Also note that we slightly rearranged the sentences in this Section (Sect. 2.4).**

**For the lines 252 – 253 (now in line 251), we changed 'this' to 'the same'. We also slightly modified the paragraph to clarify the content.**

**Lines 250 - 252: 'Comparing the time series of scPDSI (Fig. 3. (a) and (c)), the summer means between the model and OWDA are statistically similar to each other (p-value from the t-test of 0.28). The same happens between the OWDA and observation but with much lower confidence level of 1\% (p value of 0.01).'**

*23) Lines 254-257: thank you for adding this description/discussion*

**Thanks very much again to your comments during the previous revision phases.**

*24) Line 273: 'annually averaged OWDA'- seasonal? but at annual resolution?*

*OWDA website is JJA, not annual (but at annual resolution)- please clarify for reader unfamiliar with OWDA*

**Thanks for the point. We added 'summer' in the sentence. Now, the sentence is:**

**Line 272: 'The annually resolved summer (JJA) OWDA is based on tree ring reconstructions, which are known to be biased towards the growing season.'**

*25) Section 3.1: in general this model validation section is much improved and provides nice context for the reader about CESM performance specific to Med drought*

**Thanks very much again to your comments during the previous revision phases.**

*26) Line 310: '100 years running means'- suggest '100-year running means'*

**We corrected the word as suggested.**

*27) Line 348-349: I disagree with the statement 'we expect to see a similar response…if droughts are influenced by the same externally forced variability' – I think there's a logical mis-step here. If the same drivers were influencing each drought metric this would be true, but as the authors describe, the metrics use different inputs/variables- one is just basically precip, the others include evaporative demand. Soil water/PDSI is influenced by temperature and evaporation- after an eruption, cooling could decrease evaporative demand, and impact evaporation sensitive metrics (and not precip unless some large-scale circulation changes occur, I would imagine). The analysis in the next paragraph seems to support this very point (volcanic eruptions are followed by wet periods in SOIL and PDSI). So would we actually expect all metrics to co-vary with forcing?*

**Thank you for your comment. We agree with the problem that the referee points out and we were not aware of this during the previous revision phases. As we also mention in the manuscript, as all drought metrics use different inputs and calculation schemes, some are more sensitive to the changes in some input variables (for example, SPEI), while some are not (for example, PDSI) (Vicente-Serrano et al., 2015) and this sensitiveness also depends on the region of study (Raible et al., 2017). Thus, as the referee says,**

**it is expected that drought indices respond differently to the changes in precipitation and temperature caused by the eruptions.**

**Here we removed the sentence 'we expect to see … ' (Previously in lines 348 - 349) and changed lines 347 - 349:**

**Lines 346 - 349: 'In terms of the timing of the occurrence of droughts over the period of 850 - 1849 AD (Fig. 5. (d)), coherent changes among all indices are not identified, which is expected due to the different input variables and calculation schemes among drought indices. This fact also indicates that each index responds differently to the changes in precipitation and temperature, hence potential evapotranspiration, caused by the externally forced variability, e.g., the volcanic forcing.'**

*Raible, C. C., Bärenbold, O., & Gomez-Navarro, J. J. (2017). Drought indices revisited–improving and testing of drought indices in a simulation of the last two millennia for Europe. Tellus A: Dynamic Meteorology and Oceanography, 69(1), 1287492.*

*Vicente-Serrano, S. M., Van der Schrier, G., Beguería, S., Azorin-Molina, C., & Lopez-Moreno, J. I. (2015). Contribution of precipitation and reference evapotranspiration to drought indices under different climates. Journal of Hydrology, 526, 42-54.*

*28) Lines 383-384: 'Hence…other indices'. I would avoid saying this as you show in Figure 5D and make a strong case earlier that these drought indices are generally dissimilar. You also state 'in terms of timing of the occurrence of droughts...coherent changes are not identified'- so are the drought metrics co-varying or not? These sections/findings/conclusions still seem to conflict with one another. The previous sentence also seems to show the only ¼ of droughts overlap using these different metrics.*

**We also mentioned in line 379 that the SOIL overlaps not the full period but 'the 36%, 25% and 29% of droughts in the scPDSI, SPEI and SPI, respectively' (which is roughly ¼ as the referee mentions), also showing quite acceptable values of Pearson correlations with other indices over the entire period (lines 381 – 382). Thus, we assume that not fully but a part of the mechanisms given by SOIL can be transferred to other indices.**

**With this reason, we slightly modified the sentence by adding the word 'partially':**

**Lines 382 - 383: ' Hence, the results in the following sections can be partially transferred to the other indices.'**

*29) Section 3.3: thanks for comparing cesm control and forced drought frequency etc per century- much clearer*

**Thanks very much again to your comments during the previous revision phases.**

*30) Lines 389-390: these are the composites of all drought years, right? Might be helpful to clearly state this section/paragraph present results for all years, and that after you will break up analysis into different parts of drought*

**We reformulated the sentence by adding that we analyzed the entire drought events in line 390. Now the sentence becomes:**

**Lines 389 - 390: 'To get a first glance of the atmospheric circulation during drought conditions, we analyze the mean circulation conditions during all short (1 and 2 years of duration) and long (more than 3 years) Mediterranean droughts together.'**

*31) Lines 400-405: what about the significant differences in temperature in the maps shown in the temperature maps in Figure 8*

**Thanks for your point. We did not include a discussion of the differences in temperature as we focused more on the regions with similarities knowing that the internal variability is the potential drivers of Mediterranean droughts (from the result of the previous Section 3.2). Moreover, the region with significant differences in temperature anomalies show rather weak values of anomalies (for example, over North African and Asian continents). The difference in the temperature anomalies over Siberia is quite striking. However, this difference is not reflected in the difference in the circulation patterns (geopotential heights) between the control and transient simulations. Thus, we do not expect that this temperature pattern could remotely affect the Mediterranean climate.**

**In the revised manuscript, we briefly mention about the statistical dissimilarities in the temperature field in lines 407 - 412.**

**Lines 407 - 412: 'Some statistically significant dissimilarities between the control and transient simulations are noticeable mostly in the temperature anomalies. Over the regions where the temperature anomalies are statistically different the anomalies are rather weak, except the warming in Siberia. However, this positive temperature anomaly in Siberia is not associated with a geopotential height pattern, showing statistically indistinguishable geopotential height anomalies in the region between the control and transient simulations (Fig. 8). This indicates that there is no change in the circulation pattern over this region that can possibly be connected to the Mediterranean drought condition.'**

*32) Line 425-426 'Figure 7'- Figure 8?*

**Yes, Figure 8 is correct. We corrected the number in the revised manuscript.**

*33) Line 439-440: can you clarify these are composites of all drought years, to contrast w transition etc years later?*

**We modified the sentence to:**

**Lines 446 - 447: 'Considering all short and long droughts, the simulation shows that droughts occur more frequently during the positive phase of annual and winter NAO than during the non-drought period'**

*34) Line 447: 'more frequent' – change to 'more frequently'*

**We corrected the word as suggested.**

*35) Line 454: '49% in' suggest change to '49% of'*

**We corrected the word as suggested.**

*36) Line 457: suggest 'In the case of ENSO'*

**We corrected the word as suggested.**

*37) Line 458: relative to what background frequency? You did a nice job contextualizing the background frequency of NAO relative to non-drought periods, what about La Nina here?*

**We added 'relative to the non-drought period' at the end of the sentence in line 463.**

**Line 464: 'the frequency of La Niña-like is 40% in the initiation years relative to the non-drought period.'**

*38) Line 475: 'some mechanisms associated with this circulation is'- change to 'are'*

**We corrected the word as suggested.**

*39) Line 479: 'This is supported' – suggest 'This mechanism is supported' or other clarifying wording*

**We modified the sentence to:**

**Lines 485 - 487: 'The presence of the atmosphere - soil interaction during the transition years is supported by the increases in frequencies of positive surface temperature (TS) and sensible heat flux (SH), and negative soil moisture (SOIL), evapotranspiration (EV), and latent heat flux (LH) anomalies during this period'**

*40) Line 502: suggest change 'noticed' to 'apparent'*

**We changed the word as suggested.**

*41) Line 518: suggest 'reflected by' (not 'to')*

**We changed the word as suggested.**

*42) Line 523: please clarify if you mean a 'reduction in positive NAO and a reduction in La Nina like conditions', or 'a reduction in NAO, and general La Nina like conditions during drought'- right now, the wording seems ambiguous*

**The first one is correct. Thus, we modified the sentence as:**

**Lines 528 - 529: 'the non-detrended 1850 – 2099 AD period shows reduced frequencies of both positive NAO and La Niña-like conditions during droughts'**

*43) Line 526-527: 'Hence, the role of ..La Nina..': This explanation makes sense, but note that many CMIP5-class models also generally become more 'El Nino like' (weaker tropical Pacific E-W gradient) with warming, so this could be also just caused by the fact that global warming could separately impact both La Nina and local drought, right? See for example Seager et al who show that most cmip5 models become more el nino like:*

*https://doi.org/10.1038/s41558-019-0505-x*

**Thanks for the suggestion. We extended the paragraph discussing about this bias in CMIP5 model, and this issue may have some implication in our results in lines 533 - 538.**

**Lines 533 - 538: 'Note however, that model biases in representing large scale modes of variability, in particular ENSO, might be relevant. Many CMIP5 models have problems in realistically reproducing the cold SST in the east Tropical Pacific. Therefore, these models would show less La Niña events in the future warmer conditions (Seager et al. 2019). An overall increase in El Niño-like condition here (Fig. 14. (b)) can also be partially related to this bias. Nevertheless, considering this bias does not affect the result that the interaction among the regional variables is changed by the increase in temperature, causing more intensified regional land-atmosphere feedbacks during this period.'**

*44) Line 536: 'Hence, when the linear trends…are not taken into account' by this do you mean not removed, or if they are removed? please clarify wording*

**To clarify this part, we modified the sentence as:**

**Lines 547 - 549: 'Hence, when the anthropogenic effect is removed (i.e., the variables are detrended), the mechanisms involved in droughts remain unchanged during 1850 - 2099 AD. The comparison of non-detrended with the detrended variables, thus, indicates that no other factor than the anthropogenic influence in temperature is the cause of the severe dryness in this period.'**

*45) Line 579-581: 'Moreover…warming scenario' – good point, may be good to cite Seager et al (see above) or another similar paper here that discusses this point*

**We cited and included a brief discussion on Seager et al. (2019) in lines 592 - 594. Also, refer to our response #43.**

**Lines 592 - 594 : 'Many CMIP5 models show an overall warming of the Tropical Equatorial Pacific reducing the west-east gradient of SST which is different from what is observed in the present period (Seager et al., 2019). This model bias to the observation implicates reduced La Nina condition in CMIP5 models in a warmer world.'**

*46) Line 603: 'surface-based indices indicate droughts': suggest clarifying wording as this is confusing unless you have read Berg et al 2017*

*- for example, what is 'surface' here for the reader who isn't familiar with this paper/topic?*

*Overall, nice discussion of this topic.*

**We wanted to refer the upper-level surface soil moisture. We changed the word in line 617, and modified slightly the paragraph in lines 614 - 618:**

**Lines 614 - 618: 'The reason is that most of the commonly used offline drought indices, such as scPDSI, are based on a water balance that considers only the atmospheric moisture supply and demand, and these indices tend to overestimate drought risks in the future warming scenario (Berg et al., 2017; Mukherjee et al., 2018; Swann et al., 2016). Moreover, Berg et al. (2017) found that the upper-level soil moisture indicates droughts, whereas the mean 3-meter soil moisture shows wet or relatively weak dry conditions compared to the surface level.'**

*47) Line 610: 'followed'- suggest 'conducted' or similar wording change*

**We changed the word to 'conducted' as suggested.**

*48) Line 614: 'under the current climate change scenario': does this mean a specific SSP/RCP from CMIP, or just if current warming trends continue? 'scenario' is a potentially loaded word given the modeling focus of this paper. Suggest clarification.*

**We meant under the RCP warming scenarios (2.8, 4.5 and 8.5) based on Giorgi et al. (2008) and Lehner et al. (2017). Thus, we changed the word to 'under the future warming scenarios' in line 629.**

*49) Figure 2: suggest labeling 'summer' and 'winter' on maps, also labeling the dashed and solid lines/graphs with legends for easier reference*

**We added the labels and legend in the figure.**

*50) Figure 4: 'composites' of what?*

**We modified this part as 'Pearson correlation coefficients between the scPDSI and anomalies'.**

*51) Also, what is CR21? Do the authors mean 20th century reanalysis? Isn't this then CR20?*

**Yes, it is CR20. We corrected it.**

*52) Figure 5: generally, much improved and comparison is helpful for control vs transient- thank you*

*Figure 6: thank you for including the time series- much easier to interpret wavelet/coherence*

**Thanks again for your comments during the previous revision phases.**

*53) Figure 8: Interesting how distinctive the temperature patterns are during droughts in control vs forced runs (assuming I'm interpreting the stippling correctly)- is this mentioned in the main text?*

**We addressed this issue in our response #31.**

54) Also, 'distributions' – do the authors mean to say 'differences in distributions' (otherwise, what does it mean for a distribution to be significant?)

**We modified the sentence to 'where the distributions between the control and transient simulations are statistically different to each other.'**